



**Bidirectional coupling of a long-term integrated assessment model**
**REMIND v3.0.0 with an hourly power sector model DIETER v1.0.2**
Chen Chris Gong[1], Falko Ueckerdt[1], Robert Pietzcker[1], Adrian Odenweller[1,3], Wolf-Peter Schill[2], Martin
Kittel[2], Gunnar Luderer[1,3]
[1] Potsdam Institute for Climate Impact Research (PIK), Potsdam, Germany
[2] German Institute for Economic Research (DIW Berlin), Berlin, Germany
[3] Global Energy Systems Analysis, Technische Universität Berlin, Berlin, Germany
*Correspondence to*: Chen Chris Gong (chen.gong@pik-potsdam.de)
**Abstract.** Integrated assessment models (IAMs) are a central tool for the quantitative analysis of climate change mitigation
strategies. However, due to their global, cross-sectoral and centennial scope, IAMs cannot explicitly represent the temporal and
spatial details required to properly analyze the key role of variable renewable electricity (VRE) for decarbonizing the power
sector and enabling emission reductions through end-use electrification. In contrast, power sector models (PSMs) can
incorporate high spatio-temporal resolutions, but tend to have narrower sectoral and geographic scopes and shorter time
horizons. To overcome these limitations, here we present a novel methodology: an iterative and fully automated soft-coupling
framework that combines the strengths of a long-term IAM and a detailed PSM. The key innovation is that the framework uses
the market values of power generations as well as the capture prices of demand flexibilities in the PSM as price signals that
change the capacity and power mix of the IAM. Hence, both models make endogenous investment decisions, leading to a joint
solution. We apply the method to Germany in a proof-of-concept study using the IAM REMIND v3.0.0 and the PSM DIETER
v1.0.2, and confirm the theoretical prediction of almost-full convergence both in terms of decision variables and (shadow)
prices. At the end of the iterative process, the absolute model difference between the generation shares of any generator type for
any year is <5% for a simple configuration (no storage, no flexible demand) under a "proof-of-concept" baseline scenario, and
6-7% for a more realistic and detailed configuration (with storage and flexible demand). For the simple configuration, we
mathematically show that this coupling scheme corresponds uniquely to an iterative mapping of the Lagrangians of two power
sector optimization problems of different time resolutions, which can lead to a comprehensive model convergence of both
decision variables and (shadow) prices. The remaining differences in the two models can be explained by a slight mismatch
between the standing capacities in the real-world and optimal modeling solutions purely based on cost competition. Since our
approach is based on fundamental economic principles, it is applicable also to other IAM-PSM pairs.
**1 Introduction**
Thanks to decade-long policy support in many regions of the world and technological learning, the costs of both wind power
and solar photovoltaics have plummeted (IEA, 2021; Lazard, 2021). These types of variable electricity generation are now
highly cost competitive against other alternatives, such that their deployment is increasingly driven by market forces instead of
climate policies. Among the newly added renewable generations in 2020, nearly two thirds were cheaper than the cheapest new
fossil fuel (IRENA, 2020). Due to both cost declines and pressing concerns over climate change, investing in these clean and
abundant resources has become a crucial part of national and regional strategies to decarbonize the power sector (The White
House, 2021; Cherp et al., 2021; National long-term strategies, 2022; Rechsteiner, 2021; ICCSD Tsinghua University, 2022).



Given this dramatic development in the power sector over the past two decades, a universal consensus has emerged among
energy transition scholars and policy makers: emissions in the power sector are relatively "easy-to-abate" (Luderer et al., 2018;
Azevedo et al., 2021; Clarke et al., 2022). Compared with other primarily non-electrified end-use sectors such as buildings,
transport and industry, the technologies required to transform the power sector are low-cost, mature and readily available. This
trend has in recent years led to a second emerging consensus: the power sector will be the fundamental basis of a future low-
cost, efficient and climate-neutral energy system (Brown et al., 2018b; Ram et al., 2018; Ramsebner et al., 2021; Luderer et al.,
2022a). In addition to direct electrification, which requires end-use transformations of currently non-electrified demand,
emerging technological developments in hydrogen and e-fuels produced from renewable electricity have also contributed to the
broadening of potential technology portfolios for the "hard-to-abate" sectors, such as high temperature heat and chemical
productions (Parra et al., 2019; Bhaskar et al., 2020; Griffiths et al., 2021). Together, direct and indirect electrification support a
broad concept of "sector coupling", which facilitates decarbonization by powering end-use demand with variable renewable
energy sources (Ramsebner et al., 2021).
Due to the pivotal role of electrification and sector coupling in mitigation scenarios, there is an increasing demand on the scope
and level of detail of energy-economy models used to guide the energy transition and climate policies. The models would
ideally encompass a global, multi-decadal and multi-sectoral scope, such that the scenarios are relevant for international and
regional climate policies, while simultaneously incorporating a high level of spatio-temporal detail. The latter is important to
account for the specifics of variable renewable electricity generation as well as its physical and economic interplay with the
electrification of energy demand (Li and Pye, 2018; Brunner et al., 2020; Prol and Schill, 2020; Böttger and Härtel, 2022;
Ruhnau, 2022). This need for improved modeling methods or frameworks, which has to overcome the trade-off between scope
and detail, is a substantial methodological challenge. It entails realizing two main objectives:
Objective 1) Accurately model the power sector transformation over long time horizons in terms of investment and dispatch,

especially at high shares of variable renewable energy (VRE) sources. Long-term pathways for the following power

sector quantities and prices should accurately incorporate short-term hourly details:

a)   capacity and generation mix of the power sector,

b)   market values (annual average revenues per power generation unit) for variable and dispatchable plants,

c)   capacity factors of the dispatchable plants and the curtailment rates of variable renewables,

63        d)   storage capacity and dispatch.

Objective 2) Accurately model direct electrification of end-use sectors as well as indirect electrification technologies such as

green hydrogen production, where existing and emerging sources of power demand can be in-part flexibilized.

## 1.1 Current modeling approaches and limitations

Current energy system models broadly fall into two distinct categories, carried out by two research communities with little
institutional overlap: integrated assessment models (IAMs) and power sector models (PSMs), each with its own strengths and
weaknesses. IAMs are comprehensive models of global scale and span multiple decades, linking macroeconomics, energy
systems, land-use and environmental impacts (Stehfest et al., 2014; Calvin et al., 2017; Huppmann et al., 2019; Baumstark et al.,
2021; Keppo et al., 2021; Guivarch et al., 2022), therefore providing an "integrated assessment" of multiple factors (Rotmans
and van Asselt, 2001). IAMs substantially shape the IPCC assessments on long-term climate mitigation scenarios, and play an
important role in policy making (Rogelj et al., 2018; UNEP, 2019; NGFS, 2020; P.R. Shukla et al., 2022). In comparison to
IAMs, PSMs typically have narrower spatial and sectoral scopes and shorter time horizons, but provide higher resolutions and
increased technological detail (Palzer and Henning, 2014; Zerrahn and Schill, 2017; Brown et al., 2018a; Ram et al., 2018;



Sepulveda et al., 2018; Blanford and Weissbart, 2019; Böttger and Härtel, 2022). This allow PSMs to more accurately model the
power sector under high VRE shares (Bistline, 2021; Chang et al., 2021). Note that we use the term "power sector model" here
to represent all general smaller-scope models than IAMs (usually by geographical or time horizon measures), even though many
of them have sector-coupling aspects and do not only contain the traditional power sector.

IAMs and PSMs are therefore limited by a lack of spatio-temporal detail and a lack of scope, respectively. IAMs usually have a
temporal resolution no shorter than a year (Keppo et al., 2021) and therefore include simplified representations of hourly power
sector variability, which mimic the real-world dynamics to varying degrees of success (Pietzcker et al., 2017). In general, a lack
of high temporal resolutions can lead to difficulties when estimating the optimal level of variable renewable generation, often
either over- or underestimating the market value of solar or wind generation, the challenges of variable renewable integration,
the peak hourly residual demand, and the need for energy storage and baseload (Pina et al., 2011; Haydt et al., 2011; Ludig et
al., 2011; Kannan and Turton, 2013; Welsch et al., 2014; Luderer et al., 2017; Pietzcker et al., 2017; Bistline, 2021). While
approximate methods such as parameterization via residual load duration curves (RLDCs) are able to capture the supply-side
dynamics of VREs, they remain methodologically limited for representing the flexible demand-side dynamics (Ueckerdt et al.,
2015; Creutzig et al., 2017). Besides limited temporal resolutions, IAMs also usually have coarse spatial resolutions, which can
lead to an under- or overestimation of transmission grid bottlenecks, geographical variability of wind and solar resources, and of
the flexibility requirements to balance supply and demand (Aryanpur et al., 2021; Frysztacki et al., 2021; Martínez-Gordón et
al., 2021). PSMs, on the other hand, usually lack the global and sectoral scope required for addressing global climate mitigation,
in part because of limited availability of detailed data, and due to computational challenges. Furthermore, PSMs with a short-
term horizon may lack the vintage tracking of standing capacities, capacity evolution over time, as well as long-term perfect
foresight, which can help policy makers and companies to look ahead beyond the short-term business cycles, to invest early and
to actively drive technical progress. In contrast, in IAMs such as REMIND, proactive early investment is a built-in feature,
because the optimization is done from a long-term social planner's perspective. In IAMs, investing early in the technological
learning phase results in lower costs of energy expenditure later, avoiding the severity of punishment to economic growth later
in time in the form of lower consumption, which raises the welfare which the model optimizes.
**1.2 Iterative coupling for full model convergence**
IAMs and PSMs differ in scope and resolution across three main modeling dimensions: temporal, spatial and technological. A
soft-coupling approach can tap into these complementarities and combine their strengths, at potentially only a moderate increase
in computational cost. The main challenge of the soft-coupling approach is to show that the two models can converge under
coupling, which leads to a joint equilibrium that maximizes regional interannual intertemporal welfare in the IAM and
minimizes total power system costs in the PSM. Ideally, the converged model offers the "best of both worlds": it has both the
broad scope required to assess global long-term energy transitions, as well as the technical resolution required to capture the
interplay between VREs, storage and newly electrified demand on a much shorter time scale.
Approaches aiming to bridge the "temporal resolution gap" between long-term energy system models and hourly PSMs have
been proposed in the past (Deane et al., 2012; Sullivan et al., 2013; Alimou et al., 2020; Brinkerink, 2020; Seljom et al., 2020).
While these achieved some aspects of Objective (1) with adequate results, none attempted to incorporate and achieve Objective
(2). In addition, there is a methodological gap in the previous attempts to a full harmonization of the multiscale models. By a
full harmonization, we mean a comprehensive coupling of the power sector dynamics, and an eventual model convergence in
capacities, generation, and prices. In none of the previous studies, price information has been fed back into the long-term models



from the short-term models for the complete set of generation technologies, only partial price information has been exchanged
in one of the studies (Seljom et al., 2020). Without a feedback mechanism through prices, the investment in the coupled model
will very likely be sub-optimal due to two effects: 1) because of the misalignment in prices in the two models, there is a
mismatch in investment incentives, resulting in a mismatch for optimal capacities if both models are completely endogenous; 2)
in all previous studies, the capacities are fixed in the PSM and only the long-term model is allowed to invest in new capacities.
This implementation can further propagate and sustain the price mismatch due to (1) via nontrivial shadow prices from these
capacity bounds, and create in turn price distortions in the PSM that can be passed on to the IAM. Therefore, the methodological
gap in previous work prevented a comprehensive convergence of the coupled models of both quantities and prices. As we show
later in this study, without a comprehensive coupling of price information, no system-wide convergence can be achieved.
However, with price coupling as our method proposes, we could achieve all aspects of Objective (1), as well as Objective (2) for
one type of flexible demand with adequate numerical results, and therefore represents a first step to bridge the previous
methodological gap.
Compared to previous studies, our approach features three main innovations: 1), the coupling is achieved by linking market
values, and not hard fixing quantities, allowing both models to invest "as endogenously as possible"; 2), the market values of all
power sector technologies are coupled, not just the electricity price of the system or the market value of a particular technology,
allowing models to achieve close to full convergence; 3) under idealized coupling assumptions and for a simplified "proof-of-
concept" model without storage, we can mathematically derive the necessary conditions under which comprehensive model
convergence can be reached, which puts multiscale coupling on firm theoretical footing. Our coupling approach is bi-
directional, iterative and fully automated.
To showcase such a framework and its ability to achieve iterative convergence, we couple the PSM DIETER, which has an
hourly resolution (8760 hours in a year) and the IAM REMIND for a single-region Germany. Germany is a well-suited case
study for exploring high VRE shares in the power sector. The country is expected to meet stringent climate targets despite the
country's high level of residential and industrial power demand, relatively small geographical size and lack of solar endowment
during winter seasons. Nevertheless, the German government has set very ambitious targets for the expansion and use of
variable renewable energy sources (DIW Berlin, 2022). A viable zero-carbon power mix in Germany must include an adequate
amount of storage and transmission for the renewable generation, as well as "clean firm generation" such as geothermal,
biomass or gas with carbon capture and storage (CCS) (Sepulveda et al., 2018).
**2 Models**
The models used in this study are well-documented open source models (REMIND is an open source model but requires
proprietary input data to run). A side-by-side comparison of the scope, resolution and other specifications of the two models can
be found in Appendix A. The coupling scope can be found in Appendix B. Details on model input data can be found in
Supplemental Material S-1.
**2.1 IAM: REMIND**
REMIND (REgional Model of INvestments and Development) is a process-based IAM, which describes complex global energy-
economy-climate interactions (Baumstark et al., 2021). REMIND has been frequently used in long-term planning of
decarbonization scenarios, most notably in the IPCC (IPCC, 2014; Rogelj et al., 2018; P.R. Shukla et al., 2022). The REMIND
model links different modules, which describe the global economy, the energy, land and climate systems, with a relatively
detailed representation of the energy sector compared to non-process-based IAMs. The model is formulated as an interannual



intertemporal optimization problem. Due to the computational complexity of nonlinear optimization, the model simulates a time
span from 2005 to 2100 with a temporal resolution of either 5 years (between 2005 to 2060) or 10 years (between 2070 to 2100).
The years in REMIND are representative years of the surrounding 5 or 10-year period, e.g. year "2030" represents the 5-year
period 2028 to 2032. Spatially, the model represents the world composed of aggregated global regions (Fig. B1). For each
region, using a nested constant elasticity of substitution (CES) production function, the model maximizes interannual
intertemporal welfare as a function of labor, capital, and energy use (Baumstark et al., 2021). The macro-economic projections
of REMIND come from various established global socio-economic scenarios jointly used by social scientists and economists –
the so-called Shared Socioeconomic Pathways (SSPs) (Bauer et al., 2017).
By default, REMIND runs in a regionally decentralized iterative "Nash mode", where all regions are run in parallel and the
interannual intertemporal welfare is maximized for each region for each internal "Nash" iteration. Trade flows between the
regions are determined between the Nash iterations. During the Nash algorithm, REMIND regions share partial information
between each other, which are trade variables in primary energy products and goods. The Nash algorithm is said to converge,
when all markets are cleared and no region has the incentive to change their behavior regarding their trade decisions, i.e. no
resources can be reallocated to make one region better off without making at least one region worse off. A successfully
converged run of stand-alone REMIND under "Nash mode" usually consists of 30 to 70 iterations of single-region models in
parallel. Each parallel single-region model usually takes 3-6 minutes to solve. A typical REMIND run in the Nash mode lasts
2.5-6 hours depending on the level of sectoral details included. The latest version REMIND (v3.0.0) is published as an open-
source version on github (Release REMIND v3.0.0 · remindmodel/remind, 2022). REMIND is implemented as a nonlinear
programming (NLP) mathematical optimization problem. In REMIND, the nonlinearity consists of the welfare function, the
CES production functions, adjustment costs, technological learning, the extraction cost functions, the bioenergy supply function
and nonlinear constraints, among others.
**2.2 PSM: DIETER**
DIETER (Dispatch and Investment Evaluation Tool with Endogenous Renewables) is an open-source power sector model
developed for Germany and Europe. In a long-run equilibrium setting (i.e. a competitive benchmark), the model minimizes
overall system costs of the power sector for one year. DIETER determines the least-cost investment and hourly dispatch of
various power generation, storage, and demand-side flexibility technologies. In previous literature, different versions of the
model have been used to explore scenarios with high VRE shares, where storage (Zerrahn et al., 2018; Zerrahn and Schill, 2017;
Schill and Zerrahn, 2018), hydrogen (Stöckl et al., 2021), power-to-heat (Schill and Zerrahn, 2020), or solar prosumage (Say et
al., 2020; Günther et al., 2021) are evaluated with a high degree of technological detail. DIETER recently also contributed to
model comparison exercises that focused on power sector flexibility for VRE integration and sector coupling (Gils et al., 2022b,
a; van Ouwerkerk et al., 2022).
As a first step to building a model coupling infrastructure, we implemented an earlier and simpler version of DIETER (v1.0.2),
which is purely based on the General Algebraic Modeling System (GAMS). It has limited features on ramping constraints,
flexible demand, and storage. The model minimizes total investment and dispatch cost of a power system for a single region,
considering all consecutive hours of one full year. The technology portfolio contains conventional generators such as coal and
gas power plants, nuclear power, as well as renewable sources such as hydroelectric power, solar PV and wind turbines.
Endogenous storage investment and dispatch, as well as demand flexibilizations are offered as additional features that can be
turned on or off. DIETER, like many PSMs, is a linear program (LP). A typical stand-alone run (with essential features) lasts



from several seconds to several minutes for a single region. See Zerrahn and Schill, 2017 for a detailed documentation of the
initial model, which was implemented purely in GAMS.

**3 A novel coupling approach**

It is central to our approach that the price-based variables, such as the market values of electricity generation, are exchanged
between the models. This approach ensures full convergence – including both quantity convergence as well as price
convergence in the market equilibrium. Here, we first introduce the intuition behind this approach, then conduct a deep dive into
the economic theory behind energy system modeling.
Economic concepts such as market values or capture prices (Böttger and Härtel, 2022), as key variables in our coupling,
translate the physical characteristics of variable power generation or flexible consumption into economic ones. For example,
generation technologies differ with respect to physical features and constraints – solar and wind generation depends on current
weather conditions as well as diurnal and seasonal patterns, whereas this is less the case for dispatchable power plants such as
coal, gas, biomass, nuclear or storage (López Prol and Schill, 2021). One consequence of this is that, for example, prices in
hours where PV does not produce will be essentially set by other, and usually more expensive forms of generation. In cost-
minimizing PSMs, the shadow prices of the energy balance are interpreted as wholesale market prices (Brown and Reichenberg,
2021; López Prol and Schill, 2021). Therefore in general, hourly-resolution PSMs are well equipped to translate such physical
constraints of generation into (wholesale) power market price time series. By providing such prices generated by PSMs (among
other variables of the power sector dynamics) to IAMs, the latter can be indirectly informed about power market dynamics
happening on much shorter time scales, even if they lack hourly resolutions. Over iterations, the prices from PSMs act as "price
signals" to induce investment decision changes in IAMs, which can in turn provide feedback to the PSMs until the two models
converge.
One innovation of our method is that the prices used for the model coupling can be symmetrically applied on the power supply
side as well as the demand side. On the supply side, the coupling method mainly utilizes the concept of market value (i.e. annual
average revenue per energy unit of a generator) in a competitive market at equilibrium. Generally speaking, market values of
generation usually convey the degree of variability intrinsic to a given source of power supply, and reflect the generator's ability
to meet an inflexible hourly demand, especially given lower cost of variable generation compared to dispatchable technologies.
Mirroring the concept of the market value, on the demand side, there is the concept of the "capture price" of electricity demand,
which conveys the degree of demand-side flexibility. Note that there may be multiple terminologies for demand-side electricity
prices, we use "capture price" to be consistent with one of the literatures on this topic. The capture price is the average
electricity price that a flexible demand technology pays over a year. For example, flexible demand technologies such as heat
pumps, electrolyzers or electric vehicles (EVs) can take advantage of electricity at hours when the generation is cheap to obtain
a lower "capture price", whereas inflexible demand has to pay a higher price on average. Price information given from a PSM to
an IAM from both the supply and demand sides can change the IAM's inherent investment and dispatch decisions of power
generation as well as inflexible and flexible demand-side technologies.
For an intuitive understanding of our innovative coupling scheme, we take the supply-side as an example, and use a toy model
to visualize the approach of coupling via market values. The market values of electricity generating technologies have been
studied in depth (Sensfuß, 2007; Sensfuß et al., 2008; Hirth, 2013; Mills and Wiser, 2015; Hildmann et al., 2015; Koutstaal and
va. Hout, 2017; Figueiredo and Silva, 2018; Hirth, 2018; Brown and Reichenberg, 2021). The general idea of the coupling is
illustrated in Fig. 1 for a simplified case of only two types of generators – dispatchable gas turbines and solar photovoltaics with
variable output. Note that we assume the system is at a solar share of > 50% and no storage, such that the solar market value is



below average electricity price, and that of gas generation is above. Before the coupling, for a general IAM with coarse temporal
resolution and without any VRE integration cost parameterizations, there is no differentiation between the market values of gas
and solar generators – they are both equal to the electricity price. Thus, there is no differentiated revenue for one MWh
generated by variable sources and dispatchable sources. The lack of market value differentiation is a direct consequence of the
limited temporal resolution in IAMs, which cannot represent hourly dynamics. However, through a market-value-based
coupling, the IAM can be informed by the PSM via a price "markup". The annual price markup is defined as the difference
between the market value of a specific technology and the annual average revenue that all generators together earn for one unit
of generation (i.e. the annual average electricity price that a user pays). Under our soft-coupling approach, the markups from the
PSM act as price-signals that change the composition of the energy mix in the next iteration of the IAM. Since in this simple
example with a lot of PV and no storage, the gas generator is "more valuable" to the system, as it can generate electricity in
times of scarcity (night), and thus it will receive a positive markup. When this positive price incentive is transferred from the
PSM to the IAM, it increases the optimal level of investment into gas generation in the next IAM iteration. At the same time,
solar generation receives a negative price incentive, reducing the optimal level of investments in the next iteration. Ultimately,
the higher market value of gas turbines is due to: 1) its higher cost compared to solar (when gas is at <50% market share), 2) its
ability to set prices in hours of low solar output and inflexible electricity demand. As we later show through mathematical
theory of model convergence, other information besides markups also needs to be transferred such as capacity factors (annual
average utilization rates of the generators).

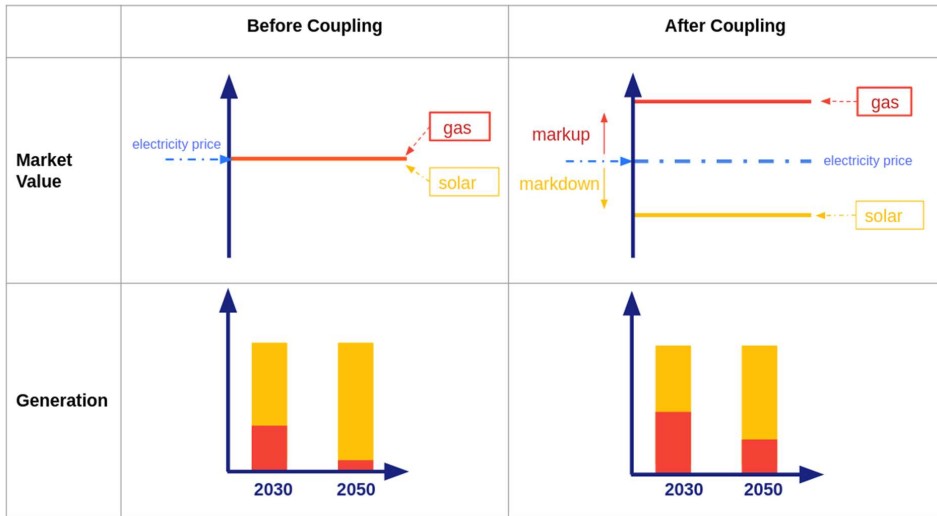


**Figure 1: Schematic illustration of the coupling approach for a simple power system in an IAM with coarse temporal**

**resolution, consisting of only gas and solar generators (no storage). Left column: before coupling; right column: after**

**coupling. Top row: endogenous prices (electricity price, market values of solar and gas generators); bottom row:**

**endogenous quantities (generation mix). The markups (as part of a larger set of interfaced variables) are the differences**

**between market values and electricity prices, and are given by the PSM of high temporal resolution as price signals to**

**the IAM. Usually, it is called a "markup" when the market value is higher than the annual average electricity price, and**

**"markdown" if it is the other way around. For simplicity, in the rest of the text we only refer to "markup" and**



**"markdown" collectively as "markup", regardless of whether the market value is higher or lower than the average**
**electricity price.**

There are several advantages to this new coupling approach centered on linking prices. First, instead of simply prescribing
quantities such as yearly generation and capacities, the approach allows endogenous investment decisions to be made by both
models as they converge towards a joint solution. This gives maximal freedom to the coupled models, while minimizing
unnecessary distortions from one model to the other when some necessary quantities are being prescribed. Second, our coupling
scheme provides an elegant treatment of both supply- and demand-side technologies using the concept of "market values" on the
one hand and "capture prices" on the other. Third, from a theoretical point of view, transferring the market values of all the
generation types in a system alongside mappings of other relevant system parameters can lead to a convergence of the solutions
of the two models under idealized coupling circumstances. It can be rigorously shown that our method contains an exhaustive
list of interfacing parameters and variables for full model convergence of both quantities and prices. To the authors' best
knowledge, the last point has not been explored or shown in any previous work.
In certain IAMs, VRE integration cost parameterization has been implemented to mimic the economic consequences of
variability of VRE, especially when the models have lower temporal resolution. Such VRE integration costs are contained in the
uncoupled default REMIND power sector modeling. However, the exact parameterization always depends on a particular set of
technological costs and parameters which might be subject to changes (Pietzcker et al., 2017), and the parametrization often
needs to be carried out anew under new assumptions and scenarios. In contrast, the model coupling approach is more general,
and no such bespoke parametrization is needed.
Inspired by the theoretical framework based on the Karush–Kuhn–Tucker (KKT) conditions for power sector optimization
problems (Brown and Reichenberg, 2021), we develop the theoretical basis for the coupling method in this section, which we
use for validating convergence in numerical coupling in later sections. In Section 3.1, we analytically formulate the fundamental
economic theory of the coupling approach. We first introduce the power sector formulations in the two uncoupled models (Sect.
3.1). Then we carry out a derivation of the convergence conditions and criteria, where we map the Lagrangians of the two
power-sector problems at different time resolutions, and derive the equilibrium condition for the coupled models (Sect. 3.2). In
Sect. 3.3, we introduce the iterative coupling interface which contains all the previously derived convergence conditions. For
REMIND information being passed on to DIETER (Sect. 3.3.1), and DIETER information being passed on to REMIND (Sect.
3.3.2), we list and define the variables and parameters being exchanged at the interface, as well as additional constraints and
implementations which serve to improve the coupling.
A complete list of mathematical symbols and list of abbreviations can be found in the appendices.
In the following sections, we first formulate the two uncoupled models, then move onto discussing coupled models. The
theoretical tools we develop here are the foundation to the numerical implementation of coupling, and serve to validate and
assess the model convergence in the result sections.
**3.1 Descriptions of uncoupled models**
REMIND and DIETER are both optimization models. REMIND maximizes interannual global welfare from 2005 to 2150,
whereas DIETER minimizes the power sector system cost for a single year and a single region. For a given REMIND "Nash"
iteration (see Sect. 2.1), the single-region economy is in long-term equilibrium after the optimization problem is solved. Since
given fixed national income, lower energy system costs mean higher consumption which leads to increased welfare (see
Appendix C for details), maximizing welfare can be assumed to correspond to minimizing energy system costs, a part of which





is power sector costs. Therefore, to reduce the complexity of our analysis, we formulate an uncoupled REMIND model based
solely on the power sector cost minimization and not the total welfare maximization. For stand-alone REMIND, the multi-year
power system cost for a single region equals the sum of all variable and fixed costs of generation,
$$Z = \sum_{y,s} (c_{y,s} P_{y,s} + o_{y,s} G_{y,s}), \tag{1}$$
where $c$ represents the fixed cost for capacity, $o$ represents the variable cost of running power generation, $P$ denotes endogenous
capacity, and $G$ denotes endogenous generation (defined as including curtailment in REMIND). $P$ and $G$ are the decision
variables of the problem. The sum in the objective function is over time index $y$ and power generating technology type $s$. The
REMIND time index y stands for one representative year, which represents 5 or 10 years centered around it. So even though the
time step is 5 to 10 years, the time resolution is one year. For example, "y=2020" represents the years 2018-2022.Capital letters
(both Latin and Greek) denote independent decision variables of the optimization problem. We classify an endogenous decision
variable as independent if it is not uniquely determined by one or more other decision variables, and has no binding constraints
applied to itself that is not already accounted for by the constraints on the decision variable(s) it depends on. Note that for
simplicity, we treat all costs in REMIND in this formulation as if they are exogenous. In reality, REMIND has endogenous fixed
costs due to technological learning as well as endogenous interest rate. Some types of variable costs such as fuel costs are also
endogenous, which are determined based on primary energy balance equations for oil, gas and biomass. $CO_2$ prices can also be
endogenous under emission constraints.
Under the simplifying assumptions made for the derivation in this paper, the only independent decision variables are capacities,
generations and curtailments. Small letters denote either exogenously given parameters or endogenous shadow prices.
For stand-alone DIETER which has a year-long time horizon, the power system cost is:
$$\underline{Z} = \sum_s \underline{c}_s \underline{P}_s + \sum_h \left[ \underline{o}_s \left( \underline{G}_{h,s} + \underline{\Gamma}_{h,vre} \right) \right], \tag{2}$$
where $\underline{G}_{h,s}$ is the endogenous hourly power generation (excluding curtailment, note that this is different from the generation
variable definition in REMIND), $h$ is the hourly index in a year from 1 to 8760, $s$ is the index for the power generating
technology in DIETER. $\underline{\Gamma}$ is hourly curtailment, only applicable in the case of variable renewables $vre$ ($vre \subset s$). Technology
type $s$ can be subdivided into two subsets: $vre$ and $dis$ ("dispatchables"). For simplicity, we abbreviate the index subscript from
$s|s = vre$ to $vre$ and $s|s = dis$ to $dis$. Here in order to differentiate from REMIND notations, we use underscore $\_$ to denote
DIETER parameters and variables. Note that for simplicity, in the derivation we treat the technology types in both models as
being identical, although in fact the technologies in the two models are not one-to-one mapped (Fig. B2). During the coupling
all interface parameters and optimal decision variables need to be upscaled or downscaled when transferred from one model to
the other.
The cost minimization of total power sector cost Z and $\underline{Z}$ under constraints yields the optimal values of the decision variables,
denoted as $(P_{y,s}^*, G_{y,s}^*)$, and $(\underline{P}_s^*, \underline{G}_{h,s}^*, \underline{\Gamma}_{h,s}^*)$.
Without coupling and under a baseline scenario, there are several constraints for each model. In the following equations we
denote the shadow price (i.e. the Lagrangian multiplier) of a constraint by the symbol following ⊥. We use small greek letters to
denote endogenous shadow prices, and small Latin and Greek letters to denote exogenous parameters. The major constraints are
as follows ("c" stands for "constraint"):
c1) Constraint on generation for meeting demand, a.k.a. "supply-demand balance equation", or "balance equation" in short:

REMIND (annual):     $d_y = \sum_s G_{y,s}(1 - \alpha_{y,s})$    $\perp \lambda_y$ ,

DIETER (hourly):     $\underline{d}_h = \sum_s \underline{G}_{h,s}$        $\perp \underline{\lambda}_h$ ,



where $d_y$ denotes annual REMIND power demand, and $\underline{d}_h$ denotes DIETER hourly demand. The shadow prices (Lagrange
multipliers) $\lambda_y$ and $\underline{\lambda}_h$ represent the annual and hourly electricity prices in REMIND and DIETER, respectively, and are
equal to the marginal cost of one additional unit of electricity generation. $\alpha_{y,s}$ is the annual VRE curtailment ratio in
REMIND. Note that technically speaking, REMIND electricity demand $d_y$ is determined endogenously, partially via
competition with other energy carriers at the final energy consumption level, such as the competition between electricity and
gaseous carriers such as natural gas or hydrogen in household heating. But because here we have reduced REMIND to only
intra-power sector dynamics for the purpose of mathematical analysis, we treat demand as exogenous.
c2) Constraint on maximum capacity by the available annual potential $\psi_s$ in a region:
REMIND:      $P_{y,s} \leq \psi_s \quad \perp \omega_{y,s}$ ,
DIETER:      $\underline{P}_s \leq \underline{\psi}_s \quad \perp \underline{\omega}_s$ .
Note that the resource constraint in REMIND is only relevant for wind, solar and hydro, and is assumed to be constant over
the model horizon. Biomass availability is not modeled via a regional potential constraint. Instead the availability of biomass
is priced in through the soft-coupling to the land-use model MAgPIE via a supply curve.
c3) Constraint on generation being non-negative:
REMIND:      $-G_{y,s} \leq 0 \quad \perp \xi_{y,s}$ ,
DIETER:      $-\underline{G}_{h,s} \leq 0 \quad \perp \underline{\xi}_{h,s}$ .
Note that there are several other similar constraints on other positive variables such as capacities and curtailment. In practice,
during the derivation they behave similarly to this positive generation constraint, therefore for simplicity, we do not include
them in the derivation.
c4) Constraint on maximum generation from capacity:
REMIND:              $G_{y,s} = \phi_{y,s} P_{y,s} * 8760 \qquad \perp \mu_{y,s}$ ,
DIETER: (variable renewables) $\underline{G}_{h,vre} + \underline{\Gamma}_{h,vre} = \underline{\phi}_{h,vre}\underline{P}_{vre} \qquad \perp \underline{\mu}_{h,vre}$

(dispatchables)  $\underline{G}_{h,dis} \leq \underline{P}_{dis} \qquad \perp \underline{\mu}_{h,dis}$ ,

where $\phi_{y,s}$ is the exogenous annual average capacity factor of the power plant $s$ in REMIND in year $y$, and $\underline{\phi}_{h,vre}$ is the
exogenously given hourly theoretical capacity factor (i.e. before curtailment) of VRE in DIETER. Note that strictly
speaking, curtailments in the uncoupled REMIND and DIETER are endogenous decision variables but are not independent
variables. However, here we use capital letter to denote hourly curtailment in DIETER as an independent decision variable to
account for curtailment costs and other curtailment constraints that can arise from a more general formulation of the model.
c5) "Historical" constraints on capacities in REMIND. This makes REMIND a so-called "brown-field model", i.e. a model
accounting for the standing capacities in the real-world. Past capacities ($y < 2020$) are hard-fixed, i.e. the variable capacities
are fixed to certain numeric values. Current capacities ($y = 2020$) are "soft-fixed", i.e. the variable capacities are fixed to a
corridor around certain standing numeric values: the lower bounds guarantee the already planned capacities, and the upper
bounds reflect the finite physical capabilities of scaling up, defined by 5% above the 2020 real-world data. For simplicity,
we use only one constraint for both past and current capacities,
$P_{y,s} \geq p_{y,s} \qquad \perp \sigma_{y,s}$  for $y \leq 2020$ ,
where $p_{y,s}$ represents the standing capacities of technology $s$ at time $y$ in REMIND in the past and present years.
c6) Near-term upscaling constraint on VRE capacity expansion, represented by an upper bound on near-term capacity addition
in model period $(y - \Delta y, y)$, $\Delta P_{y,s} := P_{y,s} - P_{y-\Delta y,s}$, where $\Delta y$ is the REMIND model time step:





$\Delta P_{y,s} \leq q_{y,s} \qquad \perp \gamma_{y,s}$  for  $y = 2025$ ,
where $q_{y,s}$ is equal to twice the added capacity during the 2010-2020 period (only applied to Germany in default REMIND).
Note that constraints (c5) and (c6) introduce interannual intertemporality into the power sector of REMIND. This additional
interannual intertemporality determines that the model equilibrium can only be strictly satisfied across the sum of all model
periods and not for a single period. Another source of intertemporality in REMIND is due to the adjustment cost, which we
ignore in the main text of this study since it introduces non-linearity in the power sector and also plays a relatively small role in
the overall dynamics.
Note that regarding the simplification of REMIND above, to the authors' best knowledge, there is no theoretical or empirical
concept that addresses the validity of drawing equivalence between welfare maximization and energy system cost minimization
in IAMs. Naively, given GDP is unchanged, decreasing energy system cost raises consumption and therefore welfare. However,
this is only valid under the assumption that energy is a substitute (and not a complement) to capital and labor, i.e. one usually
cannot raise economic output (GDP) simply by spending more on higher energy expenditure (while satisfying the same level of
energy demand). Nevertheless, this is likely a necessary condition and not a sufficient one for proving the equivalence. More
theoretical research will be needed to draw a precise and rigorous equivalence. However, in practice, we see that during our
numerical calculation the model is well behaved according to this reduced theory, which means that the parameters in the
models are in a regime where such an assumption is valid, at least in the case of IAM REMIND.

### 3.2 Economic theory of model convergence

In the last section we have discussed the stand-alone uncoupled power sector formulations in REMIND and DIETER. In this
section we discuss the coupled models and its convergence. Under simplified assumptions, we first derive the mapping between
the models which are necessary for a convergence (Sect. 3.2.1-2), then we derive theoretical relations which are later used to
validate the numerical results of the coupled run (Sect. 3.2.3).

### 3.2.1 Derivation of convergence conditions

Our aim is to develop a method under which comprehensive convergence can be reached for soft-coupled multiscale models.
We achieve this by deriving a mapping of the two problems, such that their decision variables have identical optimal solutions
and the endogenous shadow prices are also equal across the models. The convergence conditions of the coupled REMIND-
DIETER model for the power sector are the result of such a mapping. Below, we first define what is meant by a "comprehensive
model convergence", and then sketch the workflow of the derivation of a coupling framework which would result in a
comprehensive model convergence of both decision variables and shadow prices. The detailed derivation is in Appendix D.
Here, we derive the conditions under which the endogenous decision variables are identical at each model's optimum, i.e. $P_{y,s}^* =$
$\underline{P}_{y,s}^*$ , and $G_{y,s}^*\left(1 - \alpha_{y,s}\right) = \sum_h \underline{G}_{y,h,s}^*$ (or equivalently pre-curtailment generation $G_{y,s}^*$ and $\sum_h \left(\underline{G}_{y,h,s}^* + \underline{\Gamma}_{y,h,s}^*\right)$. A convergence of
the solutions of these two sets of annual decision variables for each technology $s$ and for each year $y$, along with the
convergence of shadow prices gives rise to "comprehensive model convergence". We show below that this can only be achieved
if there is a harmonization at the level of the KKT Lagrangians of the two problems, following the methods first developed by
Karush, Kuhn and Tucker (Karush, 1939; Kuhn and Tucker, 1951).
Our coupling approach fundamentally relies on mapping the parameterization of the Lagrangians for both optimization
problems. It is trivial to show that as long as the KKT Lagrangians are identical with respect to the decision variables, the
solutions of the problem are identical. For example, if an optimization problem A has Lagrangian L_1 = a_1*x+b_1*y and





another problem B has Lagrangian L_2 = a_2*x+b_2*y, where x and y are decision variables of the optimization problems.
Then if we let a_1 = a_2, b_1 = b_2, the two problems are identical, and they must have identical optimal solutions for the
decision variables x* and y*. This is the basic logic behind the Lagrangian-based method. The challenge in the case of
REMIND and DIETER is to show that when a decision variable representing the same physical quantity, for example, the
annual power generation from a technology is defined with low resolution in one problem, and is defined with high resolution in
another, that there is nevertheless a viable mapping between the two Lagrangians. In this case, the parameterization of the
Lagrangian is not only limited to exogenous parameters of the model, but also includes endogenous shadow prices and
endogenous decision variables from the other model. Due to the endogenous nature of the latter two, the parametrization in the
current-iteration model A must come from the solved results from the last iteration from model B, and vice versa. Fig. 2
illustrates the workflow of the analytical derivation of the convergence conditions.

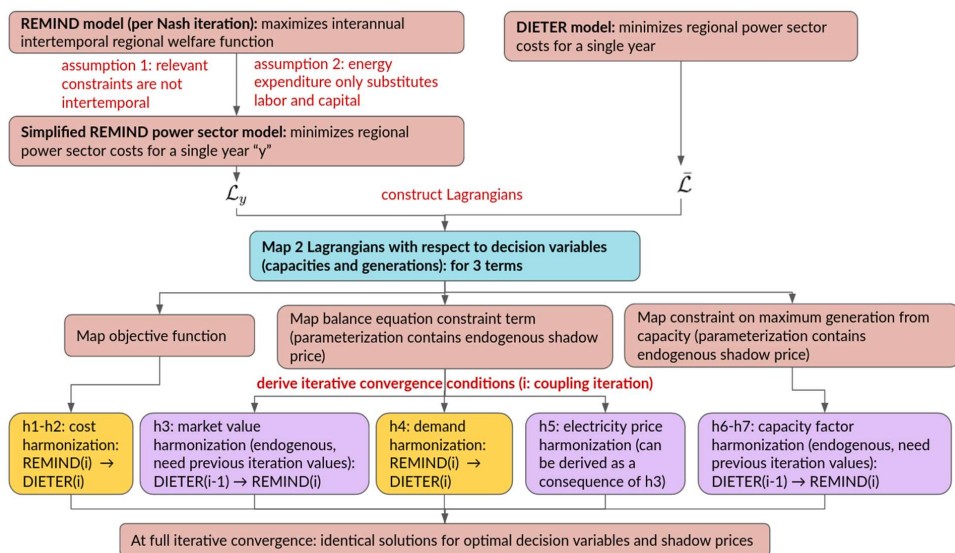


**Figure 2: The schematics of the Lagrangian-based derivation procedure for a simplified version of REMIND-DIETER**
**iterative convergence. After simplifying assumptions, we can construct the Lagrangians of the reduced REMIND model**
**and the full DIETER model for a single year (Eqs. (3)-(4)). Comparing and mapping terms in the Lagrangians (a key**
**step in bold), we discover that iterative exchange of a broad range of information is needed for a fully harmonized**
**parameterization of the Lagrangians. Under the harmonization specified in the seven convergence conditions (color**
**coded for directions of information flow), the coupled models can give rise to identical optimal solutions of the models'**
**respective (annual aggregated) decision variables, and hence a full quantity convergence. The necessary shadow price**
**convergence is shown in the detailed derivation of the harmonization conditions (h1-h7) in Appendix D.**

The analytical derivation workflow, as shown in Fig. 2, is described in detail as follows. First, we apply simplifying assumptions
to reduce the complexity of the uncoupled models (before the key step in blue in Fig. 2). Assumptions have to be made to justify
reducing the scope of the REMIND model, such that for the purpose of the analysis, it is on equal footing as DIETER. We
achieve this by reducing the global REMIND model to single-sector (the power sector), single-year, and single-region. To
reduce the REMIND model from a macroeconomic-energy model to a power-sector-only model, we make similar assumptions
as before when formulating the uncoupled REMIND power sector (see Sect. 3.1). To reduce the REMIND model further to a



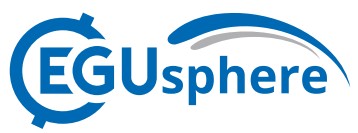

single year, we assume that the models only contain constraints in the power sector that are not intertemporal, i.e. ignoring the
brown-field and near-term constraints for now. Since for each iteration of the REMIND model under "Nash mode", inter-
regional trading happens between the iterations, the single-iteration optimization model is already for a single region, and
therefore does not require simplification. After these simplifying steps, in this part of the derivation, we can treat REMIND's
power sector as "separate" from the rest of the model, and treat the dynamics of a single year in REMIND as independent from
the dynamics of other years. Later, the numerical results of the convergence can confirm to a large degree the validity of these
assumptions, especially in the green-field temporal ranges, i.e. where the intertemporal brown-field constraints have little
influence on the dynamics. Note that with the inclusion of these intertemporal constraints in the derivation, the mapping
becomes more complicated, especially for the near-term range, i.e. before 2035. So in practice, this derivation of the coupling
interface is only an approximation to what is needed for a full convergence of DIETER and REMIND, since it deliberately
ignores such constraints. See also Sec. 6.1.
After the necessary simplification assumptions, we construct the Lagrangians for the simplified model REMIND and for
DIETER (after the blue block in Fig. 2) (Gan et al., 2013). For a single-year reduced REMIND power sector model, the
Lagrangian is:
$$\mathcal{L}_y = \underbrace{\sum_s \left(c_{y,s} P_{y,s} + o_{y,s} G_{y,s}\right)}_{\text{REMIND objective function}} + \underbrace{\lambda_y \left[d_y - \sum_s G_{y,s}(1 - \alpha_{y,s})\right]}_{\text{annual electricity balance equation constraint}} + \underbrace{\sum_s \mu_{y,s}\left(G_{y,s} - 8760 * \phi_{y,s} P_{y,s}\right)}_{\text{maximum generation from capacity constraint}} .$$
(3)

We would like to map it to the single-year DIETER Lagrangian $\underline{\mathcal{L}}$:
$$\underline{\mathcal{L}} = \underbrace{\sum_s \left[\underline{c}_s \underline{P}_s + \underline{o}_s \sum_h \left(\underline{G}_{h,s} + \underline{\Gamma}_{h,vre}\right)\right]}_{\text{DIETER objective function}} + \underbrace{\sum_h \underline{\lambda}_h \left(\underline{d}_h - \sum_s \underline{G}_{h,s}\right)}_{\text{hourly electricity balance equation constraint}} + \underbrace{\sum_{h,dis} \underline{\mu}_{h,dis}\left(\underline{G}_{h,dis} - \underline{P}_{dis}\right)}_{\text{maximum dispatchable generation from capacity constraint}}$$

$$+ \underbrace{\sum_{h,vre} \underline{\mu}_{h,vre} \left(\underline{G}_{h,vre} + \underline{\Gamma}_{h,vre} - \underline{\phi}_{h,vre} \underline{P}_{vre}\right)}_{\text{maximum renewable generation from capacity and weather constraint}} .$$
(4)

The algebraic derivation of mapping the two Lagrangians term-by-term is presented in Appendix D. From this algebraic
mapping, we can derive seven harmonization conditions (h1-h7) required for a full convergence. Conditions (h1-h7) are the
subsequent basis for most of the information exchanged at the coupling interface. Among them, conditions (h3, h5-7) (purple
blocks in Fig. 2) indicate conditions which contain endogenous information that must come from the previous iteration of
DIETER that is passed on to REMIND, such as markup and capacity factors. Conditions (h1-2, h4) (yellow blocks) indicate
conditions which contain information that come from the previous iteration of REMIND and are passed on to DIETER. For
schematics of the coupled iterations, see Appendix E.
This Lagrangian-mapping-based derivation can theoretically show that our approach (in its most simple form) necessarily leads
to model convergence, and has the advantage of being mathematically straight-forward and rigorous. The necessary information
from the power sector dynamics is all contained in the list of conditions derived from such a mapping. If the coupling contains
less information, a convergence is not possible; at the same time, for a model convergence, one does not need to pass on any
additional information beyond what is contained in this list of conditions. The list of information derived here is therefore
complete and exhaustive for a coupled convergence.
**3.2.2 List of convergence conditions**
The convergence conditions (h1-h7), which are derived in detail in Appendix D following the procedure in Sect. 3.2.1, are
summarized here:
**h1)** annual fixed costs are harmonized: $c_{y,s} = \underline{c}_{y,s}$ ,



**h2)** annual variable costs are harmonized: $o_{y,s} = \underline{o}_{y,s}$ .
**h3)** annual average market values for each generation type $s$ are harmonized via markups from DIETER. We let $\underline{\eta}_{y,s}(i-1)$
denote the markup for technology $s$ in year $y$ in the last iteration DIETER, i.e. the difference between market value and
annual average price of electricity:
$$\underline{\eta}_{y,s} = \underbrace{\frac{\sum_s \underline{\lambda}_{y,h}\underline{G}_{y,h,s}}{\sum_h \underline{G}_{y,h,s}}}_{\text{Market value}_s} - \underbrace{\frac{\sum_h \underline{\lambda}_{y,h}\underline{d}_{y,h}}{\sum_h \underline{d}_{y,h}}}_{\text{Annual average electricity price}_s} . \qquad (5)$$

This is the heart of our coupling approach, using markups as the "price signals". Intuitively, the markups represent the
market value differences between REMIND and DIETER. The harmonization of market values is implemented by
iteratively adjusting the market value for each generator type in REMIND to be the same as that in DIETER. As long as
the market values (or per-unit-generation revenues) and costs are harmonized, the economic structures of the power
market are identical and the models can converge.
Using markup Eq. (5), we modify the original objective function $Z$ in the coupled version of REMIND by subtracting the
product of markups and generations summed over all technologies and all years:
$$Z' = Z - \sum_{y,s} \underline{\eta}_{y,s}(i-1)G_{y,s}\big(1 - \alpha_{y,s}\big), \qquad (6)$$

where $Z'$ is the modified REMIND objective function in the coupled version, $i$ is the iteration index of the iterative soft-
coupling.
**h4)** annual power demands are harmonized: $\sum_h \underline{d}_{y,h} = d_y$ ,
**h5)** annual average prices of electricity are harmonized:
$$\lambda_y = \frac{\sum_h \underline{\lambda}_{y,h}(i-1)\underline{d}_{y,h}(i-1)}{\sum_h \underline{d}_{y,h}(i-1)}, \qquad (7)$$

where $(i-1)$ indicates that the endogenous results are from the last iteration. This is shown in Appendix D to be a direct
consequence of (h3) and (h4).
**h6)** annual average capacity factor for each generation type $s$ are harmonized:
$$\phi_{y,s} = \sum_h \underline{\phi}_{y,h,s}(i-1) / 8760, \qquad (8)$$

where $\underline{\phi}_{y,h,s}(i-1) = \frac{\underline{G}_{y,h,s}(i-1)}{\underline{P}_{y,s}(i-1)}$ is the hourly capacity factor in DIETER, determined by endogenous hourly generation
and annual capacities in the last iteration.
**h7)** annual curtailment are harmonized:
$$G_{y,vre}\alpha_{y,vre} = \sum_h \underline{\Gamma}_{y,h,vre}(i-1). \qquad (9)$$

In mapping the Lagrangians (Eqs. (3-4)), except the objective function, the rest of the parametrization contains endogenous
shadow prices and endogenous quantities. Since endogenous values can only be known ex post, this imposes a strict requirement
on the coupling that it must be iterative, with the endogenous part of the parameterization coming from previous iteration
optimization results – usually from the other model. The mapping of the endogenous information requires careful argument in
each case (i.e. the derivation of (h3)-(h7)). In the case of the balance equation constraint Lagrangian term (corresponding to
(c1)), the shadow prices of the constraint in current-iteration REMIND model are exogenously corrected by a set of technology-
specific "markups" (see Sect. 3.1 introduction), such that the new "corrected" market value in REMIND is manipulated to
match the market value of the previous iteration of DIETER. This is the heart of our coupling approach, using markups as the
"price signals". In the case of the constraint on maximum generation from capacity (corresponding to (c4)), the endogenous



shadow prices in the current iteration REMIND can be shown to be automatically mapped to the those in the previous iteration
of DIETER, given that the annual average capacity factors in the constraints are harmonized (h6-h7).
In actual implementation, most of the above mappings are modified for numerical stability (Sect. 3.3.2, Appendix H).

**3.2.3 Theoretical tools for validating convergence**

Here we first state the convergence criteria, which are mathematical relations which are being satisfied under model
convergence. Then we also discuss equilibrium conditions of the coupled models which alongside the convergence criteria can
be used to check numeric results to validate and assess the convergence outcome.
Under a theoretical full convergence of the coupled model,

v1) annual average electricity prices,

v2) capacities,

v3) (post- or pre-curtailment) generations,

all should be identical at the end of the coupling in both models. These are the most important criteria by which we validate full
model convergence. Technically, electricity price convergence (v1) (i.e. convergence condition (h5)) can be derived from (h3)-
(h4). Nevertheless, we check this ex post, together with quantity convergence (v2-v3). In actual coupled model runs, following
only the convergence conditions (h1-h7), the convergence criteria (v1-v3) might not be exactly fulfilled. Therefore in practice,
in order to validate the degree of numerical convergence, the alignment between REMIND and DIETER generation shares is set
to be within a few percentage points before coupled runs terminate.
Besides using convergence criteria (v1-v3), we also use a type of equilibrium condition – the so-called "zero-profit rules"
(ZPRs) to validate the numerical model convergence. ZPRs are mathematical relations which state that under market
equilibrium, prices are equal to the costs for electricity. This is not always the case, especially in the situation where there are
extra constraints in the model which distort this equality. ZPRs contain model parameters and decision variables at market
equilibrium, and they can be derived from the KKT conditions of the model (Appendix F). ZPRs are therefore reliable tools in
ascertaining the sources of market values or the price of electricity of the power sector, because according to the ZPRs, one can
always decompose the prices into the cost components, i.e. so-called levelized costs of electricity (LCOE). The decomposition
of prices into cost components is important, because the prices of electricity in the power market are overdetermined by the
energy mix, so it is possible that two different power mixes correspond to the same electricity price. In numerical results, a
slight mismatch of energy mix at the end of the coupling is unavoidable, so alongside comparing the prices, it is often helpful to
compare the makeup of the LCOE across the models, such that they also appear harmonized at the end of the iterative
convergence. Overall, ZPRs is a helpful tool for visualizing and understanding the power market dynamics, both from the point
of view of each generator type as well as from the point of view of the entire electricity system. It is worth noting, that the zero-
profit rules, which are mathematical conditions derived from an idealized modeling of the power sector as fully competitive, are
only an approximation to the real-world markets, where firm profits exist. ZPRs in its technical definition simply means that at
model equilibrium, cost equals revenue. Given that the profits are defined as the difference between revenue and cost, the profits
are zero in this situation. The name "zero-profit rule" therefore should not be overinterpreted beyond their technical contents,
and one should be aware of their theoretical origin and assumptions under which they are valid.
The ZPRs of the coupled model can be derived based on: 1), the uncoupled models; 2), the modification made to the model due
to the coupling interface (h1-h7); 3), any additional modifications made to the model during our numerical implementation. In
the last category, for a complete numerical implementation of the coupling, we add one additional capacity constraint (c7) and
(c8) for each model. The first capacity constraint (c7) is created in REMIND to circumvent the issue of extremely high markup



from peaker gas plants in the scarcity hour of the year in the DIETER model, which otherwise causes instability during the
iterative coupling. The second constraint (c8) is a simple brown-field constraint implemented in DIETER to address the fact that
DIETER is a green-field model, which is otherwise ignorant about standing-capacities in the real world. For simplicity, (c7) and
(c8) are not included in the convergence condition derivations in Sect. 3.2.1. The derivation of the ZPRs outlined by the above
three steps have been carried out in: Appendix F (uncoupled models), Appendix G (coupled REMIND only including coupling
interface, coupled DIETER including constraint (c8)), and Appendix H (coupled REMIND, including constraint (c7)).
In summary, the ZPRs for both coupled models are as follows:
a)  Coupled REMIND:

i)  Technology-specific ZPR:

$$\underbrace{\frac{\sum_y(c_{y,s}P_{y,s}+o_{y,s}G_{y,s})}{\sum_y G_{y,s}}}_{\text{Pre-curtailment LCOE}_s}+\underbrace{\frac{\sum_y(c_{y,s}P_{y,s}+o_{y,s}G_{y,s})\alpha_{y,s}}{\sum_y G_{y,s}(1-\alpha_{y,s})}}_{\text{Curtailment LCOE}_s}$$

$$=-\underbrace{\frac{\sum_y(\omega_{y,s}-\sigma_{y,s}+\gamma_{y,s}+\nu_{y,s})P_{y,s}}{\sum_y G_{y,s}(1-\alpha_{y,s})}}_{\text{Capacity shadow price}'_s}+\underbrace{\frac{\sum_y\left(\lambda_y+\underline{\eta}'_{y,s}\right)G_{y,s}(1-\alpha_{y,s})}{\sum_y G_{y,s}(1-\alpha_{y,s})}}_{\text{Market value}'_s} \qquad (10)$$

ii)  System ZPR:

$$\underbrace{\frac{\sum_{y,s}(c_{y,s}P_{y,s}+o_{y,s}G_{y,s})}{\sum_{y,s} G_{y,s}}}_{\text{Pre-curtailment LCOE}_{\text{system}}}+\underbrace{\frac{\sum_{y,s}(c_{y,s}P_{y,s}+o_{y,s}G_{y,s})\alpha_{y,s}}{\sum_{y,s} G_{y,s}(1-\alpha_{y,s})}}_{\text{Curtailment cost}_{\text{system}}}$$

$$=-\underbrace{\frac{\sum_{y,s}(\omega_{y,s}-\sigma_{y,s}+\gamma_{y,s}+\nu_{y,s})P_{y,s}}{\sum_{y,s} G_{y,s}(1-\alpha_{y,s})}}_{\text{Capacity shadow price}'_{\text{system}}}+\underbrace{\frac{\sum_{y,s}\left(\lambda_y+\underline{\eta}'_{y,s}\right)G_{y,s}(1-\alpha_{y,s})}{\sum_{y,s} G_{y,s}(1-\alpha_{y,s})}}_{\text{Electricity price}'_{\text{system}}} \qquad (11)$$

b)  Coupled DIETER:

i)  Technology-specific ZPR:

$$\underbrace{\frac{\underline{c}_s\underline{P}_s+\underline{o}_s\sum_h\left(\underline{G}_{h,s}+\underline{\Gamma}_{h,vre}\right)}{\sum_h \underline{G}_{h,s}}}_{\text{LCOE}_s}=-\underbrace{\frac{\left(\omega_s+\underline{\varsigma}_s\right)\underline{P}_s}{\sum_h \underline{G}_{h,s}}}_{\text{Capacity shadow price}'_s}+\underbrace{\frac{\sum_h \underline{\lambda}_h\underline{G}_{h,s}}{\sum_h \underline{G}_{h,s}}}_{\text{Market value}_s}. \qquad (12)$$

ii)  System ZPR:

$$\underbrace{\frac{\sum_s\left[\underline{c}_s\underline{P}_s+\underline{o}_s\sum_h\left(\underline{G}_{h,s}+\underline{\Gamma}_{h,vre}\right)\right]}{\sum_{h,s} \underline{G}_{h,s}}}_{\text{LCOE}_{\text{system}}}=-\underbrace{\frac{\sum_s\left(\omega_s+\underline{\varsigma}_s\right)\underline{P}_s}{\sum_{h,s} \underline{G}_{h,s}}}_{\text{Capacity shadow price}'_{\text{system}}}+\underbrace{\frac{\sum_h \underline{\lambda}_h\underline{d}_h}{\sum_h \underline{d}_h}}_{\text{Annual average electricity price}_{\text{system}}}. \qquad (13)$$

"Prime" sign indicates the term has been modified from the uncoupled versions due to implementation in the coupling. $\nu$ and $\varsigma$
are capacity shadow prices introduced from the additional constraints (c7-c8) (Appendix G-H). It is worth noting that constraints
(c7-c8) introduced due to coupling can impact the Lagrangians of the two models which we used to derive convergence
conditions and criteria. However, in actual coupled runs, evidently there is only a moderate distortion due to these extra
constraints. Condition (c8) even helps with convergence, because it puts most of the brown-field and near-term constraints
which REMIND sees also into DIETER (see Sect. 6.1).
Due to the fact that several sources of shadow prices cannot be incorporated during the derivation for convergence (Sect. 3.2.1),
in numerical experiments of coupled run it is appropriate to compare the following two types of prices across the two models for
price convergence:



1)   Electricity price convergence, not including any capacity shadow prices;
2)   Sum of electricity prices and all respective capacity shadow prices converge.
Under the simplified analysis of convergence (discounting brown-field constraints, scarcity prices, etc), price convergence in 1)
is predicted by theory (see also convergence condition (h5)). However, it is only under the most idealized situation.
Convergence in 2) on the other hand includes all the prices, which should match if LCOEs match across the system. We use the
first type to check price convergence over iteration, and use the second type only in the context of checking the system ZPRs
across the models because of the theoretical relations between full prices and LCOEs.
**3.3 Implementation via interface: exchange of variables**
In this section we list parameters and endogenous variables that are exchanged between REMIND and DIETER. This already
satisfies most convergence conditions, while the remaining condition (h5) is checked in Sect. 4 as part of the convergence
criteria (v1-v3). An overview of the model coupling and the flow of information under convergence conditions is shown in Fig.

3.

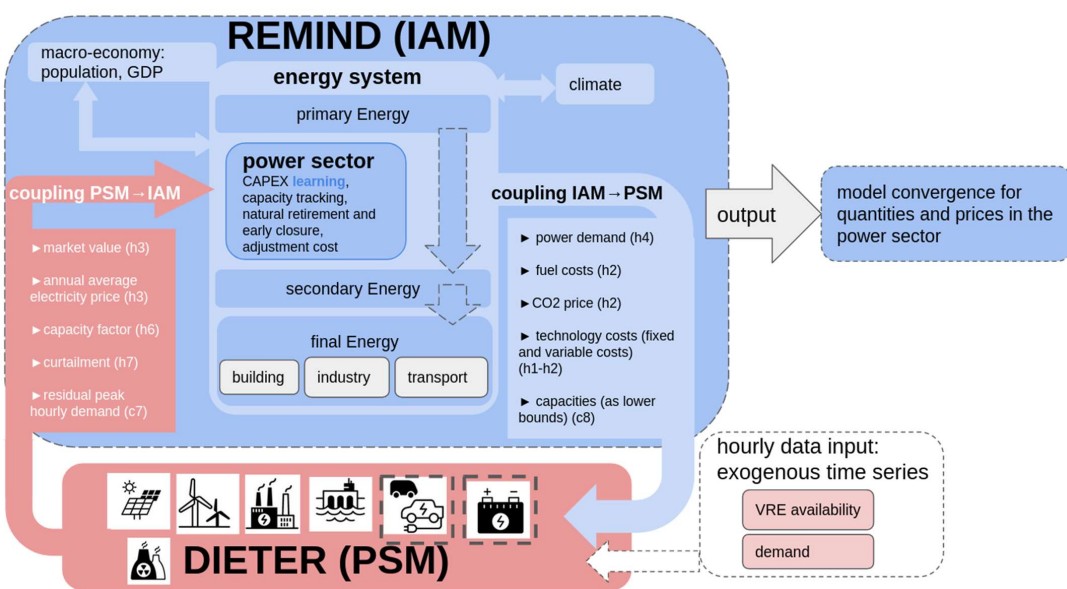


**Figure 3: The schematics of the REMIND-DIETER iterative soft-coupling. The power sector module of IAM REMIND,**
**which is between the layer of primary to secondary energy transformation, is hard-coupled with other modules inside**
**REMIND such as macro-economy, industry and transport. In PSM DIETER, the power market with generators of**
**various types is modeled with hourly resolution, with options for storage and flexible demand. The information**
**exchanged between the models (block arrows) are determined via the convergence conditions (h1-h7) derived before**
**(Sect. 3.2.1). In order to improve performance and facilitate convergence, additional constraints (c7) and (c8) are**
**included in the coupling interface. The coupling interface for REMIND → DIETER is programmed as a part of modified**
**DIETER code, and vice versa. Both interfaces are written in GAMS. For a single-region, the scheduling of coupled**
**iterations is illustrated in Fig. E1 in Appendix E. 16 DIETER optimization problems are solved for each representative**



**year of REMIND in parallel, scheduled after each internal REMIND "Nash" iteration (see Sect. 2.1 for a description of**
**the iterative "Nash" algorithm).**

During the coupling, the following exchanges of parameters and variables take place iteratively in both directions via the
interface.

### 3.3.1 REMIND to DIETER

The following information flow from REMIND to DIETER.
1. Technology fixed costs (convergence condition (h1)):
a.   Annualized capital investment cost: It is calculated from endogenously determined overnight investment cost, plant
lifetime, and the endogenously determined interest rate. The overnight investment cost is determined from floor cost,
learning rate and the endogenous global accumulated deployment. Note that investment costs decrease according to
endogenous learning rate. Interest rate is about 5% on average but is endogenous and time dependent in REMIND;
b.   Annualized operation and maintenance (O&M) fixed costs (OMF): They are a fixed share of the capital costs;
c.   Adjustment cost: It is technology-specific and is proportional to the capital investment cost. See Appendix I for its
implementation.
2. Technology variable costs (convergence condition (h2)):
a.   Primary energy fuel costs: They are endogenously determined as the shadow prices of the primary fuel balance
equations in REMIND. Import prices, domestic prices of extraction, amount of regional reserve, and the amount of fuel
demand can all influence the fuel cost. The relevant fuel costs include coal, gas, biomass and uranium. The fuel costs
can have interannual intertemporal oscillatory components which can cause instability during iteration if coupled
directly. We mitigate this by conducting a linear fit to the time series before passing them to DIETER;
b.   Conversion efficiency of each generation technology;
c.   O&M variable costs (OMV);
d.   $CO_2$ emission cost: Exogenous or endogenous $CO_2$ price from REMIND multiplied by the carbon content of a type of
fossil fuel and divided by the conversion efficiency of a generation technology gives the $CO_2$ cost of 1MWh of
generation. Note that in REMIND, biomass is considered to contain zero carbon emission when combusted.;
e.   Grid cost: In REMIND the stylized grid capacity equation is proportional to the amount of pre-curtailment VRE
generation. So effectively the grid cost is a variable cost. Note that in future work, grid costs can be modeled in more
detail either in DIETER or in another PSM. Here, we use the parameterized grid costs which are implemented in
default REMIND as an approximation to the necessary grid cost.
3. Power demand (convergence condition (h4)). REMIND informs DIETER of the total power demand $d_y$ of a representative
year $y$. In the next iteration of DIETER, the exogenous time series for the hourly demand from a historical year (2019) is
scaled up to demand of the last iteration REMIND, $d_y(i-1)$, such that the annual total power demand in DIETER is equal
to that of REMIND for each coupled year: $\underline{d}_h = \underline{d}_{2019,h} * \frac{d_y(i-1)}{\sum_h \underline{d}_{2019,h}}$ .
4. Pre-investment capacities $P_{y-\Delta y/2,s}/(1-ER)$ as an additional brown-field constraint (see constraint (c8) in Appendix G).
$ER$ is the endogenous early retirement rate in REMIND.
5. Total regional renewable resources for wind, solar and hydro (constraint (c2)), such that DIETER capacities are constrained
by the same total available resources as in REMIND.



6. Annual average theoretical capacity factors of VREs and hydroelectric in REMIND (convergence condition (h6)). We note
the pre-curtailment utilization rates of VRE capacity as "theoretical capacity factors", as these can be achieved in theory if
there is no curtailment. They are usually determined by meteorological factors such as wind and solar potential, as well as
the efficiency of the turbines or solar photovoltaic modules. In contrast, the post-curtailment utilization rate of VRE are "real
capacity factors", as these are the real utilization rates after optimal endogenous dispatch. The time series of theoretical
utilization rate of VRE generations of one historical year in DIETER are scaled up such that the annual average theoretical
capacity factors in DIETER equals the exogenous parameters in REMIND:
$$\underline{\phi}_{h,vre}(y) = min\left(0.99, \underline{\phi}_{h,vre}(y=2019) * \frac{\phi_{vre}}{\sum_h \underline{\phi}_{h,vre}(y=201\ )}\right).$$
In DIETER, to be realistic, the rescaled hourly capacity factor for solar and wind has an upper bound at 99%. The slight
mismatch of the capacity factors due to this additional upper bound is negligible
**3.3.2 DIETER to REMIND**
The following information is passed from last-iteration DIETER to REMIND:
1. Market values $\underline{MV}'_{y,s}$ and the annual average electricity price $\underline{J}'_y$ (convergence condition (h3)), where $\underline{MV}'_{y,s}$ is the annual
average market value without the surplus scarcity hour price, and $\underline{J}'_y$ is the annual average electricity price without the
surplus scarcity hour price.
2. Peak hourly residual power demand $\underline{d}_{residual}$ as a fraction of total annual demand $\sum_h \underline{d}_h$ (constraint (c7)). This produces the
peak residual demand in REMIND $d_{residual,y}$ that is proportional to the last-iteration DIETER peak to total demand ratio
$\frac{d_{residual}(y,i-1)}{\sum_h \underline{d}_h(y,i-1)}$, and the in-iteration total annual demand $d_y(i)$:
$$d_{residual,y}(i) = \frac{d_{residual}(y,i-1)}{\sum_h \underline{d}_h(y,i-1)} * d_y(i) \,,$$
where $\underline{d}_{residual}$ was defined in Appendix H (Eq. (H1)).
3. Annual capacity factors of dispatchable plants $\underline{\phi}_{dis} = \frac{\sum_h \underline{G}_{h,dis}}{\underline{P}_{dis}* 8760}$ (convergence condition (h6)).
4. Annual solar and wind curtailment ratio: curtailment as a fraction to total annual post-curtailment generation $\frac{\sum_h \underline{\Gamma}_{h,vre}}{\sum_h \underline{G}_{h,vre}}$
(convergence condition (h7)).
For the information flowing from DIETER to REMIND, we use an innovative method of multiplicative "prefactors", which can
stabilize the coupling and increase the speed towards model convergence. The prefactors are automatic linear stabilizers of the
current-iteration variables in REMIND. They depend on current-iteration endogenous variables in REMIND, and are multiplied
usually with the last-iteration endogenous DIETER results that are exogenously passed to REMIND. This allows some degree of
endogeneity in these exchanged variables, and their values can be adjusted according to the updated dynamics in the current
REMIND iteration, such as interregional trading or price-demand elasticity, under which the exogenous last-iteration DIETER
optimality can be used as an approximate starting point but do not necessarily hold exactly.
The prefactors usually depend on the differences between generation shares in the two models: e.g. the prefactor for markup is a
linear function of the difference between the current-iteration REMIND endogenous generation share and last-iteration DIETER
generation share. We illustrate the mechanism of prefactors using markup for solar as an example: A lower market value for
solar is consistent with a higher solar share, according to the well-known self-cannibalization effect of decreasing VRE market
value as the VRE share increases (Hirth, 2018). Therefore, we can introduce an automatic stabilization measure through a
negative feedback loop: If the REMIND endogenous share is larger than in the last DIETER iteration, in which case the in-





iteration market value should be lower than the last-iteration DIETER market value, the multiplicative prefactor for market
value should be so constructed such that it is smaller than one. This lowers the market value for solar, and decreases the in-
iteration REMIND markup $\eta_{y,s}(i)$, hence preventing over-incentivizing the solar generation using the old market value based on
the last-iteration energy mix. Overall, this produces a stabilizing effect on the system by making the markup as a price signal
responsive to endogenous quantity change. We use prefactors ubiquitously when passing variables from DIETER to REMIND,
such that during the iteration REMIND can adjust more smoothly and easily. We discuss the implementation of these prefactors
in detail in Appendix H.2.

## 4 Numerical convergence under "proof-of-concept" baseline scenario

In this section, we check the convergence behavior for prices and quantities (capacity and generation) in coupled model runs
using the convergence validation criteria from the last section. Comparing the numerical results with the theoretical prediction,
we can validate that REMIND-DIETER soft-coupling indeed produces almost full convergence.
Throughout this section, we only use one scenario – a "proof-of-concept" baseline scenario. Under the "proof-of-concept"
scenario of the coupled run, we disable storage (i.e. batteries and hydrogen) and flexible demand (i.e. electrolyzers) in both
models, as this allows us to use the theoretically derived convergence criteria from Sect. 3, which would become overly
complex in a model with storage and flexible demand. The coupled run is under a baseline scenario, i.e. there is no additional
climate policy implementation. Since this is a configuration created only for comparing to the theoretical prediction, it is not
meant to be a policy-relevant configuration. In more policy-relevant coupled runs, we turn on storage and flexible demand (see
Sect. 5). For schematics and computational runtimes of the coupled iterations, see Appendix E.
For the coupled runs, we define a baseline scenario for single-region Germany under SSP2 assumptions, corresponding to the
"middle-of-the-road" scenario (for a definition of the SSPs, see Koch and Leimbach, 2022). Specifically, this means that
REMIND runs for all global regions in parallel, but DIETER only runs for Germany. Only information in the German power
sector is exchanged for the two models. We use a low CO2 price to represent "no additional policy", which is 30\$/tCO2 in 2020
and 37\$/tCO2 for years beyond 2020. According to the 2011 Nuclear Energy Act of Germany, remaining nuclear capacities are
set to early retire in REMIND within the time period until 2022. We assume hydroelectric generation in Germany to come from
run-of-the-river. In DIETER, we cap dispatchable generation's annual capacity factors at 80% for non-nuclear power plants, and
85% for nuclear power plants, so the dispatch results are in line with real-world power sectors. This constraint only adjusts the
capacity factor constraint (c4), which would pose no additional distortion to our mathematical analysis.
Due to the particular implementation of offshore wind in REMIND, DIETER wind offshore capacities are fixed to that of
REMIND to avoid too much distortion. Since in our scenarios, offshore wind capacity in Germany is relatively small compared
to other generators, this fixing presents only a minor distortion to the coupling. Hydroelectric generation in REMIND is
assumed to have an average annual capacity factor of around 25%. This capacity factor is implemented as a bound in DIETER.
For simplicity, instead of a time series profile for hydroelectric generation, we allow the hourly capacity factor to be no higher
than 90%, meaning hydro is close to being dispatchable in all our scenarios. In the German context, hydro usually means run-of-
the-river, which has a variable output. Nevertheless, we find the 90% maximum hourly capacity factor a reasonable assumption
to make, since in our runs we do not yet consider pumped hydro as a technology in this study, so a more dispatchable quality of
hydro can be assumed. Results presented in this section belong to the same coupled run under the "proof-of-concept" scenario.





**4.1 Electricity price convergence**
According to theoretical convergence criteria (under simplifying assumptions, Sect. 3.2.1-3), at numerical convergence, the
electricity price of REMIND should be equal to the price of DIETER. However, REMIND is interannual intertemporal, whereas
DIETER is only year-long, so we compare the differences over time, as well as the interannual average of the price differences
(Fig. 4).

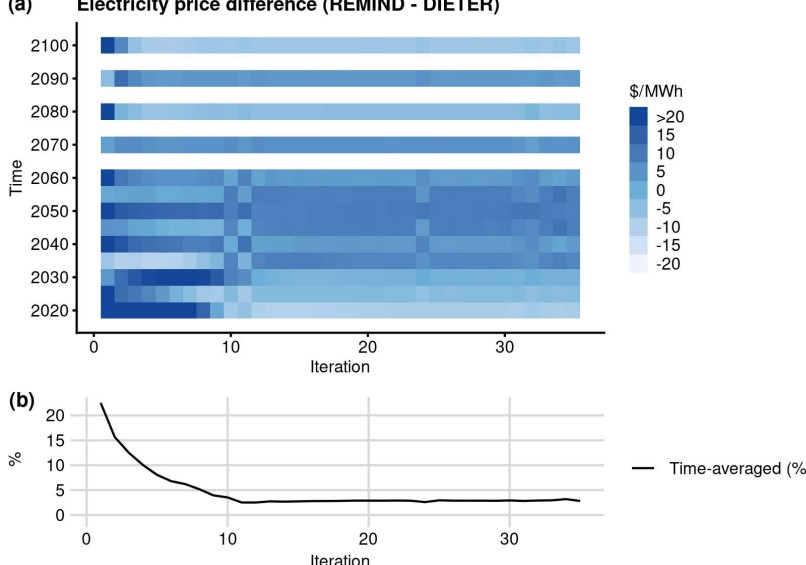


**Figure 4: Annual average electricity price convergence behavior of a coupled run for Germany under a "proof-of-**
**concept" baseline scenario. (a): the difference between the annual electricity price time series of REMIND and the**
**annual average electricity price time series in DIETER as a function of coupled iteration. (b): the interannual average of**
**the differences in (a) as a share of REMIND price. Due to the interannual intertemporal nature of REMIND, in (a) the**
**price difference can appear to have oscillatory components, obscuring the visual assessment of convergence. As a result,**
**we show the trend of price convergence over iterations more clearly in panel (b) by taking the temporal average of the**
**price differences. The REMIND price in both plots is a running average of three neighboring time periods to visually**
**smooth out oscillations.**

In Fig. 4a, the price difference oscillates from period to period. As the coupling starts, the REMIND price is much higher than
DIETER, especially in the earlier years. After around the 10th iteration, the difference in early years starts to reverse: DIETER's
price becomes higher than REMIND. Around 2040-2060, REMIND has a higher average price than DIETER, due to the VRE
market values being higher than their LCOE. This is discussed later in Sect. 4.3.2.
In Fig. 4b, we calculate the difference between two time series – the time-averaged power prices in the two models. We observe
the difference between them decreases over the iterations, showing a clear converging trend, and stabilizes at around 3% of the
REMIND price. There are two observations regarding the price convergence of the coupled run. First, the convergence happens
rather quickly within 10 iterations. Second, the converged value of the price difference is not exactly 0, but slightly above 0, at a





few percent of the full price (a few $/MWh). Under ideal convergence conditions, according to (v1), the two prices should be
equal at full convergence for every coupled year. However, in practice, the average prices do not perfectly match, as there are
several sources of distortions from capacity shadow prices. The capacity shadow prices come from many sources in both
models: extra constraints such as (c7-c8) which are not part of the analysis leading to (v1), constraints that are in REMIND but
not in DIETER (c5-c6), and exogenous wind offshore capacity in DIETER. Some of these capacity shadow prices in both
models can be more or less consistent with each other (such as standing capacity constraint in DIETER and brown-field
constraints in REMIND), but others are not and can distort two models in different ways, causing some degrees of misalignment
in prices. As discussed before, prices can be overdetermined by the energy mix (Sect. 3.2.3). Therefore, some of the capacity
shadow prices – even though not aligned between the two models – can nevertheless cancel each other (especially averaged over
time), potentially causing the price differences to be moderate. To examine exactly how well the prices at the end of the
coupling match, we need to check the cost decomposition of prices. This is discussed later in Sect. 4.3.
Also note that Fig. 4b presents a time-averaged price comparison, and on average the difference between the prices in the two
models is small at the end of the coupling. However, when one compares the maximal deviation for any single year at the end of
the coupling, it can be as high as 10$/MWh, e.g. around 2050 (Fig. 4a). This is much larger than the 3% averaged deviation in
Fig. 4b. However, compared to default REMIND prices (which we cannot show due to limited space), we are fairly confident
that the oscillation of coupled REMIND results from internal dynamics that are also visible in the default uncoupled version. So
a time-averaged treatment is adequate in displaying total price convergence here.

### 743 4.2 Quantity convergence

Besides price convergence, the capacity and generation decision variables must also converge within a certain tolerance at the
end of the coupling. This is reflected in the generation mix (Fig. 5) and the capacity mix (Fig. 6) at the end of the coupled run.
Due to the existence of several sources of mismatch between the two models already mentioned in the last section, which is
already manifested in the mismatch in electricity prices of the two models, a certain degree of mismatch in quantities is also to
be expected. Nevertheless, the agreement between the two endogenous sets of decision variables is satisfactory. For this coupled
run, the differences of the generation share of any single technology between the two models are smaller than 4.4% for each
year until 2100. Figure (5b) highlights some subtle model differences in generation. For example, after 2040, REMIND favors
solar and coal, whereas DIETER tends to have more combined cycle gas turbines (CCGT) and wind onshore. Due to the low
capacity factor of OCGT and solar compared to the capacity factors of the other generators, the capacity mix differences
between models are amplified for these two technologies (Fig. 6). But overall, the generation mixes and the capacity portfolios
at the end of coupled run are generally similar.



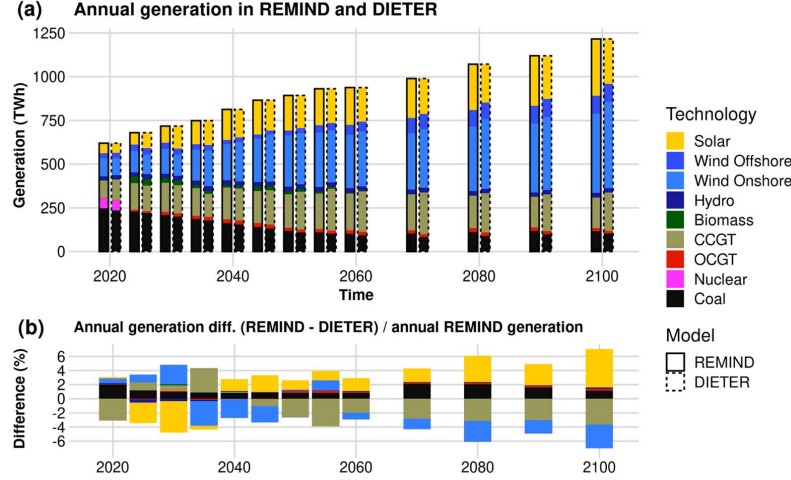

**Figure 5: Annual electricity generation convergence at the final iteration of a coupled run for Germany under the "proof-of-concept" baseline scenario. (a) Side-by-side comparison of the two generation portfolios at the end of the coupled run. (b) The difference between the generation mix in the two models as a share of total REMIND generation.**

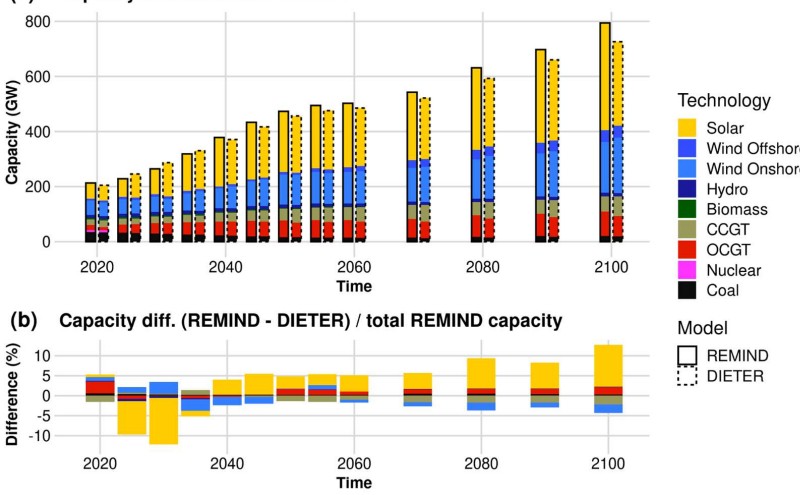

**Figure 6: Capacity convergence at the final iteration of a coupled run for Germany under the "proof-of-concept" baseline scenario. (a) Side-by-side comparison of the two models' capacity mix at the end of the coupled run. (b) The capacity difference between the two models as a share of total REMIND capacity.**

For periods that are policy relevant in the short- to medium-term (i.e. before 2070), the convergence for quantities is generally slightly worse in the near-term, i.e. in the 2020s and 2030s, likely due to the capacity bounds mismatch in the near-term (such as the capacity bounds (c5-c6) in REMIND not being completely replicated by standing capacity constraint (c8) in DIETER). If DIETER does not contain identical bounds as REMIND, then its endogenous decision will have more of a green-field rationale





than REMIND does, the latter of which is more constrained in the near-term. In case an improvement of near-term convergence
is desired, these bounds could be implemented more carefully, and more technology-specific. Due to the limited scope, we only
apply a generic standing capacity constraint (c8) in DIETER to represent the basket of various constraints. The convergence of
quantities is also not perfect in the green-field periods, such as after 2040, where both models are less constrained by near-term
dynamics. The reason for this is likely due to the fact that in DIETER, hydroelectric generation is not economically competitive
against other cheaper forms of generation such as solar and wind. But in REMIND it is economically competitive, likely due to
the long life-time of the plants. Semi-exogenous wind offshore capacitates in both models could also play a role. This is
discussed in more detail in Section 6.1.

**4.3 Zero-profit rules for the coupled model**

As our analytical discussion showed before in Sect. 3.2.3, model equilibria in the form of ZPRs are useful in validating
convergence in a more detailed way by decomposing prices into cost components as well as any perturbation from capacity
shadow prices. In this section, we first compare the system LCOE, price and capacity shadow prices of the two models for ZPRs
on the system level, then we show the technology-specific ZPRs. Using this validating step, we can visually ascertain that the
cost components and prices/market values in the two models are remarkably similar on the system level as well as on the
technological level, demonstrating that the underlying principle behind the coupled convergence holds to a good degree.

**4.3.1 System-level zero-profit rule**

At the convergence of the soft-coupled model, we expect ZPRs to be satisfied for the two systems individually (Eq. (11) for
REMIND and Eq. (13) for DIETER), i.e. each price times series also matches the LCOE time series to a good degree, barring
distortions from the capacity shadow prices. This is to say, under full convergence, the time series of system LCOE, and the sum
of the time series of the electricity prices and time series for capacity shadow prices for both models should overlap one another
within numerical tolerance. The costs and prices at the last iteration of the coupled run are summarized in Fig. 7. The electricity
prices derived from the shadow prices of the balance equations are shown in dark grey: (a), REMIND electricity price $\lambda_y$, (b)
DIETER annual average electricity price $\underline{J}_y = \frac{\sum_h \lambda_{y,h} \underline{d}_{y,h}}{\sum_h \underline{d}_{y,h}}$ . Adding all the sources of capacity shadow prices, we obtain the blue
lines: (a) REMIND capacity constraints (c5-c7), (b) DIETER capacity constraint (c8). All capacity shadow prices have been
converted to per energy unit via capacity factors. (Note: Fig. 4 shows the difference between the black lines, without considering
the capacity shadow prices. See Sect. 3.2.3.)
From Fig. 7, we can conclude that the ZPR for DIETER is satisfied to very good accuracy for every year (the blue line – the
sum of electricity price and capacity shadow price has exactly the same value as the sum of LCOE bars). For REMIND, the ZPR
is satisfied year-on-year to a lesser degree, but on average to a good degree given the interannual fluctuations. The prices in
coupled REMIND become very erratic for the early years (2020-2025), likely due to the interaction between the historical or
near-term bounds in REMIND and the exchanged information from DIETER for those years. The LCOEs component structures
match well across the models for most years, which serves as additional visual support on price convergence shown in Fig. 4,
i.e. the cost structures behind the prices are harmonized as well at the end of coupling. The origins of the differences between
LCOEs and prices, as well as the degree with which capacity shadow prices account for them, can be found when one examines
the LCOE and market values of specific technologies, which are analyzed next.



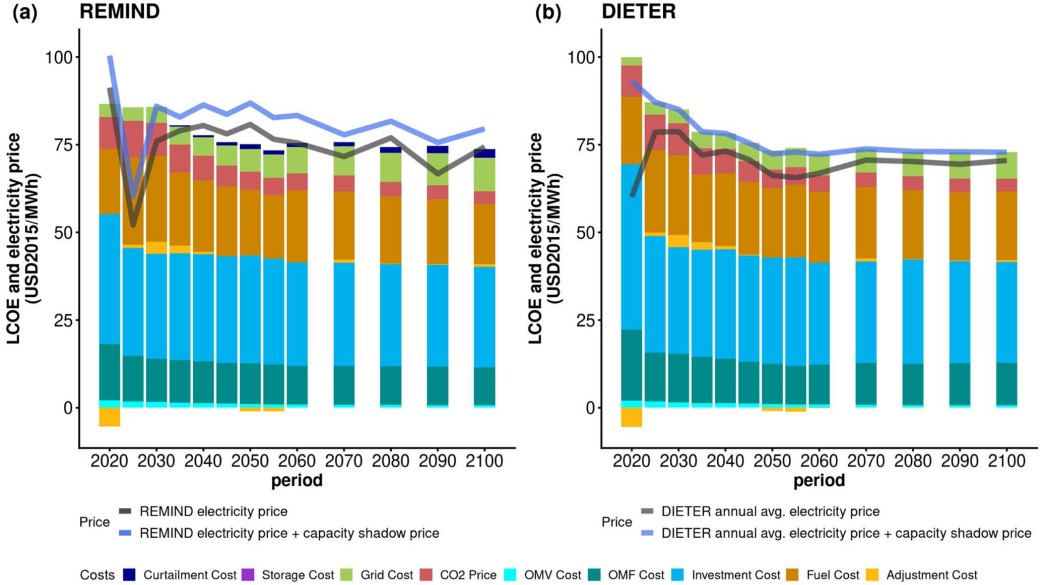


**Figure 7: Cost components of the system LCOEs (bars), electricity prices (grey lines) and the sum of electricity prices and capacity shadow prices for (a) REMIND and (b) DIETER under "proof-of-concept" baseline scenario. Visually the ZPRs for both models are satisfied within numerical tolerance. The intertemporal structure of the LCOE breakdown is very similar for most of the coupled periods. For DIETER, a small remaining difference exists between the price (grey line) and the LCOE (bars), which can be entirely explained by the capacity shadow price due to the standing capacity constraint. The REMIND price time series is a rolling average of 3 time periods. The large negative adjustment costs in 2020 are due to coal and nuclear phase-out.**

**4.3.2 Technology-specific zero-profit rule**

After validating ZPRs on the system level, we further dive into each technology and check the ZPRs for each technology in both
models at the last iteration of the coupled run (Fig. 8).



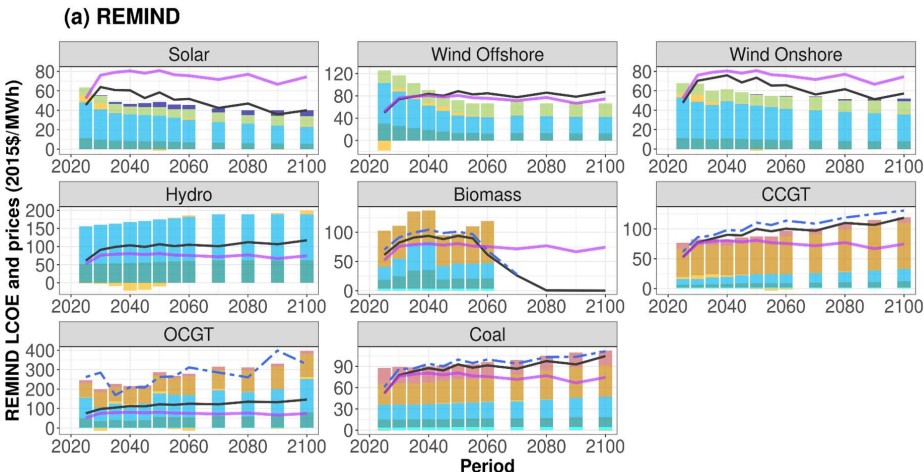

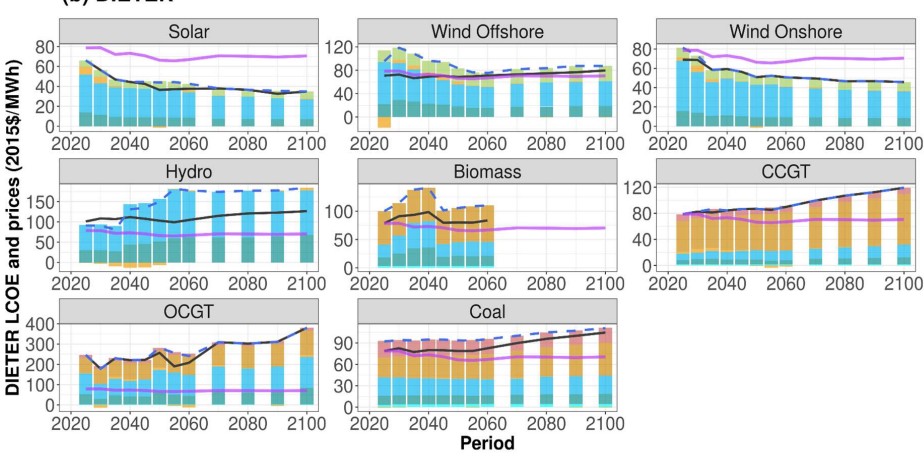

**Figure 8: Technology-specific costs and market values for (a) REMIND and (b) DIETER under "proof-of-concept"**
**scenario. Cost components of the technology LCOE are plotted in stacked bars. Market values are shown in solid black**
**lines. The sum of market values and all sources of capacity shadow prices are shown in dashed lines: for DIETER (two-**
**dash blue lines), they contain mostly the standing capacity shadow price, and to a small extent the capacity shadow**
**prices of the resource constraint; for REMIND (dashed blue lines), they contain mostly the peak demand capacity**
**shadow price, and small capacity shadow prices due to brown-field and resource constraints. Electricity prices are**
**shown in purple solid lines as references. Due to large positive shadow prices in 2020 due to fixings to the historical**
**capacities, only periods beyond 2020 are shown. REMIND market values and capacity shadow prices are a rolling**
**average of 3 time periods.**




In Fig. 8(b), DIETER LCOE and market values for the eight types of generators are shown. As expected from the ZPR, the
LCOE always matches the sum of the market value and capacity shadow prices for each technology, and for each year (Eq.
(12)). The difference between the dashed and solid lines are largely the generation capacity shadow prices. It is worth noting
that at the end of convergence, the sizes of the shadow prices are in general small for the main generator types, e.g. solar, wind
onshore, CCGT and OCGT. This indicates the fact that for these technologies for most periods, the optimal DIETER generation
mix is close to that of a green-field model. That is, DIETER hardly faces any exogenous constraints (except resource constraints
that are aligned with those of REMIND) and can make fully endogenous investment and dispatch decisions based on cost
information alone. On the whole, DIETER at the coupled convergence experiences only a small amount of distortion from the
brown-field model REMIND, especially concerning the "model suboptimal" real-world standing capacities from biomass, hydro
and coal.
In Fig. 8(a), we show the REMIND LCOE and market values for the same generation technologies. Due to the intertemporal
nature of REMIND, the sum of market value and capacity shadow price for each technology, and for each year matches the
LCOE generally slightly less well than DIETER. This means for REMIND the ZPR (Eq. (10)) for each generator type is also
satisfied to a good degree for main generator types, e.g. solar, wind onshore, coal, CCGT and OCGT. The mismatch in biomass
and hydro might come from the shadow price from historical capacities.
Since the differences between market values and costs are accounted for by capacity shadow price to a large degree, it is worth
interpreting physically the sources of these "hidden" costs/revenues. For REMIND, the capacity shadow prices consist of those
in (c2), (c5), (c6), as well as the "peak residual demand constraint" from DIETER (c7). Constraint (c7) is created to circumvent
high markups especially from peaker gas plants (Appendix H.1). Because peaker gas plants generate power mostly only at hours
with high prices (especially scarcity hour price), and therefore have very high market values compared to annual average
electricity price. The high market values of OCGT – usually more than 5 times the average annual electricity prices – acts as a
large incentive in the next iteration REMIND, and leads to overinvestment in capacities. Over iterations, this causes oscillations
in the quantities and prices in the coupled model and prevents model convergence. To circumvent the issue of high markup, we
implement (c7) as an equivalent peak residual demand constraint. As can be shown mathematically (Appendix H), (c7)
generates essentially the scarcity hour price, and it is very easy to validate this for OCGT in Fig. 8(a). The capacity shadow
price derived from this peak residual demand constraint, when translated to energy terms and added to the market value,
correctly recovers the LCOE for OCGT, recovering the original ZPR (Appendix H.1.2). This indicates that under multiscale
model coupling, an extra constraint is an effective way to circumvent potential issues of numerical divergence due to the large
impact from short-term dynamics, such as the large market value of peaker gas plants.
For DIETER, the two sources of capacity shadow price are the total renewable potential limit (constraint (c2) in Sect. 3.1), and
the standing capacity constraint from REMIND (constraint (c8) in Sect. 3.2.3). For the first type, the resulting capacity shadow
price is a hidden "positive cost" from the perspective of the power user. Since endogenously DIETER would like to invest more,
but is limited by the natural resources available. An example for this first type is hydroelectric power between 2020 and 2035,
due to the limited resource (run-of-the-river) in Germany. It is worth noting that from the generator's perspective, the capacity
shadow price from resource constraint can be interpreted as an extra resource rent. The second type of capacity constraint
originates from the standing capacity, the latter is received by DIETER from REMIND as a lower bound. This constraint usually
results in a hidden "negative cost" from the perspective of a power user, i.e. a part of the cost (LCOE) does not get passed on to
the electricity price, so the users get part of the capacity "for free". (This can also be interpreted as subsidies for generators to
sustain these unprofitable capacities.) This is because based on greenfield cost optimization, DIETER endogenously would
invest less in certain technologies. However, since the standing capacities account for the existing generation assets in the real





world, which can be model suboptimal, the overall costs are above a greenfield equilibrium and above the prices the user pays.
We find examples of such a capacity shadow price manifested in biomass, coal and hydroelectric, all of which are part of the
existing German power capacity mix, but evidently not all of them for any given period are "green-field optimal" based on pure
cost consideration in DIETER. Interestingly, after 2035, the sign of the capacity shadow price for hydroelectric generators
reverses. This is likely due to the continuous decline of the VRE costs after 2035 tips the power sector into a regime where
hydroelectric becomes less economically competitive in DIETER, at least compared to REMIND. As a result, the standing
constraint from REMIND starts to be binding on the capacity from below, relieving the resource constraint binding from above.
For DIETER, the capacity shadow price from standing capacities also indicates the degree of disagreement between DIETER
and REMIND. For most future years, REMIND standing capacity constraints are not binding in DIETER for solar, wind
onshore, CCGT and OCGT, indicating good agreement between the models. The small amount of shadow prices near 2060 for
OCGT and solar in Fig. 8(b) are likely due to the time step size change in REMIND which causes a small jump in the interest
rates near these years.
Lastly, in Fig. 4 before we observe a slightly higher average electricity price in REMIND than in DIETER, especially in the
intermediate years. This could be due to fixed offshore wind capacities, which are never economical to be invested
endogenously in the parameterization used here. This generates a high capacity shadow price until around 2045-2060, visible in
both DIETER and REMIND.
**5 Scenario results under baseline and policy scenarios**
In this section, we present baseline and policy scenario results for Germany, using a more realistic configuration of the coupled
model with electricity storage and flexible electrolyzer demand for green hydrogen production which is then used outside the
power sector (e.g. in industry or heavy trucks). We show results for a baseline scenario and a net-zero by 2045 climate policy
scenario. Note that due to REMIND's global scope, under the net-zero scenario we also assume a larger climate policy
background of 1.5C goal for end-of-century temperature rise globally (corresponding to a 500Gt of CO2 emission budget until
2100), and a larger regional goal of EU-wide net-zero emission. Both scenarios consider nuclear phaseout law in Germany.
In Sect. 5.1, we present long-term power sector development. In Sect. 5.2, we present short-term power sector hourly dispatch
and price results. In the following, we broadly describe how these additional features are implemented:
1. Storage: We use a simple storage implementation where DIETER makes endogenous investment into two kinds of storage
technologies:
1) lithium-ion utility-scale batteries;
2) onsite green hydrogen production via flexible electrolyzers, storage and combustion for power production.
The principle of the coupling remains mostly unchanged. REMIND receives the price markups from generation technologies
as in the case before without storage. However, for simplicity, the capacities of storage are not part of endogenous
investment in REMIND. In REMIND, the energy loss due to storage conversion efficiency is taken as a fraction of total
demand from DIETER as a parameter, and stabilized with a prefactor for each type of renewable generation (similar to the
case of curtailment rate in Sect. 3.3.2, 4). Our battery cost development is given in Supplemental Material S1-2.
The reason we only allow DIETER to endogenously invest in storage technologies, is that the additional intertemporal
optimization offered in REMIND is relatively less important than that for the investment of generation technologies. In
REMIND, intertemporality mainly accounts for two aspects in the real-world: 1) implementing adjustment cost and 2)
tracking of standing capacity. The adjustment costs simulate system inertia to rapid capacity addition or removal. In the case
of battery and other storage technologies, the ramp up of deployment faces relatively fewer inertia compared to wind and





solar. Compared to generation technologies such as wind and solar, the storage technologies tend to have lower total
capacities, meaning their ramp up rate is usually lower. Also, their deployment is mostly constrained by their higher cost.
For utility storage technologies, they are mostly not yet deployed at scale, which means there is very little existing capacity,
the investment for storage in REMIND is mostly green-field, rendering it unnecessary to give DIETER a standing capacity
of them.
2.  Flexible demand: As a simple representation of flexible demand, we choose to implement a common Power-to-Gas (PtG)
technology, namely the so-called "green hydrogen" electrolysis. We split the total power demand required to produce green
hydrogen from REMIND from the total power demand $d_y(i-1)$ (Sect. 3.3.1, 3) – both demands are endogenous in
REMIND. We implement the electrolysis demand as completely flexible in DIETER, i.e. there is no ramping cost or
constraint. Thereby flexibilizing part of endogenous total power demand $d_y(i-1)$ in REMIND. As a result, the cost
minimization in DIETER automatically allocates the flexible demand to hours where electricity costs are low due to the
existence of low-cost VRE. The economic value of flexible demand can be quantified by the capture price. The annual
capture price of demand-side technology $s_d$ is the annual average price of the hours when the flexible demand consumes
electricity, weighted by the hourly flexible power demand by electrolyzers: $\underline{CP}_{s_d} = \frac{\sum_{h,s_d} \underline{d}_{h,s_d} \lambda_h}{\sum_{h,s_d} \underline{d}_{h,s_d}}$.
**This concept is equivalent to the market value for a variable or dispatchable generator, but here for a flexible or**
**inflexible demand source. Similar to before, we implement a stabilization measure using a prefactor (Appendix H.2, 5).**
**5.1. Long-term development**
This section presents scenario results of the coupled model with a long-term view on capacity and generation, using either the
proof-of-concept scenario or more realistic configurations.
**5.1.1 Baseline scenario**

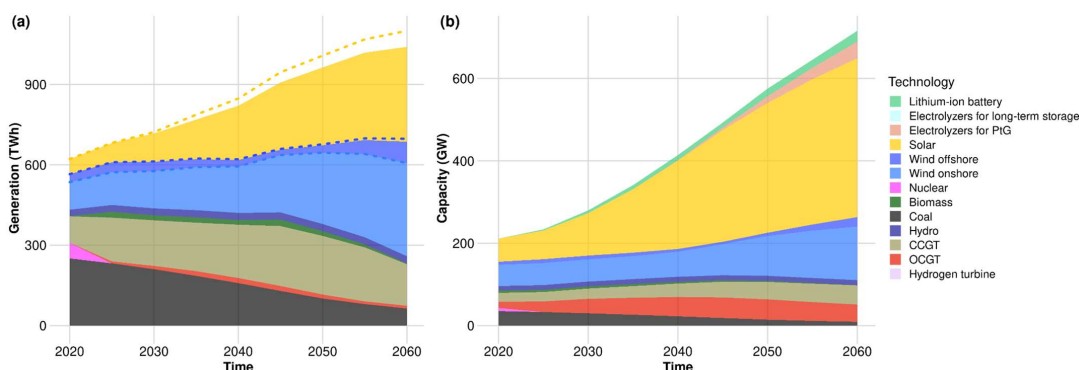


**Figure 9: DIETER-REMIND converged results of the long-term (a) generation and (b) capacity expansion for**
**Germany's power sector in the baseline scenario, assuming a constant 37$/tCO2 CO2 price. Dashed lines represent**
**generation before storage loss and curtailment. Storage generation is not visualized in (a).**

In Fig. 9(a), under baseline scenario, and with available storage and flexible demand, we observe a more than 35% increase of
the total power demand from 2020 to 2045, and more than 65% by 2080. This is due to an increase in end-use electrification.



The increased electrification comes from a moderate growth in electricity use in the building sector and a more significant
growth in EV fleet. In the building sector, the final energy share of electricity is projected to increase from 28% in 2020 to 39%
in 2045. The final energy share of electricity in the transport sector is 22% by 2045, up from 2% in 2020. Note that even under
no additional climate policies, based on only the increase in EVs shares in new-cars sales in many world markets today, we
expect higher power usage from EVs in the future. Within the energy mix, we see a slow decline in coal generation over time,
which is replaced by CCGT generation and a significant increase of VRE. VRE share reaches above 50% by 2045, but slightly
less than half of the energy mix still contains coal and gas power. In terms of capacity expansion (Fig. 9(b)), due to both lower
generation cost and higher power demand, solar capacity expands by almost 5 times from today until 2045. However, the
moderate VRE shares mean that the requirement on battery capacity is not high, namely only 12GW of batteries by 2045. Due
to the low CO2 price, long-term electricity storage through hydrogen does not appear to be economically competitive and is not
invested under the baseline.
By comparing the above baseline scenario (with storage and flexible demand) (Fig. 9) with the "proof-of-concept" baseline
scenario (without storage or flexible demand) before (Fig. 5 and 6), it is clear that while battery storage and partial demand
flexibility play a role after 2040 in increasing the VRE share in Fig. 9, in the near term, the scenarios with and without available
storage and demand flexibility look very similar under no additional climate policies. However, due to technological learning
effect, even absent additional CO2 price policy, the energy mix here has a relatively high VRE share (>60%) after 2050
compared to the basic case without storage and demand-side flexibilization. However, due to the low CO2 price there is still a
significant share of dispatchable technologies such as CCGT and OCGT, which is more economical than the implementation of
long-term power storage via electrolysis and hydrogen turbines.
**5.1.2 Net-zero policy scenario**

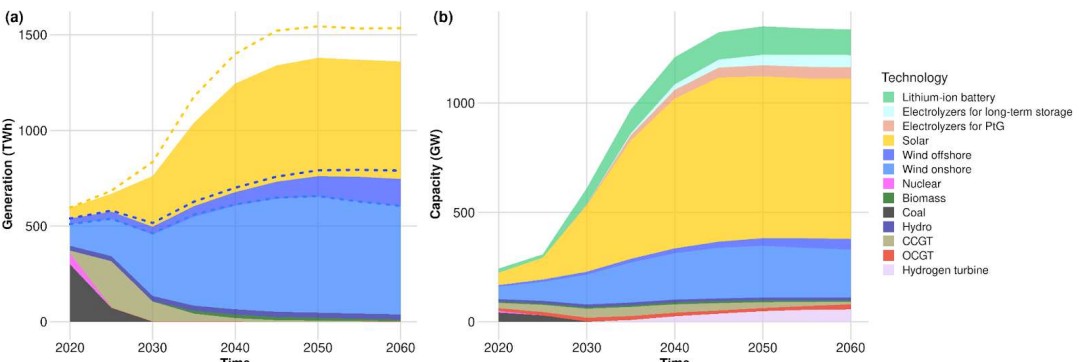


**Figure 10: DIETER-REMIND results of the long-term generation and capacity expansion for Germany's power sector**
**in the "net-zero 2045" scenario. CO2 price is endogenously determined based on the climate goal. It is 115$/tCO2 for**
**2030, 292$/tCO2 for 2035, 464$/tCO2 for 2040, and 636$/tCO2 for 2045. Dashed lines represent pre-curtailment**
**generation. Storage generation is not visualized in (a).**

In Fig. 10, under stringent climate policy (economic-wide carbon neutrality in 2045), with available storage and partially
flexibilized demand (for hydrogen production used in other sectors), the total power demand more than doubles, and the power
mix is dramatically transformed. Compared to both the baseline case without storage and demand-side flexibilization (Fig. 5 and





6) and the baseline scenario with storage and flexible demand (Fig. 9), a very high VRE share in the generation mix is reached already by 2040 (>94%). This is mostly due to an earlier investment in VRE to drive down the cost, combined with the increased deployment of both short- and long-term storage and flexibilization of part of the demand. Capacities for storage increase significantly: lithium-ion batteries from 18GW in 2020 to 125GW in 2045, and 37 GW of hydrogen electrolysis and hydrogen turbine capacity (with ~40TWh of H2 storage capacity). Despite high storage capacities, due to high VRE share, curtailment and storage loss still increases quite significantly with time, especially for solar PV. But note that in a coupled run where interregional transmission expansion is possible connecting Germany and the rest of Europe, this loss can be reduced (see Sec. 6.3). In terms of capacity expansion (Fig. 10(b)), gas power plants are mostly replaced, as hydrogen turbines fill the role of peaking dispatchable plants that guarantee supply for peak demand hours. The CCGT gas turbines are equipped with CCS.

Under the stringent climate policy scenario, dramatic changes in the end-use sectors will be underway in the form of direct electrification and substitution of fossil gas with hydrogen. In the building sector, the final energy share of electricity is projected to increase from 28% in 2020 to 66% in 2045. In transport, the final energy share of electricity is 56% by 2045. In the industry sector, the share of electricity increases from 25% to 63%. By 2045 there is also a notable increase in the use of green hydrogen produced from 45GW flexible electrolyzers (at about 42% average annual capacity factor), amounting to 0.5EJ (3.5 million tons) per year in the final energy, which is primarily used in industry.

## 5.2. Short-term dispatch

In this section, results of hourly resolution are shown and discussed for a selected model year. We use established methods such as residual load duration curves (RLDCs) to visualize the hourly dispatch result, as well as show the hourly generation and dispatch time series for some typical days in summer and winter.

### 5.2.1 Residual load duration curve model comparison

RLDCs can be used to visualize the dispatch of energy system models. Each subsequent curve is calculated by subtracting the generation of a technology from the hourly residual demand curve, and then sorting the remaining demand in descending order. On the left-side of RLDC graphs one can easily check the amount of residual demand not met by variable wind and solar production. The top-most line in RLDC graph is the load duration curve for inflexible demand (excluding the demand from flexible electrolysis for hydrogen production used in other sectors).

In a baseline configuration without flexibilized demand or storage, despite lacking the explicit hourly dispatch, via bidirectional soft-linkage, REMIND could achieve a final dispatch result that replicates DIETER to a satisfactory degree (Fig. 11). This is a combined effect of a convergence of capacities (Sect. 4.2) and full-load hours at the end of the coupled run. In the peak residual demand hour (the leftmost point in the RLDC), the DIETER-coupled REMIND accounts for the requirement of dispatchable capacities via the constraint (c7), and the composition of the mix is replicated from DIETER and correctly guarantees that the peak hourly demand is met.



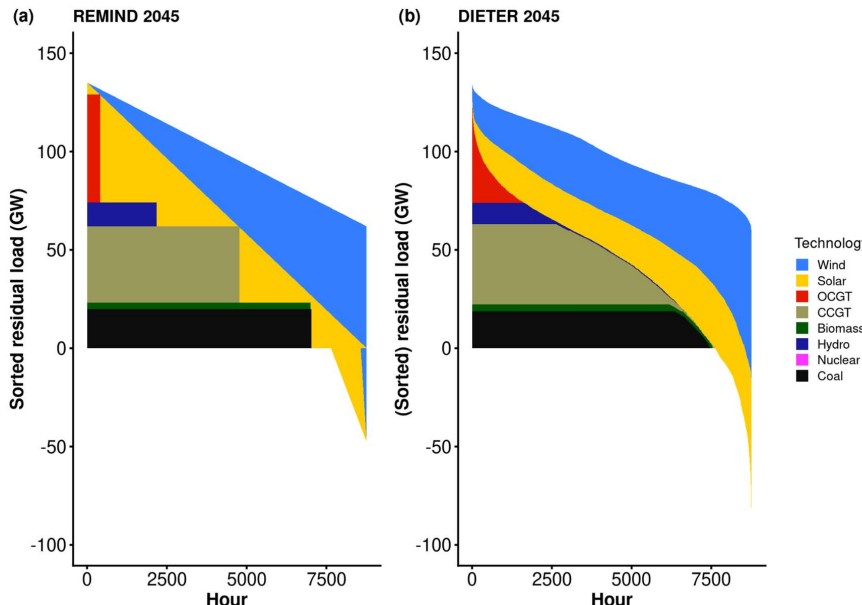

992

**Figure 11: Side-by-side RLDC comparison between (a) REMIND and (b) DIETER for the simple configuration under**
**the baseline scenario without storage or flexible demand. The DIETER RLDC (panel (b)) is constructed by subtracting**
**hourly generation from hourly load and sorting, with dispatchable generation technologies plotted in order of their**
**annual average capacity factors. VREs are arranged such that the generation with higher curtailment rate (i.e., solar, in**
**this case) is on the inside of the graph. To construct the REMIND RLDC (panel (a)), the dispatchable generations are**
**sorted by their capacity factors and stacked from the bottom. The rectangles depicting dispatchable generation are made**
**up by the width equal to the full-load-hour and the height equal to the capacity. The top-most lines on either side are**
**load-duration curves (sorted hourly demand, which is entirely inflexible under this setup). For the purpose of better**
**visualizations, solar and wind RLDCs are tilted at an angle for REMIND and plotted in the same order as the DIETER**
**RLDC. For simplicity, in REMIND wind and solar RLDC share the same top pivot point in peak residual demand hour.**

In net-zero policy with storage and flexible electrolysis demand, comparing dispatch results under both scenarios (Fig. 11 and
12) for model year 2045, it can be observed that under a stringent emission constraint, the system allocates a significant amount
of short-term storage to replace the dispatchable generation such as coal and CCGT. Long-term storage such as hydrogen
electrolysis combined with hydrogen turbines further reduce the capacity factor of remaining OCGT and CCGT. Besides
storage, there is also a significant amount of deployment of flexible electrolysis demand for producing hydrogen (PtG), which is
not used in the power sector, but in industry or heavy-duty transport. The use of PtG technologies leverages cheap variable wind
and solar energy to achieve the goal of sector coupling. By way of storage and PtG, a significant share of the curtailment can be
utilized (more than 70%), either by shifting the supply to times of low VRE production via storage, or by producing hydrogen
using surpluses which can be used in other sectors.





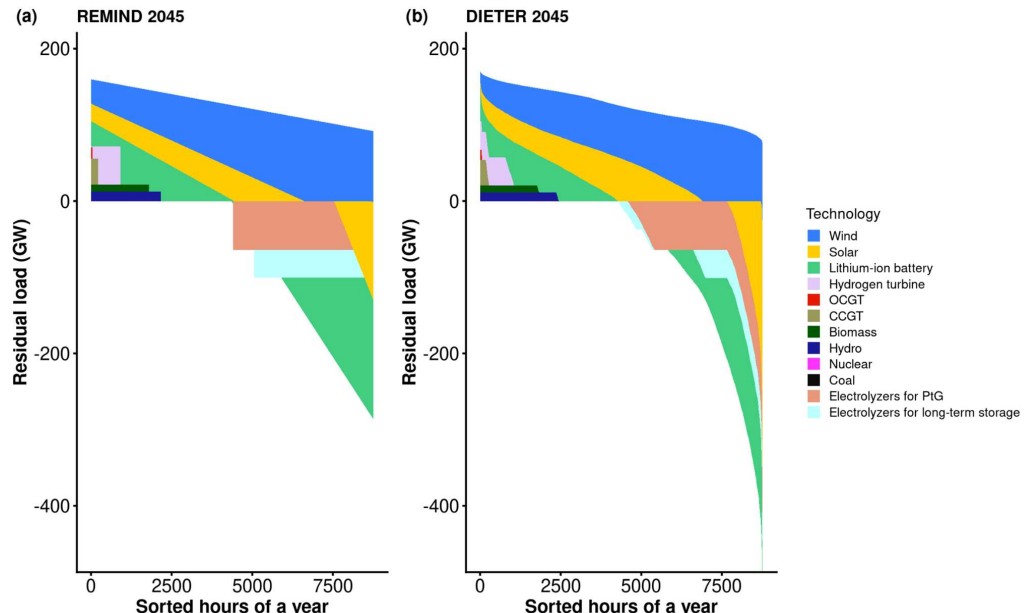

**Figure 12: Side-by-side comparison between (a) REMIND and (b) DIETER RLDCs for net-zero by 2045 scenario with**

**storage and flexibilized demand for Germany. The storage loading and discharging in DIETER RLDC (panel (b)) is**

**constructed by subtracting hourly loading or discharging from hourly inflexible load and sorting. The REMIND RLDC**

**(panel (a)) is constructed similar to Fig. 11. The top-most lines on either side are load-duration curves for inflexible**

**demand. For better visual comparison, in REMIND solar RLDC starts at 80% of the peak residual demand.**

**5.2.2 Hourly dispatch and power consumptions for typical days in summer and winter**

To more directly inspect the results of the hourly dispatch under various scenarios, we visualize the hourly generation and

demand for typical days. Due to the climate in Germany, solar potential is particularly low during winter months. Therefore it is

important to observe the periods in both summer and winter.

From the optimal hourly dispatch results of typical days from the coupled model, we observe that compared to baseline (Fig.

13a-b), in 2045 for a net-zero year (Fig. 13c-d), there is a significant amount of surplus solar generation in the summer during

the day, and some amount of surplus wind generation in the winter during nights and days. Under a net-zero scenario, the

generation from fossil fuel plants in the baseline is replaced by battery dispatch (especially in summer) and hydrogen turbines

(especially in winter), and the peaker plants, which under baseline are turned on in the summer evening, are partially replaced by

solar over-capacity and batteries. A significant share of renewable surplus energy is used for the production of green hydrogen –

hydrogen made from zero-carbon electricity. Due to the complete flexibility of electrolyzers, the capture price of hydrogen

production is only around ⅓ of the average price of electricity (Supplemental Material S2 and Fig. S1 in Supplemental

Material).

In winter, hydrogen turbines serve as a baseload for the few days when wind generation is insufficient to meet the demand. To

ensure supply during longer winter periods of "renewable droughts" with little wind and solar output, e.g., over a 2-3-day period

(hour 540-600 in Fig. 13d), long-term duration storage with hydrogen electrolysis and hydrogen turbines, as well as some

dispatchable generation (such as CCGT with CCS and integrated biomass gasification combined cycle) play a major role.

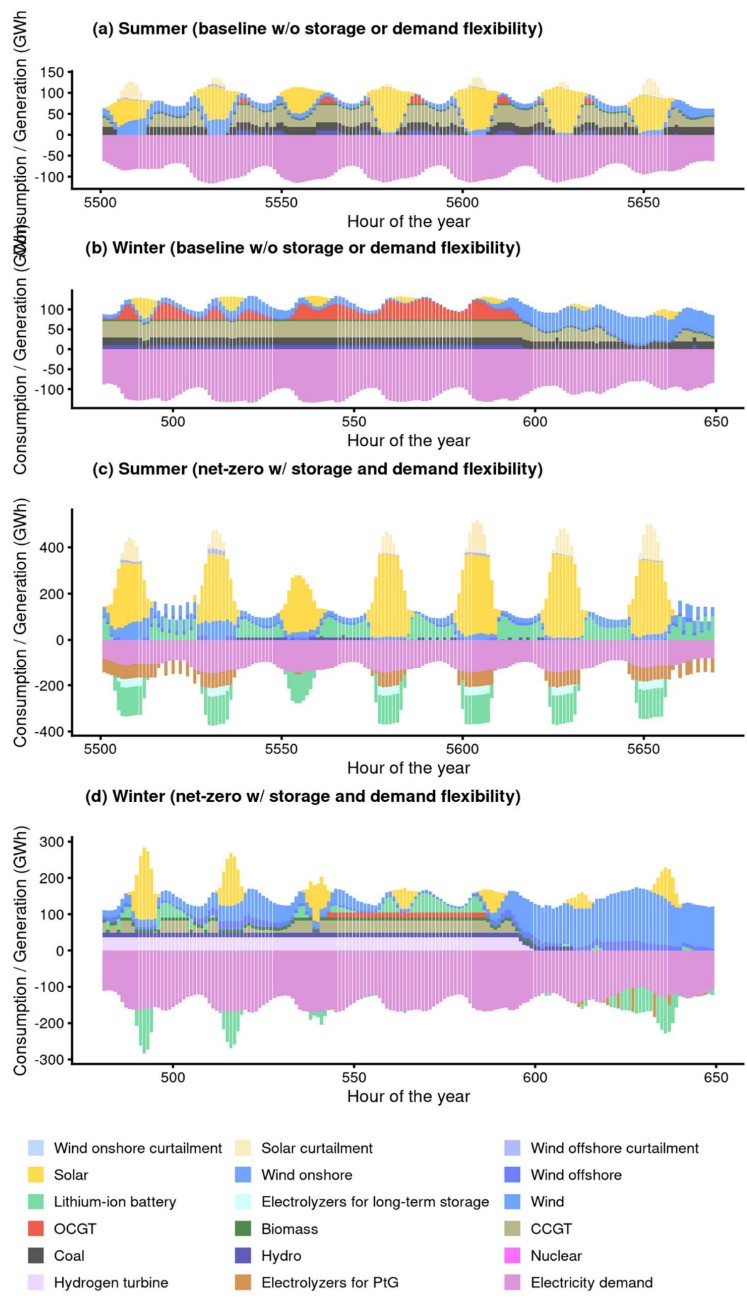

**Figure 13: Comparison of hourly generation (positive) and consumption/storage loading (negative) for a few consecutive typical days in two seasons in Germany in 2045. (a) Summer, under "proof-of-concept" baseline scenario, no storage or flexible demand; (b) Winter, under "proof-of-concept" baseline scenario, no storage or flexible demand; (c) Summer, net-zero scenario, with storage and flexible demand; (d) Winter, net-zero scenario, with storage and flexible demand.**



**6 Discussion**

In this section, we discuss the reasons for remaining differences between the coupled models, as well as the assumptions and limitations of the soft-coupling.

**6.1 Remaining discrepancies**

In all our test runs, at the end of the coupling, it is always the case that the two models cannot be perfectly harmonized, and there is a slight residual difference in the convergence results (Section 4). The reason is two-fold.

The first reason is "legacy mismatch", i.e. a mismatch in brown-field standing capacity constraints in the two models. The coupling method we develop here is mostly based on price information for achieving convergence. Therefore, capacity constraints that are present in the standalone long-term model but not in the standalone hourly dispatch model need to be transferred. These standing capacities are hard to evaluate purely based on economic terms, as they are ultimately a result of real-world actions and policies, which might not align with the simplified economic incentives in techno-economic energy models. Therefore, the only way this information can be transferred from the "brown-field" model to the "green-field" model is by implementing a lower capacity bound in the latter. However, this bound nevertheless might not capture all the shadow prices caused by the standing capacities in REMIND. This is ultimately due to the specific generic form of the constraint we implemented, i.e. we pass on the pre-investment capacities as a lower bound regardless of technology types. In general, hidden "legacy revenues", which are manifested as the shadow prices of economically less competitive generators in DIETER, such as biomass, coal, hydroelectric (solid line lower than bars in Fig. 8), provide incentives for brown-field models to deploy them over long-term, but does not provide enough economic case for the green-field model. This results in an observed phenomenon in the coupled run, that if these "legacy" capacities and their impact on the costs have not been fully transferred to the green-field model, the prices of the green-field model tend to be lower than the coupled brown-field models, causing distortion to the convergence of quantities. The effect of legacy mismatch and illustrative test run results are discussed in more detail in Supplemental Material S3.

The second reason for the discrepancies at the end of the coupling is that there are actual mismatches in the Lagrangian harmonization itself, which can originate from multiple sources. It could due to intertemporal constraints and dynamics (such as adjustment costs and brown-field constraints) not linearly reducible to single-year dynamics, resulting in misalignment between multi-period REMIND and single-year DIETER. It could be also due to slight numerical inaccuracies of the interest rate estimate, which is not explicit in REMIND, but are derived from endogenous and intertemporal consumption. Lastly, there could be a mismatch due to a linear fitting of REMIND endogenous time series of fuel costs (biomass, oil, coal, uranium) before passing this information to DIETER which might a small amount of mismatch for fuel costs between REMIND and DIETER.

**6.2 Limitation of the coupling methodology**

There are limitations to our proposed methodology, both in terms of converging two multiscale power sector models, as well as other potential applications of model convergence. Firstly, in terms of the problem presented here – a multiscale power sector model coupling, the method derived here is only necessary for a full convergence, but may not be sufficient, i.e. a full convergence is not guaranteed. A number of additional factors could prevent a full convergence. One is the "legacy mismatch" and misalignment in Lagrangian mappings mentioned above in Sect. 6.1. Another factor is the role prefactors play (Sect. 3.3.2, Appendix H.2). The prefactors help stabilize the coupling by turning exogenous values obtained from last-iteration DIETER to endogenous values in REMIND, such that they can be adjusted to be in line with the optimal mix of current iteration. However,



they usually contain some small positive or negative parameters that are determined heuristically (e.g. $\underline{b}_{y,s}$ in Eq. (H13)). These
heuristic parameters usually come from rough estimates based on relations between variables in the system and generation
shares, e.g. how much market value of solar generation will decrease when solar generation share increases by a certain
percentage. In practice, while the prefactors help stabilize the run and improve convergence speed, choosing the wrong prefactor
parameters can lead to divergence or instability. Second, another limitation when it comes to modeling power market multiscale
coupling, is the number of products in the market. In the formulation here, both models describe the general equilibrium of a
competitive market with one type of homogenous goods, i.e. electricity. However, if we introduce heat as a by-product, such as
from a combined heat and power plant, then there are two types of goods: heat and electricity. The feasibility of coupling
models with more than one type of goods/market is not yet explored. Thirdly, there are multiple iterative processes that are
internal to REMIND, which happen concurrent with the DIETER-REMIND coupled convergence. Among these processes,
DIETER and the REMIND "Nash" algorithms (for inter-regional trading) both run between the internal REMIND "Nash"
iterations, which means they are external to the REMIND single-region optimization problems and therefore are soft-linked.
Nevertheless, in our runs, we observe the power sector convergence to be rather swift and smooth, and happen in parallel to
other iterative processes, such as the "Nash" algorithm and the $CO_2$ price path algorithm (for climate policy runs). However, a
systematic monitoring of the multiple internal convergence processes in REMIND during the REMIND-DIETER convergence
processes under other model setups and configurations is still to be more thoroughly researched.
More generally, the approach developed here – the Lagrangian mapping method for converging two multiscale optimization
problems – could be useful for a general modeling of market equilibrium of multiple time resolutions. In this study, the
resolution in the coupled problems is specifically only meant for temporal resolution. However, mathematically speaking,
coupling models of different spatial resolutions (or both temporal and spatial resolutions) should be very similar. At least in
theory, the soft-coupling approach developed here should be applicable to increasing the resolution in any arbitrary
independent/orthogonal dimension of the problem of finding equilibrium market dynamics. In theory, it is also possible to build
a multi-layer coupled problem architecture, where at each level the low-resolution variables can be disaggregated into finer
resolution along some dimensions. However, further research is needed to explore the feasibility and convergence performance
of such schemes.

### 6.3 Limitation of coupled results

Since the nature of this study is a proof-of-concept, the scenario results presented should be primarily interpreted as such.
Nevertheless, it may be useful to enumerate a list of limitation for a more accurate interpretation of the results:
1)  The power-sector is only coupled for one single global region, i.e. information exchange only occurs for the variables
of one region – Germany, while all other regions contain the low-resolution version of the power sector of uncoupled
REMIND. The former coupled one-region result is based on a time series of VRE production today in a world of low to
medium VRE share and very limited power grid expansion (in 2019). The latter results of the uncoupled regions
however are parametrized based on results from detailed PSM under a more optimistic assumption of transmission
build-out, which allows VRE pooling from an expanded EU-wide power grid to smooth out regional weather variations
(Ueckerdt et al., 2017). Note that in standalone REMIND, while by default there are no annual electricity import and
export imbalances between countries and regions, transmission during the year is implicitly assumed, especially for the
EU region. Comparing the capacity and generation mixes of the coupled and uncoupled runs (Appendix J), we find that
in the uncoupled case, there are slightly more solar and wind capacities and generations, and much less gas generation





in the long term. EU-wide transmission expansion would pool both supply and demand variability, thus reducing the
need for dispatchable capacity for meeting the peak demand.
2)   Due to the scope of this study, we implemented a limited set of options on storage and sector coupling technologies in
this study, and neglected the additional supply-side details for the German power market (such as the reserve market).
Many potentially significant technological options consisting of pumped hydro storage, compressed-air energy storage,
vehicle-to-grid, and flexible heat-pumps are not explicitly modeled.

3)   Ramping costs for dispatchable generators are not considered, although the effect should be small (Schill et al., 2017).
4)   In terms of power transmission and trading inside Germany, we assume a very simple "copperplate" spatial resolution,
not explicitly modeling transmission bottlenecks inside the region.

5)   Near-term events: we have not modeled the current gas and energy crisis in Europe, which is likely to imply an
overestimation of near-term gas availability in the power sector. Relatedly, we are likely to have overestimated the
early retirement of coal power plants, which are capped at maximum 9% per year of current capacity early retirement
rate in REMIND if it is uneconomical relative to cheaper sources of generation. We have included the COVID shock to
the GDP projection.

6)   Only one weather year (2019) is used for the DIETER input data. From the perspective of sufficient power supply
under all weather conditions with few blackout events, this could introduce an underestimation of the need for reserve
capacity, storage and demand-side flexibility.

## 7 Conclusion and Outlook

In this study, we develop a new method of soft-coupling an IAM with a coarse temporal resolution and a PSM with an hourly
temporal resolution. Our coupling method can be shown both mathematically and in practice to produce a convergence of the
two systems to a sufficient degree. This method allows the incorporation of the temporal details of variable renewable
generation explicitly in large-scope IAM modeling frameworks, and increases the accuracy of power sector dynamics in long
term models. Furthermore, it allows a more explicit modeling of the power sector and sector-coupling, a vision of the energy
transition where end-use demand sectors such as building, industry and transport make economic use of the generation from
variable sources by
1)   directly using the power at the time of production for inflexible form of demand,
2)   shifting time of power supply via battery and other power storage technology,
3)   transforming it to another energy carrier or product ahead of time of consumption and at times of surplus wind and
solar production (e.g. PtG), without conversion back to electricity.

The fully coupled framework allows a more explicit modeling of economic competition of these options under high shares of
variable renewables, finding more accurate optimal paths under long-term climate scenarios towards a net-zero power sector and
the wider economy globally. In future research we plan to expand the study in the direction of demand-side management and
flexibilization, and later possibly in the direction of heat storage.
Coupling DIETER to the global model REMIND for the single region Germany, this study serves as a proof-of-concept. Our
main innovation is two-fold: we derive convergence theoretically, and show almost full convergence numerically. Theoretically,
we derive the coupling methodology by mapping the KKT Lagrangians of the simplified versions of the two models. One key
aspect of the mapping consists of iterative adjustment of the market value (i.e. the annual average revenue of one energy unit of
generation) or the capture price (i.e. the annual average price of one energy unit of consumption) in the low-resolution IAM
such that they take on the values as those in the high-resolution PSM. By finding the set of mathematical coupling conditions



necessary for an iterative convergence as defined by the convergence of both quantities and prices, we could then design the
coupling interface accordingly, such that at the end of the coupling a joint optimal result can be found.
Numerically, we compare the converged results of the two models by examining the long-term power mix (both capacity and
generation quantities), prices of electricity, as well as generation dispatch (via RLDC), and find good agreement between the
two models at the end of coupled convergence despite some slight mismatches. For a "proof-of-concept" baseline scenario
under simple configuration without storage or flexible demand, we could achieve an energy mix with 4.4% tolerance for any
technology's absolute share difference in each time step. For a climate policy scenario under a more realistic configuration with
storage and flexible demand, we could achieve 6-7% tolerance. The cost breakdown and prices of power generations for both
models are found to be very similar at the end of the iterative process, providing additional evidence that the quantity
harmonization follows the underlying principle of the price and cost harmonization. The remaining differences can be partially
explained by the lack of full harmonization of the brown-field and near-term capacity constraints, as well as potential
mismatches due to numerical techniques aimed at enhancing performance and stability. Using the coupling methodology, we
provide scenarios for power sector transition under a stringent German climate goal. Under this scenario, we observe a least-cost
pathway consisting of an almost complete transformation to a wind- and solar-based power system. The results indicate an
increasing role of storage and dispatchable capacity in a deep decarb scenario, consistent with the findings of previous PSM
studies, but which is now transferred to the long-term models via soft-coupling.
For future works, besides expanding the research program on sector coupling into a direction containing a broader technological
portfolio, we also aim to apply this framework to other world regions of interest in the REMIND model. Another important
aspect would be to represent the variability-smoothing effect of transmission grids by using the same coupling framework to
couple REMIND to other power sector models with more explicit modeling of transmission bottlenecks and expansion for two
or more regions.
**Appendices**
**Appendix A: Comparison of model scope and specification**

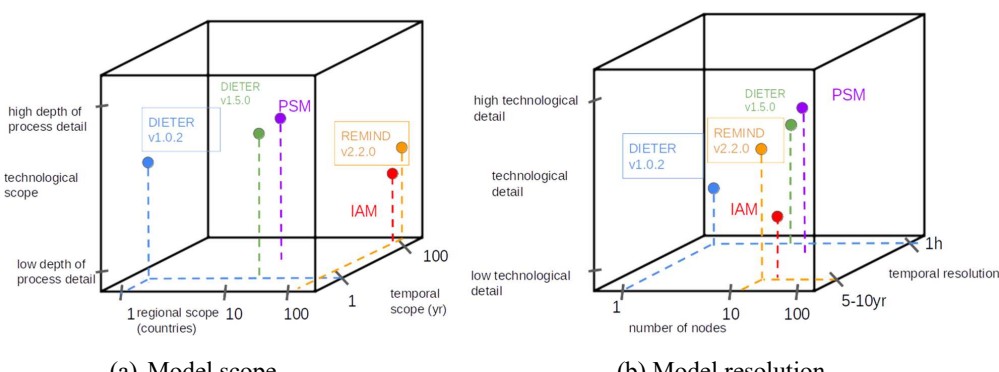

(a) Model scope                                    (b) Model resolution

**Figure A1: Comparison of resolution and scope for REMIND and a typical IAM, as well as two versions of DIETER**
**(v1.0.2 is used in this study) and a typical PSM.**



| Model name and version | REMIND v3.0.0 (dev) | DIETER v1.0.2 |
|---|---|---|
| model type | IAM | PSM |
| **Scope and resolution** | | |
| spatial scope | entire globe | single region (Germany) |
| intertemporal scope of "perfect foresight" | 2005-2100 (in actual model it is 2005-2150) | any year-long period |
| temporal resolution | 5- or 10-year time-step | hourly |
| regional resolution | single EU region | single EU region |
| sectoral scope | all energy sectors (transport, building, industry), industrial processes, air pollution, land-use sector, etc | power sector |
| available climate policy options | CO2 price, early-phase nuclear and coal phase out (for Germany), EU-ETS | CO2 price |
| **Power sector dynamics** | | |
| endogenous hourly dispatch | no | yes |
| differentiated market value for various technologies | no | yes |
| price-demand elasticity | yes | no |
| capital cost of technology | endogenous via learning curve (Leimbach et al., 2010) | exogenous |
| vintage tracking of existing capital stock | yes | no |
| transmission assumption | copper plate within region | copper plate within region |
| **Model code and data specification** | | |
| programming language | GAMS | GAMS |
| input data openness | partially open data | fully open data (for Germany) |
| source code openness | open | open |
| solver | CONOPT | CPLEX |

**Table A1: Comparison between the coupled models REMIND and DIETER.**

Because IAMs usually start out with certain assumptions for the development of macroeconomic metrics such as for GDP and population, which in turn determine the corresponding energy service levels to a larger degree prior to optimizing the energy system mix to meet demand, they are also frequently referred to as "top-down" energy system models. PSMs usually start out



modeling the fine spatiotemporal detail of the real-world power systems, expanding the capacity installation of power
generating plants, grid transmission and storage at minimum cost. Such models are also known as "unit commitment models"
for electrical power production (Padhy, 2004). Later in model development PSMs usually expand to include other energy
services such as heating and transportation which are electrified. In this way PSMs are also often referred to as "bottom-up"
models. Reviews and intercomparison of IAMs have been carried out recently where various IAMs are analyzed and
harmonized (Weyant, 2017; Butnar et al., 2019; Keppo et al., 2021; Wilson et al., 2021; Giarola et al., 2021).
For methodological reasons, we have to set the length of the model time horizon to be until 2150, which is longer than the valid
model time horizon until 2100. This is because without the extra years after 2100, the model has much less time to utilize the
capacities installed in the few decades before 2100, making it more difficult to justify the installation of new capacity
economically. This is manifested in a model artifact, where in the last few model periods investment in capacities decrease in
general. By extending the time horizon, this "boundary" effect is pushed further to the future, so the artifact only appears after
2100. Therefore the meaningful model results for REMIND are only between 2005-2100, even though years until 2150 are also
modeled and coupled.
Both models have open published source code. Partially thanks to the PSM community's advocacy of "open models", which
encompasses all steps from input data, model source code to numerical solvers (openmod - Open Energy Modelling Initiative,
2022), many research institutions also responded to their calls to openly publish their models. For example, the IAM used in this
study – REMIND, has for two years opened its source code on popular hosting site GitHub.
**Appendix B: Model coupling scope**
While REMIND and DIETER can both model a European-wide system with spatial subdivision (see Fig. B1 for REMIND
regional division), the soft-coupling currently is only applied to Germany. The coupling is from 2020 to 2150 for every defined
REMIND period. All common and available REMIND generating technologies are enabled for the coupling, as shown in Fig.
B2. The information for the species of technologies in REMIND are upscaled and coupled to DIETER, whereas information
from DIETER is then downscaled during the feedback loop that completes the coupled iteration.

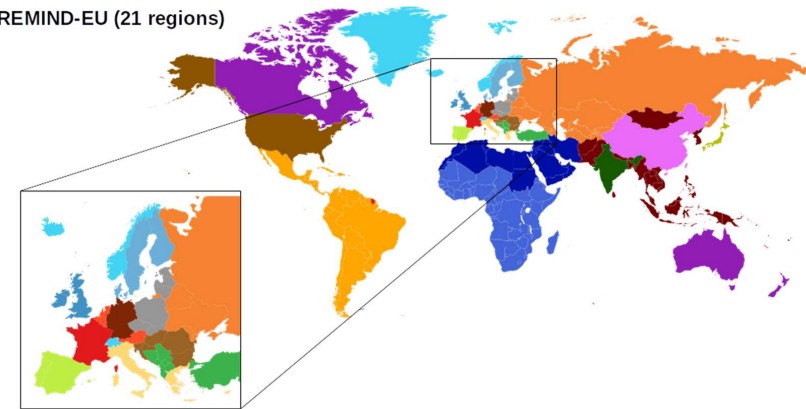


**Figure B1: REMIND regional resolution used in this study (21 global regions, including detailed differentiations of EU**
**regions). The spatial resolution of REMIND is flexible and depends on the resolution of the input data. Regional**
**mapping is from the REMIND-EU model (Rodrigues et al., 2022).**




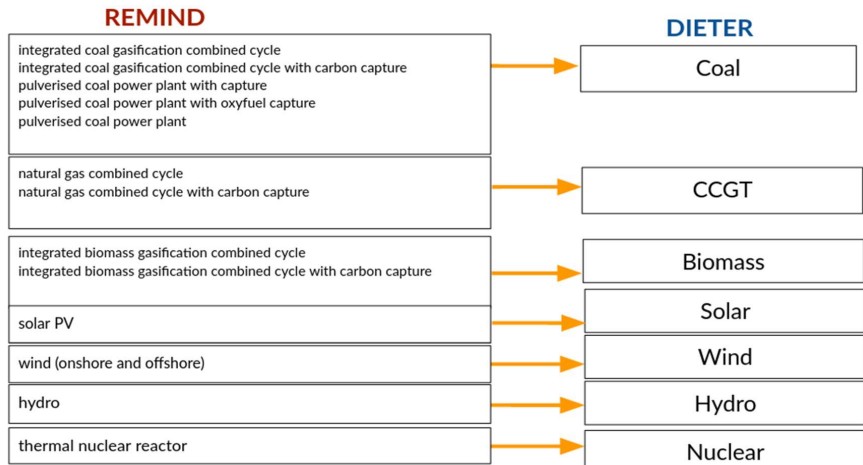


**Figure B2: Mapping of coupled technologies between REMIND and DIETER.**
**Appendix C: REMIND's interannual intertemporal objective function for single-region**
Single-region interannual intertemporal welfare is an aggregated utility, which in turn is a logarithm function of consumption. In
REMIND, the total welfare of a region is maximized and is equal to
$$W_{reg} = \sum_{y=2005}^{2150} \frac{1}{(1+\varrho_{reg})^{y-2005}} * \Delta y * V_{y,reg} * ln\left(\frac{\chi_{y,reg}}{\Gamma_{y,reg}}\right),$$

where regional consumption is $\chi_{y,reg}$ at model time $y$, and the weight of the consumption determined by the pure rate of time
preference $\varrho_{reg}$ and population $V_{y,reg}$. The consumption $\chi_{y,reg}$ at time $y$ is in turn equal to the difference between regional
income (gross domestic product) minus export (which is not available for consumption) and saving (i.e. investments), subtracted
by the cost of the energy system (including the power sector) and other costs in the economy. For simplicity we do not discuss
several other expenditures such as capital investment for energy service, other energy related expenditures such as R&D and
innovation, taxes, cost of pollution and land-use change.
**Appendix D: Deriving the soft-coupling convergence conditions**
In Sect. 3.2.1, we sketch the derivation procedure and offer a short summary of the analytical results. Here we describe the
derivation procedure of the coupled convergence framework in detail.
Using the Lagrangian multiplier method, based on the objective functions (Eqs. (1-2)) and constraints (c1-c6) in Sect. 3.1 we
can construct the KKT Lagrangians (Karush, 1939; Kuhn and Tucker, 1951; Gan et al., 2013):
REMIND:



$\mathcal{L} = \underbrace{\sum_{y,s}(c_{y,s}P_{y,s} + o_{y,s}G_{y,s})}_{\text{REMIND objective function}} + \underbrace{\sum_y \lambda_y \left[ d_y - \sum_s G_{y,s}(1 - \alpha_{y,s}) \right]}_{\text{annual electricity balance equation constraint}} + \underbrace{\sum_{y,s}\omega_{y,s}(P_{y,s} - \psi_s)}_{\text{resource constraint}} + \underbrace{\sum_{y,s}\xi_{y,s}(-G_{y,s})}_{\text{positive generation constraint}}$
$+ \underbrace{\sum_{y,s}\mu_{y,s}(G_{y,s} - 8760 * \phi_{y,s}P_{y,s})}_{\text{maximum generation from capacity constraint}} + \underbrace{\sum_{y\le2\ ,s}\sigma_{y,s}(p_{y,s} - P_{y,s})}_{\text{standing capacity constraint}} + \underbrace{\sum_{y=202\ ,s}\gamma_{y,s}(P_{y,s} - P_{y-\Delta y,s} - q_{y,s})}_{\text{near-term ramp-up capacity constraint}},\ \text{(D1)}$
DIETER:
$\underline{\mathcal{L}} = \underbrace{\sum_s \left[ \underline{c}_s \underline{P}_s + \underline{o}_s \sum_h (\underline{G}_{h,s} + \underline{\Gamma}_{h,vre}) \right]}_{\text{DIETER objective function}} + \underbrace{\sum_h \underline{\lambda}_h \left( \underline{d}_h - \sum_s \underline{G}_{h,s} \right)}_{\text{hourly electricity balance equation constraint}} + \underbrace{\sum_s \underline{\omega}_s \left( \underline{P}_s - \underline{\psi}_s \right)}_{\text{resource constraint}} + \underbrace{\sum_{h,s} \underline{\xi}_{h,s}(-\underline{G}_{h,s})}_{\text{positive generation constraint}}$
$+ \underbrace{\sum_{h,dis} \underline{\mu}_{h,dis}(\underline{G}_{h,dis} - \underline{P}_{dis})}_{\text{maximum dispatchable generation from capacity constraint}} + \underbrace{\sum_{h,vre} \underline{\mu}_{h,vre} \left( \underline{G}_{h,vre} + \underline{\Gamma}_{h,vre} - \underline{\phi}_{h,vre}\underline{P}_{vre} \right)}_{\text{maximum renewable generation from capacity and weather constraint}}.\ \text{(D2)}$
Comparing Lagrangians $\mathcal{L}$ and $\underline{\mathcal{L}}$, there are notable similarities between the terms. But first, we can reduce the complexity by
noticing that there are terms containing capacity shadow prices that are either trivial or already harmonized: resource constraint
shadow prices $\omega$ are already identical for both models by design (constraint (c2) in Sect. 3.1); positive generation constraint
shadow price $\xi$ is 0 due to KKT conditions for both models (constraint (c3)). These constraint terms can be safely excluded from
the subsequent mapping. We then note the important fact that REMIND Lagrangian is a sum over multiple years, whereas
DIETER Lagrangian is for each year. To make a direct comparison and therefore mapping possible, we assume that the brown-
field and near-term constraints are not binding. After this simplifying assumption, we realize that REMIND becomes linearly
independent in terms of the temporal slices, because by now the only yet-to-be-harmonized constraints left in the standalone
models are (c1) and (c4), which are both constraints for each year and do not result in temporal correlations. Note that this
simplifying assumption is assumed to be valid only for the derivation in this section. Later in actual simulations, we see that
these bounds generate shadow prices which are not necessarily small, impacting the degree of convergence especially in earlier
years. These constraints are also temporally localized in early periods, exerting little impact on later, more "green-field" years.
In fact, when including brown-field constraint into DIETER (c8), the model convergence is improved (Sec. 6.1).
After the aforementioned simplifications, we can construct a single-year REMIND Lagrangian $\mathcal{L}_y$:
$\mathcal{L}_y = \underbrace{\sum_s (c_{y,s}P_{y,s} + o_{y,s}G_{y,s})}_{\text{REMIND objective function}} + \underbrace{\lambda_y \left[ d_y - \sum_s G_{y,s}(1 - \alpha_{y,s}) \right]}_{\text{annual electricity balance equation constraint}} + \underbrace{\sum_s \mu_{y,s}(G_{y,s} - 8760 * \phi_{y,s}P_{y,s})}_{\text{maximum generation from capacity constraint}},\qquad \text{(D3)}$
and map it to the single-year DIETER Lagrangian $\underline{\mathcal{L}}$:
$\underline{\mathcal{L}} = \underbrace{\sum_s \left[ \underline{c}_s \underline{P}_s + \underline{o}_s \sum_h (\underline{G}_{h,s} + \underline{\Gamma}_{h,vre}) \right]}_{\text{DIETER objective function}} + \underbrace{\sum_h \underline{\lambda}_h \left( \underline{d}_h - \sum_s \underline{G}_{h,s} \right)}_{\text{hourly electricity balance equation constraint}} + \underbrace{\sum_{h,dis} \underline{\mu}_{h,dis}(\underline{G}_{h,dis} - \underline{P}_{dis})}_{\text{maximum dispatchable generation from capacity constraint}}$
$+ \underbrace{\sum_{h,vre} \underline{\mu}_{h,vre} \left( \underline{G}_{h,vre} + \underline{\Gamma}_{h,vre} - \underline{\phi}_{h,vre}\underline{P}_{vre} \right)}_{\text{maximum renewable generation from capacity and weather constraint}}.\qquad \text{(D4)}$
These are the same as Eqs. (3)-(4).
Comparing $\mathcal{L}_y$ and $\underline{\mathcal{L}}$, we can map them by matching the following four terms in the Lagrangians individually:
A) annual total power sector costs: $Z_y = \sum_s(c_{y,s}P_{y,s} + o_{y,s}G_{y,s})$ and $\underline{Z} = \sum_s \left[ \underline{c}_{y,s}\underline{P}_{y,s} + \underline{o}_{y,s}\sum_h(\underline{G}_{y,h,s} + \underline{\Gamma}_{y,h,vre}) \right]$,
B) annual revenue of usable (post-curtailment) generation for each generator $s$: $\lambda_y G_{y,s}(1 - \alpha_{y,s})$ and $\sum_h \underline{\lambda}_{y,h}\underline{G}_{y,h,s}$,
C) annual payment made by the consumers: $\lambda_y d_y$ and $\sum_h \underline{\lambda}_{y,h}\underline{d}_{y,h}$,



D)   maximum generation from capacity constraint term for each generator $s$: $\mu_{y,s}\left(G_{y,s} - 8760 * \phi_{y,s}P_{y,s}\right)$ and
$\sum_h \underline{\mu}_{y,h,s}\left(\underline{G}_{y,h,s} + \underline{\Gamma}_{y,h,s} - \underline{\phi}_{y,h,s}\underline{P}_{y,s}\right)$ (we write the two terms for VRE and dispatchable into one term for DIETER here
for simplicity, i.e. $\underline{\Gamma}_{y,h,dis} = 0$ and $\underline{\phi}_{y,h,dis} = 1$ for dispatchables).
The following conditions (h1-h7) can be derived from the harmonization of terms (A)-(D). Each term is harmonized by
matching the values in front of decision variables at the aggregated levels, namely capacities and annual generations.
Term A) can be mapped if:
**h1)** annual fixed costs are harmonized for each generator species $s$: $c_{y,s} = \underline{c}_{y,s}$ ,
**h2)** annual variable costs are harmonized for each generator species $s$: $o_{y,s} = \underline{o}_{y,s}$ .
Term B) can be mapped if:
**h3)** for each generator species $s$, the annual average revenue per unit generation, i.e. the market value, is harmonized by
exogenously manipulating the market value in REMIND to be the same as the last-iteration annual average market value in
DIETER. We achieve this by adding a correction term, thereby modifying REMIND original objective function $Z$ to $Z'$:
$Z' = Z - \sum_{y,s} \underline{\eta}_{y,s}(i-1)G_{y,s}\left(1 - \alpha_{y,s}\right),$
where $\underline{\eta}_{y,s}(i-1)$ is the markup for technology $s$ in DIETER in the last iteration $i-1$, $i$ is the index of the iteration of the
iterative soft-coupling. $Z'$ is the modified REMIND objective function in the coupled version.
The detailed derivation is as follows.
Lagrangian term B for the models have the physical meanings of total annual revenue of usable (post-curtailment)
generation. (Annual revenue is equal to the product of usable generation and annual market value.) We denote total annual
revenue from technology $s$ as $\Theta_{y,s}$ for REMIND and $\underline{\Theta}_{y,s}$ for DIETER. Then for REMIND the revenue (term B) is
$\Theta_{y,s} = \lambda_y G_{y,s}\left(1 - \alpha_{y,s}\right),$              (D5)
and for DIETER
$\underline{\Theta}_{y,s} = \sum_h \underline{\lambda}_{y,h}\underline{G}_{y,h,s}.$              (D6)
To harmonize terms $\Theta_{y,s}$ and $\underline{\Theta}_{y,s}$, our goal is to create a one-to-one mapping of the values in front of the decision variable
annual aggregated post-curtailment generation of technology $s$, which is $G_{y,s}\left(1 - \alpha_{y,s}\right)$ for REMIND and $\sum_h \underline{G}_{y,h,s}$ for
DIETER, the latter is namely a direct sum of the hourly generations. However, we notice for DIETER revenue $\underline{\Theta}_{y,s}$ is a
weighted sum of the hourly generation, and the direct sum cannot be separated in a straight-forward way. So first we have
to rewrite $\underline{\Theta}_{y,s}$(Eq. (D6)) by first dividing then multiplying by the aggregated annual generation:
$\underline{\Theta}_{y,s} = \dfrac{\sum_h \underline{\lambda}_{y,h}\underline{G}_{y,h,s}}{\sum_h \underline{G}_{y,h,s}} \sum_h \underline{G}_{y,h,s}.$              (D7)
We notice that the multiplicative term in front of the DIETER annual aggregated generation $\sum_h \underline{G}_{y,h,s}$ is $\dfrac{\sum_h \underline{\lambda}_{y,h}\underline{G}_{y,h,s}}{\sum_h \underline{G}_{y,h,s}}$, which
is nothing other than the market value of generation technology $s$ (see also Eq. (F24)).
We now take a look at revenue $\Theta_{y,s}$ on the REMIND side, which is equal to $\lambda_y G_{y,s}\left(1 - \alpha_{y,s}\right)$ (Eq. (D5)). To map (D5) to
the DIETER revenue term $\underline{\Theta}_{y,s}$ (Eq. (D7)) in terms of the aggregated decision variable $G_{y,s}\left(1 - \alpha_{y,s}\right)$ and $\sum_h \underline{G}_{y,h,s}$, we
essentially would like the multiplicative term in front of the generation variable in $\Theta_{y,s}$, which is $\lambda_y$, to be also $\dfrac{\sum_h \underline{\lambda}_{y,h}\underline{G}_{y,h,s}}{\sum_h \underline{G}_{y,h,s}}$
like in DIETER. This means the DIETER-corrected revenue in REMIND *should* be
$\Theta'_{y,s} = \dfrac{\sum_h \underline{\lambda}_{y,h}\underline{G}_{y,h,s}}{\sum_h \underline{G}_{y,h,s}} G_{y,s}\left(1 - \alpha_{y,s}\right).$              (D8)




To harmonize $\Theta_{y,s}$ and $\underline{\Theta}_{y,s}$, we can simply add a linear correction term to compensate for the difference between them.
Noticing in Eq. (D5), the multiplicative term in front of the REMIND generation variable $G_{y,s}(1 - \alpha_{y,s})$ is $\lambda_y$, which can
be interpreted as the REMIND market value, we realize essentially for a linear correction term, we should add the market
value difference $\Delta MV_{y,s}$ between the two models
$$\Delta MV_{y,s} = \underline{MV}_s - MV_s = \frac{\sum_h \underline{\lambda}_{y,h} \underline{G}_{y,h,s}}{\sum_h \underline{G}_{y,h,s}} - \lambda_y \;, \tag{D9}$$
to the multiplicative term $\lambda_y$ in $\Theta_{y,s}$, so $\lambda_y$ is canceled. Note that in Eq. (D9), as discussed before, the DIETER market
value is dependent on technology index $s$, whereas the REMIND one does not.
After adding the linear correction term, the modified revenue in REMIND $\Theta'_{y,s}$ after harmonization is:
$$\Theta'_{y,s} = \Theta_{y,s} + \Delta MV_{y,s} G_{y,s}(1 - \alpha_{y,s}) = (\Delta MV_{y,s} + \lambda_y) G_{y,s}(1 - \alpha_{y,s}), \tag{D10}$$
plugging in (D9),
$$\Theta'_{y,s} = \left( \frac{\sum_h \underline{\lambda}_{y,h} \underline{G}_{y,h,s}}{\sum_h \underline{G}_{y,h,s}} - \lambda_y + \lambda_y \right) G_{y,s}(1 - \alpha_{y,s}) = \frac{\sum_h \underline{\lambda}_{y,h} \underline{G}_{y,h,s}}{\sum_h \underline{G}_{y,h,s}} G_{y,s}(1 - \alpha_{y,s}), \tag{D11}$$
which is as desired in (D8).
In practice, in the case of annual shadow price $\lambda_y$ in REMIND, we find that the coupling behaves more stable numerically,
if we use the annual average electricity price of DIETER instead of the last-iteration electricity price of REMIND $\lambda_y$ in
(D9). The equivalence between the two prices is expressed later in (h5). We can use this substitution, since as we show
later that (h5) can be derived from market value harmonization (h3) and demand harmonization (h4). With this
substitution, the correction term which we call $\underline{\eta}_{y,s}$ is in fact:
$$\underline{\eta}_{y,s} = \underline{MV}_s - \underline{J} = \frac{\sum_h \underline{\lambda}_{y,h} \underline{G}_{y,h,s}}{\sum_h \underline{G}_{y,h,s}} - \frac{\sum_h \underline{\lambda}_{y,h} \underline{d}_{y,h}}{\sum_h \underline{d}_{y,h}}, \tag{D12}$$
where $\underline{J} = \frac{\sum_h \underline{\lambda}_{y,h} \underline{d}_{y,h}}{\sum_h \underline{d}_{y,h}}$ is the annual average electricity price in DIETER. We calculate (D12) using the last iteration
DIETER solutions. Note that compared to the earlier (D9), we have simply replaced the second term REMIND annual
price with DIETER annual price.
It is not hard to recognize $\underline{\eta}_{y,s}$ as the "markup" for technology $s$ in DIETER, where markup as defined before is the
difference between the market value of a technology $\underline{MV}_s$ and the load-weighted annual average electricity price $\underline{J}$ (see
Sect. 3.1 introduction).
Now we have concluded the derivation for the markup term $\underline{\eta}_{y,s}$ in (h3).
Although the multiplicative terms in front of decision variables in the two models can be harmonized via the correction
term (D12), we notice that it contains endogenous values, i.e. hourly generation $\underline{G}_{y,h,s}$ and hourly shadow price $\underline{\lambda}_{y,h}$ in
DIETER. Since any endogenous value can only be known ex post, this means the Lagrangian mapping relies on
endogenous values from the last iteration, i.e.
$$\underline{\eta}_{y,s}(i-1) = \underline{MV}_s(i-1) - \underline{J}(i-1) = \frac{\sum_h \underline{\lambda}_{y,h}(i-1)\underline{G}_{y,h,s}(i-1)}{\sum_h \underline{G}_{y,h,s}(i-1)} - \frac{\sum_h \underline{\lambda}_{y,h}(i-1)\underline{d}_{y,h}(i-1)}{\sum_h \underline{d}_{y,h}(i-1)}.$$
Now, using the markup term $\underline{\eta}_{y,s}$, we define the linear correction term for the revenue in REMIND $\Theta_{y,s}$ as
$$\Delta\Theta_{y,s} = \underline{\eta}_{y,s}(i-1) G_{y,s}(1 - \alpha_{y,s}) .$$
The physical meaning of $\Delta\Theta_{y,s}$ is the revenue difference in the two models for technology $s$, given that the post-
curtailment generations are expressed in terms of REMIND variables.



The coupled REMIND has a modified objective function $Z'$ based on a linear correction. The correction term $\Delta\theta_{y,s}$ need to
be summed over $s$ and $y$ and subtracted – due to the negative sign in front of term B, from the REMIND objective function
$Z$, since the objective term as a part of the Lagrangian can be directly manipulated:
$$Z' = Z - M = Z - \sum_{y,s} \Delta\theta_{y,s} = Z - \sum_{y,s} \underline{\eta}_{y,s}(i-1)G_{y,s}(1-\alpha_{y,s}),$$
where we call the total system revenue differences $M$, again, these are revenues where the post-curtailment generations are
expressed in terms of REMIND variables (and not DIETER variables).
Now we have concluded the derivation for the convergence condition (h3).
Depending on the starting point of the REMIND power system, and due to the internal iterative changes of REMIND
results due to the adjustments in trade between regions during the "Nash" algorithm, coupled convergence usually can only
be achieved over multiple iterations. Therefore the derived markup equation (Eq. (D12)) in general can be only expected to
reflect the actual market value differences approximately in the two models. This is the reason that in the iterative
algorithm after the first iteration, we add $M(i) - M(i-1)$ to the objective function $Z$, as the quantities and prices
gradually converge between the two models. As convergence is approached, the total revenue difference between iteration
$M(i) - M(i-1)$ should go to zero. This is confirmed by the numerical experiments (not shown).
Term C) can be mapped if:
**h4)** annual power demand in the two models are harmonized: $d_y = \sum_h \underline{d}_{y,h}$  ,
**h5)** annual average price of electricity is mapped to each other $\lambda_y = \frac{\sum_h \underline{\lambda}_{y,h}\underline{d}_{y,h}}{\sum_h \underline{d}_{y,h}}$ (dividing term (C) by (h4)). Because
electricity price is by definition equal to total annual system revenue divided by total annual demand, (h5) can be shown to
hold true, given technology-specific revenues are harmonized in (h3) and demand are harmonized in (h4). (If technology-
specific revenues are harmonized in (h3), then the system revenues which are technology-specific revenues summed over
technologies are also harmonized.) (h5) therefore can be seen as a derived condition from (h3) and (h4).
Term D) can be mapped if:
**h6)** annual average capacity factors are harmonized, i.e. $\phi_{y,s}$ in REMIND is set to equal to the endogenous last-iteration
DIETER result for each generation type $s$:
$$\phi_{y,s} = \sum_h \underline{\phi}_{y,h,s}/8760 ,$$
where $\underline{\phi}_{y,h,s} = \frac{\underline{G}_{y,h,s}}{\underline{P}_{y,s}}$ is the hourly capacity factor in DIETER. Without explicit manipulation of the shadow prices $\mu_{y,s}$ and
$\underline{\mu}_{y,h,s}$, we show the following claim is true, i.e. by above capacity factor harmonization, the terms containing endogenous
shadow prices will be automatically mapped. Showing this requires careful mathematical argument, which we make in
detail in the case of dispatchable, and later argue the case is similar for renewable.
For dispatchable generators the argument is as follows. (For simplicity we use the generic index $s$.)
We first rewrite REMIND term D by plugging in the harmonization condition $\phi_{y,s} = \sum_h \underline{\phi}_{y,h,s}/8760$:
$$\mu_{y,s}\left(G_{y,s} - 8760 * \phi_{y,s}P_{y,s}\right) = \sum_y \mu_{y,s}\left(G_{y,s} - \sum_h \underline{\phi}_{y,h,s}P_{y,s}\right),$$
and it should be mapped to the term $\sum_{y,h} \underline{\mu}_{y,h,s}\left(\underline{G}_{y,h,s} - \underline{P}_{y,s}\right)$ in DIETER.
Splitting the two terms, these four terms need to be harmonized:
$\mu_{y,s}G_{y,s}$    and    $\sum_h \underline{\mu}_{y,h,s}\underline{G}_{y,h,s}$                                                    (D13)



$\mu_{y,s} \sum_h \underline{\phi}_{y,h,s} P_{y,s}$   and    $\sum_h \underline{\mu}_{y,h,s} \underline{P}_{y,s}$          (D14)
for all $y, s$.
To show the mappings (D13)-(D14) are automatically satisfied given (h6), we first consider two simplified power sector
toy problems, Q1 and Q2, with only dispatchable technologies. Both problems have identical objective functions $\tilde{Z} =$
$\sum_s (\tilde{c}_s \tilde{P}_s + \tilde{o}_s \tilde{G}_s)$, and the fixed and variable cost parameters $\tilde{c}_s$ and $\tilde{o}_s$ are identical. Both problems have identical hourly
balance equation constraint, but with two different kinds of maximum generation constraint, Q1 has an inequality
constraint for each hour, Q2 has an aggregated annual equality constraint:
Q1: $min\ Z$, s.t. $\tilde{G}_{h,s} \le \tilde{P}_s$   $\perp \tilde{\mu}_{h,s}, \tilde{d}_h = \sum_s \tilde{G}_{h,s}$   $\perp \tilde{\lambda}_h$
Q2: $min\ Z$, s.t. $\sum_h \tilde{G}_{h,s} = 8760 * \tilde{\phi}_s \tilde{P}_s$   $\perp \tilde{\mu}'_s$ , $\tilde{d}_h = \sum_s \tilde{G}_{h,s}$   $\perp \tilde{\lambda}'_h$
Then the Lagrangians are:
$$\tilde{\mathcal{L}}_1 = \underbrace{\sum_s \left(\tilde{c}_s \tilde{P}_s + \tilde{o}_s \sum_h \tilde{G}_{h,s}\right)}_{\text{objective function}} + \underbrace{\sum_h \tilde{\lambda}_h \left(\tilde{d}_h - \sum_s \tilde{G}_{h,s}\right)}_{\text{hourly electricity balance equation constraint}} + \underbrace{\sum_{h,s} \tilde{\mu}_{h,s} \left(\tilde{G}_{h,s} - \tilde{P}_s\right)}_{\text{maximum generation from capacity constraint}}$$
$$\tilde{\mathcal{L}}_2 = \underbrace{\sum_s \left(\tilde{c}_s \tilde{P}_s + \tilde{o}_s \sum_h \tilde{G}_{h,s}\right)}_{\text{objective function}} + \underbrace{\sum_h \tilde{\lambda}'_h \left(\tilde{d}_h - \sum_s \tilde{G}_{h,s}\right)}_{\text{hourly electricity balance equation constraint}} + \underbrace{\sum_s \tilde{\mu}'_s \left(\sum_h \tilde{G}_{h,s} - 8760\tilde{\phi}_s \tilde{P}_s\right)}_{\text{maximum generation from capacity constraint}} \quad .$$
The relevant KKT conditions:
Stationarity condition for Q1:
$\frac{\partial \tilde{\mathcal{L}}_1}{\partial \tilde{P}_s} = \tilde{c}_s - \sum_h \tilde{\mu}_{h,s} = 0$          (D15)
Stationarity condition for Q2:
$\frac{\partial \tilde{\mathcal{L}}_2}{\partial \tilde{P}_s} = \tilde{c}_s - 8760\tilde{\phi}_s \tilde{\mu}'_s = 0$          (D16)
Since the fixed cost $\tilde{c}_s$ are equal for the two models, from Eqs. (D15)-(D16) we can derive the relation between the two
shadow prices:
$8760 * \tilde{\phi}_s \tilde{\mu}'_s = \sum_h \tilde{\mu}_{h,s}$ .          (D17)
Note that for the toy models, the identical balance equation constraints do not contain capacity $P$, which is why the balance
equation constraints do not influence the stationary conditions for $P$ (Eqs. (D15)-(D16)).
We now show (D14) is automatically mapped given capacity factor harmonization (h6). We first write the equality
condition for the REMIND-DIETER case, analogous to the toy model result (D17),
$8760 * \phi_{y,s} \mu_{y,s} = \sum_h \underline{\mu}_{y,h,s}$    .          (D18)
Note that we can apply the toy model case to the REMIND-DIETER coupling case in rather straight-forward way, because
in the case of REMIND-DIETER, the objective function terms have been already harmonized by (h1)-(h2), and the balance
equation constraint terms do not contain $P$, so they have no bearing on the generation-capacity constraint term, just like in
the case of the toy models.
Plugging (h6) $\phi_{y,s} = \sum_h \underline{\phi}_{y,h,s}(i-1)/8760$ into (D18), we have derived the equality for the parameter mapping required
in (D14), i.e.,
$\mu_{y,s} \sum_h \underline{\phi}_{y,h,s}(i-1) = \sum_h \underline{\mu}_{y,h,s}$.
To show (D13), we first use hourly capacity factor from DIETER,
$\underline{G}_{y,h,s} = \underline{\phi}_{y,h,s} \underline{P}_{y,s}$ ,          (D19)

as well as the primal feasibility condition from REMIND $G_{y,s} = 8760 * \phi_{y,s} P_{y,s}$ (Eq. (F9)), to rewrite both sides of the mapping in (D13) in capacity terms. For REMIND, plugging in (F9),

$$\mu_{y,s} G_{y,s} = \mu_{y,s} * 8760 * \phi_{y,s} P_{y,s} \,, \tag{D20}$$

and for DIETER, plugging in (D19),

$$\sum_h \underline{\mu}_{y,h,s} \underline{G}_{y,h,s} = \sum_h \underline{\mu}_{y,h,s} \underline{\phi}_{y,h,s} \underline{P}_{y,s} \,. \tag{D21}$$

Take the complementary slackness condition of DIETER $\underline{\mu}_{h,s}\,(\underline{G}_{h,s} - \underline{P}_s) = 0$ (Eq. (F16)), insert (D19) on the left-hand-side, we obtain

$$\underline{\mu}_{h,s}\,(\underline{G}_{h,s} - \underline{P}_s) = \underline{\mu}_{h,s}(\underline{\phi}_{y,h,s} \underline{P}_{y,s} - \underline{P}_s) = 0 \,.$$

Rearranging, we get

$$\underline{\mu}_{y,h,s} \underline{\phi}_{y,h,s} \underline{P}_s = \underline{\mu}_{y,h,s} \underline{P}_s \,, \tag{D22}$$

for each hour $h$.

Plug (D22) and then (D18) into the right-hand-side of (D21), to obtain

$$\sum_h \underline{\mu}_{y,h,s} \underline{G}_{y,h,s} = \sum_h \underline{\mu}_{y,h,s} \underline{P}_{y,s} = 8760 * \phi_{y,s} \mu_{y,s} \underline{P}_{y,s} \,. \tag{D23}$$

Compare (D20) with (D23), they now have identical parameters in front of the capacity variable $P_{y,s}$ and $\underline{P}_{y,s}$, as desired. We concluded the proof that by exogenously setting the annual capacity factor of REMIND to that of the last iteration DIETER, we automatically harmonize the generation-capacity constraint term of the Lagrangian, in the case of dispatchable generators.

**h7)** for VRE, annual curtailment rates are harmonized $G_{y,vre} \alpha_{y,vre} = \sum_h \underline{\Gamma}_{y,h,vre}$ , i.e. by exogenously setting curtailment rate in REMIND $\alpha_{y,vre} = \sum_h \underline{\Gamma}_{y,h,vre}(i-1) \;/ G_{y,vre}$, taking the endogenously determined curtailed power $\underline{\Gamma}_{y,h,vre}$ from the last iteration DIETER. This in general also harmonizes terms other than term D, as it harmonizes the definition for generation variable in DIETER which is post-curtailment and REMIND definition for generation variable which is pre-curtailment. For VREs the derivation is conceptually similar to the above case for dispatchable in (h6), since we can define a real capacity factor (post-curtailment) similar to the capacity factor for the dispatchable generators above,

$$\underline{\phi}_{y,h,vre} = \underline{G}_{h,vre} / \underline{P}_{vre} \,.$$

Due to the limitations of this paper, we will not present the derivation here. A detailed derivation is available upon request.

## Appendix E: Coupling iteration schematics

Coupled region: Germany

Coupled REMIND time horizon: 2020-2150 (2010-2015 are not coupled since they are historical years and have mostly hard-fixed quantities)



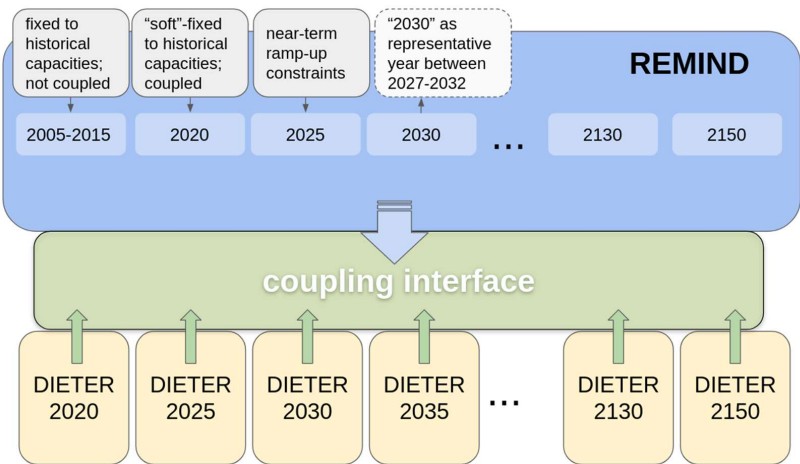


**(a) Graphic illustration of the bi-directional coupling in the temporal dimension.**

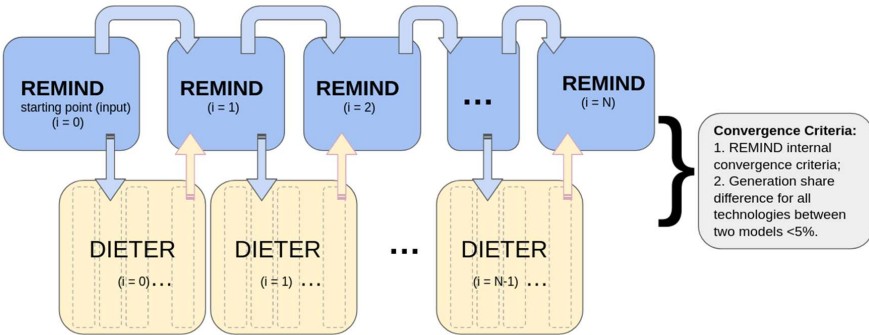


**(b) Graphic illustration of the bi-directional coupling in the iteration dimension.**
**Figure E1: A graphic description of the model iterative coupling. (a) The temporal slices of REMIND which are mapped**
**to multiple parallel year-long DIETER problems are illustrated here. The convergence conditions are iteratively mapped**
**at the interface. (b) Every i-th iteration of REMIND takes the (i-1)-th iteration of REMIND as a starting point for**
**optimization, and the endogenous output of the (i-1)-th DIETER as exogenous input parameters. When the convergence**
**conditions are met, i.e. REMIND satisfies its internal convergence condition, and the coupled models differ in their**
**generation share of each technology at most by a certain percentage (e.g. 5% for baseline run without storage), the**
**coupled run halts.**

Under simple configuration (no storage, no flexible demand), every REMIND run takes around 3 minutes and DIETER run
takes a few seconds to solve. Under more detailed configurations (with storage and flexible demand) and climate policies, every
REMIND run takes around 4 minutes and a DIETER run takes a few minutes to solve. The entire REMIND-DIETER coupled
run for a single region Germany under simple configuration is around 3~4 hours. It is around 6~10 hours for the more detailed
configurations under climate policies.





**Appendix F: Derivation of the equilibrium conditions for uncoupled REMIND and DIETER**

In this appendix, we discuss the equilibrium conditions of the uncoupled models, resulting in a rigorous formulation of the so-called "zero-profit rules" (ZPRs). We first construct the Lagrangians and compute KKT conditions, then derive the ZPRs for the standalone versions of REMIND reduced power-sector model and DIETER model.

Using the objective functions and constraints in Sect. 3.1, we can construct Lagrangians for the two standalone models. Using the KKT conditions derived from the Lagrangians, we can show that if the historical and resource constraint are non-binding, i.e. shadow prices $\omega$, $\sigma$ and $\gamma$ are zero, then each generator would have recovered their fixed, variable cost and curtailment cost through their total market revenue, i.e. each producer of electricity gets "zero profit", given that the profits are defined as the difference between revenue and cost. When the capacity constraints exist and are binding, we arrive at a modified version of the original ZPR, which describes the relation between cost, revenue and the capacity shadow prices.

Here we first construct the Lagrangians and derive the KKT conditions from them (Sect. F.1) for both models. Then both models' ZPRs are derived, two for each model, namely, the technology-specific ZPR and the system ZPR (Sect. F.2).

**F.1 Lagrangians and KKT conditions**

The Lagrangians of the uncoupled model have been constructed in Appendix D (Eqs. (D1)-(D2)). From the KKT conditions for minimization, we can ascertain the following first-order conditions at stationarity for each model.

For REMIND,

    1) Stationary conditions:

$$\frac{\partial \mathcal{L}}{\partial P_{y,s}} = 0 \Rightarrow c_{y,s} + \omega_{y,s} - 8760 * \mu_{y,s}\phi_{y,s} - \sigma_{y,s} + \gamma_{y,s} = 0 \ , \tag{F1}$$

$$\frac{\partial \mathcal{L}}{\partial G_{y,s}} = 0 \Rightarrow o_{y,s} - \lambda_y\big(1 - \alpha_{y,s}\big) - \xi_{y,s} + \mu_{y,s} = 0 \ . \tag{F2}$$

    2) Complementary slackness:

$$\omega_{y,s}\big(P_{y,s} - \psi_s\big) = 0, \tag{F3}$$

$$\xi_{y,s}G_{y,s} = 0 \ , \tag{F4}$$

$$\mu_{y,s}(G_{y,s} - 8760 * \phi_{y,s}P_{y,s}) = 0, \tag{F5}$$

$$\sigma_{y,s}\big(p_{y,s} - P_{y,s}\big) = 0, \qquad (y \leq 2020) \ , \tag{F6}$$

$$\gamma_{y,s}\big(P_{y,s} - P_{y-\Delta y,s} - q_{y,s}\big) = 0, \qquad (y = 2025) \ . \tag{F7}$$

    3) Primal feasibility:

$$d_y - \sum_s G_{y,s}\big(1 - \alpha_{y,s}\big) = 0 \ , \tag{F8}$$

$$G_{y,s} - 8760 * \phi_{y,s}P_{y,s} = 0, \tag{F9}$$

    4) Dual feasibility:

$$\xi_{y,s} \geq 0, \omega_{y,s} \geq 0, \sigma_{y,s} \geq 0 \ , \gamma_{y,s} \geq 0. \tag{F10}$$

For DIETER,

    1) Stationary conditions:

$$\frac{\partial \mathcal{L}}{\partial \underline{P}_s} = 0 \Rightarrow \underline{c}_s + \underline{\omega}_s - \sum_h \underline{\phi}_{h,s}\underline{\mu}_{h,s} = 0 \ , \ \underline{\phi}_{h,s} = 1 \text{ for dispatchables}, 0 < \underline{\phi}_{h,s} < 1 \text{ for renewables} \tag{F11}$$

$$\frac{\partial \mathcal{L}}{\partial \underline{G}_{h,s}} = 0 \Rightarrow \underline{o}_s - \underline{\lambda}_h - \underline{\xi}_{h,s} + \underline{\mu}_{h,s} = 0 \ , \tag{F12}$$

$$\frac{\partial \mathcal{L}}{\partial \underline{\Gamma}_{h,vre}} = 0 \Rightarrow \underline{o}_{vre} + \underline{\mu}_{h,vre} = 0 \ . \tag{F13}$$



2) Complementary slackness:

$$\underline{\omega}_s \left( \underline{P}_s - \underline{\psi}_s \right) = 0,$$ (F14)

$$\underline{\xi}_{h,s} \, \underline{G}_{h,s} = 0,$$ (F15)

$$\underline{\mu}_{h,dis} \left( \underline{G}_{h,dis} - \underline{P}_{dis} \right) = 0,$$ (F16)

3) Primal feasibility:

$$\underline{d}_h = \sum_s \underline{G}_{h,s},$$ (F17)

$$\underline{G}_{h,vre} + \underline{\Gamma}_{h,vre} = \underline{\phi}_{h,vre} \underline{P}_{vre} \,,$$ (F18)

4) Dual feasibility:

$$\underline{\omega}_s \geq 0, \; \underline{\xi}_{h,s} \geq 0, \; \underline{\mu}_{h,dis} \geq 0 \,.$$ (F19)

**F.2 Derivation of the zero-profit rules**

**F.2.1 REMIND**

The derivation of ZPRs is very similar to the one in Brown and Reichenberg, 2021. Starting with the total costs for technology $s$ for all years, and applying various KKT conditions (after " | "),

$$\sum_y \left( c_{y,s} P_{y,s} + o_{y,s} G_{y,s} \right)$$

$$= \sum_y \left\{ \left( -\omega_{y,s} + 8760 * \mu_{y,s} \phi_{y,s} + \sigma_{y,s} - \gamma_{y,s} \right) P_{y,s} + \left[ \lambda_y (1 - \alpha_{y,s}) + \xi_{y,s} - \mu_{y,s} \right] G_{y,s} \right\} \quad | \text{ (F1), (F2)}$$

$$= \sum_y \left\{ \left( -\omega_{y,s} + 8760 * \mu_{y,s} \phi_{y,s} + \sigma_{y,s} - \gamma_{y,s} \right) P_{y,s} + \left[ \lambda_y (1 - \alpha_{y,s}) - \mu_{y,s} \right] G_{y,s} \right\} \quad | \text{ (F4)}$$

$$= \sum_y \left\{ \left( -\omega_{y,s} + \sigma_{y,s} - \gamma_{y,s} \right) P_{y,s} + \lambda_y G_{y,s} (1 - \alpha_{y,s}) \right\} \quad | \text{ (F5)}$$

Rearranging, we arrive at the ZPR of multi-year uncoupled REMIND for technology cost-revenue balance:

$$\underbrace{\sum_y \left( c_{y,s} P_{y,s} + o_{y,s} G_{y,s} \right)}_{\text{Generation costs}_s} = - \underbrace{\sum_y \left( \omega_{y,s} - \sigma_{y,s} + \gamma_{y,s} \right) P_{y,s}}_{\text{Capacity shadow revenues}_s} + \underbrace{\sum_y \lambda_y G_{y,s} (1 - \alpha_{y,s})}_{\text{Generation revenues}_s}.$$ (F20)

Normally, when there are no capacity shadow prices, or when the capacity constraints are not binding, the cost exactly equals revenue. However, when capacity shadow prices are non-zero, i.e. the constraints (c2) and (c5-c6) are binding, the capacity shadow prices act as a distortion to the equality relation between costs and revenues. As an example, the shadow price $\omega_{y,s}$ from limited generation resources (e.g. hydroelectric power in Germany) would be positive $\omega_{y,s} > 0$, when the constraint is binding, and would appear as a "positive cost", or a "negative revenue" in the modeled power market. We can therefore put it either on the left (cost) or right (revenue) side of the equation. Here we group it together with revenues.

One observes that from the right-hand-side of Eq. (F20), there is no differentiation between the annual market values of variable and dispatchable generations such as gas and solar – they are both equal to the annual electricity price $\lambda_y$.

From Eq. (F20), we can derive a ZPR between levelized cost of electricity (LCOE), capacity shadow price and market value (MV), for each generator type. Taking Eq. (F20), we separate the pre-curtailment LCOE from the LCOE due to curtailment, then divide by total post-curtailment generation $\sum_y G_{y,s} (1 - \alpha_{y,s})$ for the generator type $s$, to obtain the technology-specific ZPR:

$$\underbrace{\frac{\sum_y \left( c_{y,s} P_{y,s} + o_{y,s} G_{y,s} \right)}{\sum_y G_{y,s}}}_{\text{Pre-curtailment LCOE}_s} + \underbrace{\frac{\sum_y \left( c_{y,s} P_{y,s} + o_{y,s} G_{y,s} \right) \alpha_{y,s}}{\sum_y G_{y,s} (1 - \alpha_{y,s})}}_{\text{Curtailment LCOE}_s} = - \underbrace{\frac{\sum_y \left( \omega_{y,s} - \sigma_{y,s} + \gamma_{y,s} \right) P_{y,s}}{\sum_y G_{y,s} (1 - \alpha_{y,s})}}_{\text{Capacity shadow prices}_s} + \underbrace{\frac{\sum_y \lambda_y G_{y,s} (1 - \alpha_{y,s})}{\sum_y G_{y,s} (1 - \alpha_{y,s})}}_{\text{Market Values}_s}.$$ (F21)



The pre-curtailment LCOE is the cost of one unit of generated electricity – regardless whether it is curtailed or being used to
meet demand, whereas the curtailment LCOE is the cost of one unit of curtailed electricity. Together they add up to post-
curtailment LCOE, i.e. the cost of one unit of usable electricity.
To obtain the ZPR for the whole power system in REMIND, we first sum Eq. (F20) over all generator types $s$, and obtain the
ZPR for system cost and revenue. Then dividing by total post-curtailment system generation, and split the LCOE into pre-
curtailment and curtailment components, we get

$$\underbrace{\frac{\sum_{y,s}(c_{y,s}P_{y,s} + o_{y,s}G_{y,s})}{\sum_{y,s}G_{y,s}}}_{\text{Pre-curtailment LCOE}_{\text{system}}} + \underbrace{\frac{\sum_{y,s}(c_{y,s}P_{y,s} + o_{y,s}G_{y,s})\alpha_{y,s}}{\sum_{y,s}G_{y,s}(1 - \alpha_{y,s})}}_{\text{Curtailment LCOE}_{\text{system}}} = -\underbrace{\frac{\sum_{y,s}(\omega_{y,s} - \sigma_{y,s} + \gamma_{y,s})P_{y,s}}{\sum_{y,s}G_{y,s}(1 - \alpha_{y,s})}}_{\text{Capacity shadow price}_{\text{system}}} + \underbrace{\frac{\sum_{y,s}\lambda_y G_{y,s}(1 - \alpha_{y,s})}{\sum_{y,s}G_{y,s}(1 - \alpha_{y,s})}}_{\text{Electricity Price}_{\text{system}}},$$
   (F22)

i.e. the LCOE of the system for usable (pre-curtailment) power, which is equal to the sum of the system LCOE for total power
generated and the curtailment cost, can be recovered by the average electricity price of the system minus system-wide capacity
constraint shadow price per energy unit.
The ZPRs of REMIND hold for the aggregate over multiple years.
From Eqs. (F21)-(F22), we learn that when a market equilibrium can be found, i.e. when the optimization problem can be
successfully solved, there is an equality relation between the generation cost and market value for each generator type, and
similarly between generation cost and price of electricity for the entire system. Capacity shadow prices due to various extra
capacity constraints imposed on the models, distort the equality relation between costs and prices by a linear term, making the
prices be either higher or lower than the costs at the market equilibrium.

### F.2.2 DIETER

Similar to uncoupled REMIND, from KKT conditions, at stationarity, we can obtain the cost-revenue ZPR for a single
technology $s$ for standalone DIETER. We take the total costs for technology $s$ for all years, and applying various KKT
conditions (after " | "),
$\underline{c_s}\underline{P_s} + \sum_h \left[\underline{o_s}\left(\underline{G_{h,s}} + \underline{\Gamma_{h,vre}}\right)\right]$
$= \left(-\underline{\omega_s} + \Sigma_h \underline{\phi}_{h,s}\underline{\mu}_{h,s}\right)\underline{P_s} + \Sigma_h \left(\underline{\lambda_h} - \underline{\mu}_{h,s} + \underline{\xi}_{h,s}\right)\left(\underline{G_{h,s}} + \underline{\Gamma_{h,vre}}\right)$      | (F11),(F12)
$= -\underline{\omega_s}\,\underline{P_s} + \Sigma_h \underline{\phi}_{h,vre}\underline{\mu}_{h,vre}\underline{P}_{pre} + \Sigma_h \left(\underline{\lambda_h} - \underline{\mu}_{h,vre} + \underline{\xi}_{h,vre}\right)\left(\underline{G}_{h,vre} + \underline{\Gamma}_{h,vre}\right) + \Sigma_h \underline{\mu}_{h,dis}\underline{P}_{dis} + \Sigma_h \left(\underline{\lambda_h} - \underline{\mu}_{h,dis}\right)\underline{G}_{h,dis}$
| split $\Sigma_h \underline{\phi}_{h,s}\underline{\mu}_{h,s}$ into $vre$ and $dis$, apply (F15) for dispatchable, i.e. $\underline{\xi}_{h,dis}\,\underline{G}_{h,dis} = 0$
$= -\underline{\omega_s}\,\underline{P_s} + \Sigma_h \underline{\phi}_{h,vre}\underline{\mu}_{h,vre}\underline{P}_{pre} + \Sigma_h \left(\underline{\lambda_h} - \underline{\mu}_{h,vre} + \underline{\xi}_{h,vre}\right)\left(\underline{G}_{h,vre} + \underline{\Gamma}_{h,vre}\right) + \Sigma_h \underline{\lambda_h}\underline{G}_{h,dis}$      | (F16)
$= -\underline{\omega_s}\,\underline{P_s} + \Sigma_h \underline{\lambda_h}\underline{G}_{h,vre} + \Sigma_h \left(\underline{\lambda_h} + \underline{\xi}_{h,vre}\right)\underline{\Gamma}_{h,vre} + \Sigma_h \underline{\lambda_h}\underline{G}_{h,dis}$      | (F18), apply (F15) for VRE, i.e. $\underline{\xi}_{h,vre}\underline{G}_{h,vre} = 0$
$= -\underline{\omega_s}\,\underline{P_s} + \Sigma_h \underline{\lambda_h}\underline{G}_{h,vre} + \Sigma_h \underline{\lambda_h}\underline{G}_{h,dis}$      | (F12) & (F13) $\Rightarrow \underline{\lambda_h} + \underline{\xi}_{h,vre} = 0$
Rearranging, we arrive at the ZPR of single-year uncoupled DIETER for technology-specific cost-revenue balance:

$$\underbrace{\underline{c_s}\underline{P_s} + \underline{o_s}\sum_h\left(\underline{G_{h,s}} + \underline{\Gamma}_{h,vre}\right)}_{\text{Annual generation cost}_s} = -\underbrace{\underline{\omega_s}\underline{P_s}}_{\text{Annual capacity shadow revenue}_s} + \underbrace{\sum_h \underline{\lambda_h}\underline{G}_{h,s}}_{\text{Annual generation revenue}_s}.$$
   (F23)

Dividing Eq. (F23) by annual aggregated generation of technology $s$, we obtain the technology-specific ZPR for DIETER,

$$\underbrace{\frac{\underline{c_s}\underline{P_s} + \underline{o_s}\sum_h\left(\underline{G_{h,s}} + \underline{\Gamma}_{h,vre}\right)}{\sum_h \underline{G}_{h,s}}}_{\text{LCOE}_s} = -\underbrace{\frac{\underline{\omega_s}\underline{P_s}}{\sum_h \underline{G}_{h,s}}}_{\text{Annual capacity shadow price}_s} + \underbrace{\frac{\sum_h \underline{\lambda_h}\underline{G}_{h,s}}{\sum_h \underline{G}_{h,s}}}_{\text{Market value}_s}.$$
   (F24)

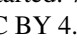


One observes that from the term of $\underline{Market\ Value}_s$, compared to the REMIND case (right-hand-side of Eq. (F21)), DIETER
has differentiated annual market values of gas and solar generators.
Summing Eq. (F24) over $s$, dividing both sides by total annual generation $\sum_{h,s} \underline{G}_{h,s}$, using identity $\underline{d}_h = \sum_s \underline{G}_{h,s}$ for
simplification, we obtain the ZPR for the whole power system in DIETER,
$$\underbrace{\frac{\sum_s[\underline{c}_s P_s + \underline{o}_s \sum_h(\underline{G}_{h,s} + \underline{\Gamma}_{h,vre})]}{\sum_{h,s} \underline{G}_{h,s}}}_{LCOE_{system}} = - \underbrace{\frac{\sum_s \omega_s P_s}{\sum_{h,s} \underline{G}_{h,s}}}_{\text{Annual capacity shadow price}_{system}} + \underbrace{\frac{\sum_h \lambda_h d_h}{\sum_h \underline{d}_h}}_{\text{Annual average electricity price}_{system}} . \qquad (F25)$$
Similar to the case of REMIND, Eqs. (F24)-(F25) show us the equality relations between cost and value (or price) for each
generator type and for the system hold also for DIETER at its market equilibrium. Compared to REMIND, there are no brown-
field or near-term capacity shadow price contributions in DIETER in the standalone versions. The DIETER ZPRs hold for one
year instead of the aggregate of multiple years like in REMIND. For simplicity, even though it is possible to write the LCOE in
pre-curtailment and curtailment terms, but because for DIETER it is relatively cumbersome to do, we do not do it here.
In summary, at REMIND and DIETER power market equilibriums, each generator exactly recovers its cost of one unit of
generation through market value, and obtains "zero profit" under a completely competitive market over its modeling time. In the
aggregate, the entire power sector obtains its cost of one unit of generation through the price of electricity that the consumer
pays. Both types of relations can be distorted by the existence of capacity shadow prices.
**Appendix G: Derivation of the equilibrium conditions for the coupled models**
Here in this Appendix, we gradually build up the derivation for the ZPRs of the coupled REMIND and DIETER, which will be
used later for validating numerical results. The derivation consists of three steps:
1) ZPRs for the uncoupled model REMIND and DIETER;
2) ZPRs for coupled model REMIND and DIETER (simplified version, only considering convergence condition (h1-h7));
3) ZPRs for coupled model REMIND and DIETER (full version, also considering (c7 and c8)).
Step (1) is entirely derived in Appendix F.
For step (2), based on the uncoupled ZPRs, we recognize that from convergence condition (h1-h7), the only condition which
impacts the form of the ZPR is (h3), because the markup terms modify the objective function of the (simplified) coupled version
of REMIND (Eq. (6)). Following similar procedure as in Appendix F, we can derive the technology-specific ZPR for the
coupled REMIND (simplified version) as follows:
$$\underbrace{\frac{\sum_y(c_{y,s} P_{y,s} + o_{y,s} G_{y,s})}{\sum_y G_{y,s}}}_{\text{Pre-curtailment LCOE}_s} + \underbrace{\frac{\sum_y(c_{y,s} P_{y,s} + o_{y,s} G_{y,s})\alpha_{y,s}}{\sum_y G_{y,s}(1 - \alpha_{y,s})}}_{\text{Curtailment cost}_s} = - \underbrace{\frac{\sum_y(\omega_{y,s} - \sigma_{y,s} + \gamma_{y,s})P_{y,s}}{\sum_y G_{y,s}(1 - \alpha_{y,s})}}_{\text{Capacity shadow price}_s} + \underbrace{\frac{\sum_y(\lambda_y + \underline{\eta}_{y,s})G_{y,s}(1 - \alpha_{y,s})}{\sum_y G_{y,s}(1 - \alpha_{y,s})}}_{\text{Market Value}_s} . \qquad (G1)$$
Compared with the ZPR of the uncoupled version (F24), the only difference is that we replace the market value in the uncoupled
REMIND $\lambda_y$ with the DIETER-markup corrected market value $\lambda_y + \underline{\eta}_{y,s}$. DIETER's ZPR is unchanged at this step.
Step (3) involves two extra capacity constraints, one in each model, the first of which, (c7), is discussed in detail in Appendix H.
The implementation of (c7) further modifies Eq. (G1) and results in the ZPRs of the coupled REMIND. The other constraint
(c8) will be the focus of discussion here. It only modifies the ZPRs for the coupled DIETER and not for the coupled REMIND.
Constraint (c8) is a brown-field capacity constraint implemented in DIETER to address the fact that DIETER is a green-field
model, which is otherwise ignorant about standing-capacities in the real world. There are many ways we can implement this
standing capacity constraint in DIETER. The most straight-forward way is to implement the "standing capacity" at the
beginning of each REMIND period, which REMIND sees before it invests additional capacities, as a lower bound on





endogenous capacities in DIETER. This helps put DIETER and REMIND on equal footing before the 5- or 10-year investment
period starts, allowing us to compare their investment intentions.

c8) "standing capacity constraint" in DIETER, i.e. DIETER capacities at time $y$ need to be larger or equal to the REMIND

standing capacities at the beginning of the time period:

$\underline{P}_s \geq P_{y-\Delta y/2,s}/(1 - ER) \qquad \perp \underline{\varsigma}_s$ ,

where the time-step $\Delta y$ is divided by 2 because the representative year in REMIND is in the middle of the time step, $ER$ is

the endogenous early retirement rate in REMIND.

The reason we implement the standing capacity in this way, is in part because as a proof-of-concept, we want to give DIETER
endogenous freedom to invest in all model years, so we use only the pre-investment capacities as "soft" corridors to bound the
DIETER capacities from below. If we were to transfer precisely the brown-field and near-term constraints from REMIND to
DIETER, it requires a complete list of constraints for each technology, and an identical implementation of all of them in
DIETER. This may raise the precision of convergence in some years for some technologies, but in practice it can be more
complicated to implement than a generic lower bound for all technologies.
To obtain the ZPRs of coupled DIETER, we simply modify the capacity shadow price term of the uncoupled DIETER ZPRs
(Eqs. (F24)-(F25)) by the additional capacity shadow price $\underline{\varsigma}_s$ from (c8):
$\underline{Capacity\ shadowprice'}_s = \frac{(\omega_s + \underline{\varsigma}_s)\,\underline{P}_s}{\sum_h \underline{G}_{h,s}}$,                                                                (G2)
$\underline{Capacity\ shadowprice'}_{system} = \frac{\sum_s (\omega_s + \underline{\varsigma}_s)\,\underline{P}_s}{\sum_{h,s} \underline{G}_{h,s}}$ .                                       (G3)
**Appendix H: Additional methods for numerical stability in coupled runs**
Here, we introduce the two methods we employed to improve numerical stability of the coupled runs: 1) the dispatchable
capacity constraint by peak demand to avoid high markups being exchanged (Sect. H.1); 2), endogenous prefactors for all
quantities from last-iteration DIETER to current-iteration REMIND (Sect. H.2).
**H.1 Dispatchable capacity constraints by peak demand**
**H.1.1 Description of the capacity constraint and price manipulation in DIETER post-processing**
Scarcity hour price can occur in a PSM run, which is the highest hourly price in a year, and it is usually equal to the annuitized
fixed cost of Open Cycle Gas Turbine (OCGT) (capital investment cost and fixed O&M costs) (Hirth and Ueckerdt, 2013). In
our simulations, the scarcity prices are usually above 50$/KWh. If we include the scarcity price into the markups, OCGT will
receive an annual markup usually more than 5 times higher than the annual average electricity price. The high markup results in
OCGT plants receiving too high an incentive in the next iteration REMIND, and the model overshoots (overinvests) in
capacities. Over iterations, this causes oscillations in the quantity and prices in the coupled model. For better numerical stability,
instead of passing on the full markups from DIETER, we only pass on the portion of the annual markups unrelated to scarcity
hour prices, and replace the exchange of the part of the markup due to scarcity hours from DIETER to REMIND with
implementing an additional capacity constraint in REMIND for coupled runs. The two actions can be later shown to be
mathematically equivalent. Generators other than OCGT which produce at the scarcity hours also get paid in the hour at this
high price. However, because they also produce at other hours with lower prices, their average market values are only
moderately impacted by the scarcity hour price, and do not in general lead to instability issues.



Below, we first introduce the aforementioned capacity constraint implemented on the side of REMIND, then discuss the
corresponding manipulation of the markups in DIETER. Lastly, we show their mathematical equivalence, and state the modified
ZPR of coupled REMIND due to these actions.
The extra capacity constraint states that the sum of all dispatchable capacities needs to be at least as large as the peak residual
demand:
c7) $\sum_{dis} P_{y,dis} > d_{y,residual} \quad \perp v_{y,dis}$,
where $d_{y,residual}$ is peak residual demand in REMIND and is semi-endogenous. $d_{y,residual}$ is a function of the peak hourly
residual demand in the last iteration of DIETER $\underline{d}_{residual}(y, i - 1)$. The peak hourly residual demand in DIETER is in turn
defined as the maximum hourly amount of inflexible demand not met by wind, solar or hydro generations, and hence must
be met by dispatchable generations (under no storage conditions):
$\underline{d}_{residual} = max_h\big(\underline{d}_h - \underline{G}_{h,Solar} - \underline{G}_{h,Wind} - \underline{G}_{h,Hydro}\big).$ (H1)
$v_{y,dis}$ is the shadow price of the capacity constraint for dispatchable technology $dis$.
For the exact implementation of (c7) in coupled run, see Sect. 3.3.2, 2. Under storage implementation, in addition to the
variable renewable contribution, the hourly storage discharge is also subtracted from the residual demand.
Simultaneous to implementing this capacity constraint, we remove the surplus scarcity prices in post-processing of DIETER
before passing it onto REMIND. In DIETER, we define the scarcity price as the maximum hourly price in a year:
$\underline{\lambda}_{y,h_{scar}} = max_h(\underline{\lambda}_{y,h})$, (H2)
and the surplus scarcity hour price is the difference between the scarcity price and the second highest price:
$\underline{\lambda}_{y,surplus} = \underline{\lambda}_{y,h_{scar}} - max(\underline{\lambda}_{y,h|h \neq h_{scar}}) = max_h(\underline{\lambda}_{y,h}) - max(\underline{\lambda}_{y,h|h \neq h_{scar}}),$ (H3)
where $h_{scar}$ is the scarcity hour when scarcity price occurs, corresponding to the peak residual demand hour.
Using this, we manipulate the market value and annual average electricity price in DIETER ex post, excluding the surplus
scarcity hour price:
$\underline{MV}'_s = \dfrac{\sum_{h|\ h \neq h_{scar}} \underline{G}_{h,s}\underline{\lambda}_h + \sum_{h|\ h_{scar}} \underline{G}_{h,s} * max(\underline{\lambda}_{h|h \neq h_{scar}})}{\sum_{h=1}^{8760} \underline{G}_{h,s}},$ (H4)
$\underline{J}' = \dfrac{\sum_{h|\ h \neq h_{scar}} \underline{d}_h\underline{\lambda}_h + \sum_{h|\ h_{scar}} \underline{d}_h * max(\underline{\lambda}_{h|h \neq h_{scar}})}{\sum_{h=1}^{8760} \underline{d}_h}.$ (H5)
where $\underline{MV}'_s$ is the annual average market value without the surplus scarcity hour price, and $\underline{J}'$ is the annual average electricity
price without the surplus scarcity hour price. Thus, the corresponding modified markup term without the surplus scarcity hour
price is:
$\underline{\eta}'_s = \underline{MV}'_s - \underline{J}'.$ (H6)
Note that since the above manipulation is done in a post-processing step, the LCOE in DIETER is still fully covered by MV, as
the KKT conditions and ZPRs still hold by default in an optimized DIETER model.
With the implementation of (c7), the coupled ZPR (Eq. (G1)) is then further modified to include the new shadow price $v_{y,s}$ as
well as the modified markup $\underline{\eta}'_{y,s}$ (without surplus scarcity price). (We write from now on $v_{y,dis}$ simply as $v_{y,s}$.) Then,
technology-specific ZPR of coupled REMIND is:
$\underbrace{\dfrac{\sum_y(c_{y,s}P_{y,s} + o_{y,s}G_{y,s})}{\sum_y G_{y,s}}}_{\text{Pre-curtailment LCOE}_s} + \underbrace{\dfrac{\sum_y(c_{y,s}P_{y,s} + o_{y,s}G_{y,s})\alpha_{y,s}}{\sum_y G_{y,s}(1-\alpha_{y,s})}}_{\text{Curtailment LCOE}_s} = -\underbrace{\dfrac{\sum_y(\omega_{y,s} - \sigma_{y,s} + \gamma_{y,s} + v_{y,s})P_{y,s}}{\sum_y G_{y,s}(1-\alpha_{y,s})}}_{\text{Capacity shadow price}'_s} + \underbrace{\dfrac{\sum_y(\lambda_y + \underline{\eta}'_{y,s})G_{y,s}(1-\alpha_{y,s})}{\sum_y G_{y,s}(1-\alpha_{y,s})}}_{\text{Market Value}'_s}$ (H7)
System ZPR of coupled REMIND is:





$$\underbrace{\frac{\sum_{y,s}(c_{y,s}P_{y,s}+o_{y,s}G_{y,s})}{\sum_{y,s}G_{y,s}}}_{\text{Pre-curtailment LCOE}_{\text{system}}} + \underbrace{\frac{\sum_{y,s}(c_{y,s}P_{y,s}+o_{y,s}G_{y,s})\alpha_{y,s}}{\sum_{y,s}G_{y,s}(1-\alpha_{y,s})}}_{\text{Curtailment cost}_{\text{system}}} = -\underbrace{\frac{\sum_{y,s}(\omega_{y,s}-\sigma_{y,s}+\gamma_{y,s}+\nu_{y,s})P_{y,s}}{\sum_{y,s}G_{y,s}(1-\alpha_{y,s})}}_{\text{Capacity shadow price}'_{\text{system}}} + \underbrace{\frac{\sum_{y,s}(\lambda_y+\underline{\eta}'_{y,s})G_{y,s}(1-\alpha_{y,s})}{\sum_{y,s}G_{y,s}(1-\alpha_{y,s})}}_{\text{Electricity Price}'_{\text{system}}}$$ (H8)

These are the ZPRs of the coupled REMIND for the full version.
**H.1.2 Equivalence between surplus scarcity price in DIETER and capacity shadow price due to peak residual demand in**
**REMIND**
Because of the intuitive relation between the scarcity price and the peak residual demand – i.e., that scarcity price occurs in the
hour with peak hourly residual demand due to the pricing power of the peaker gas turbines in the hour where VRE is most
scarce, we can draw a quantitative equivalence between the scarcity price contribution to the markup and the capacity constraint
shadow price $\nu_y$. This means that the revenue the plant receives in scarcity hour in capacity terms (i.e. capacity credit), can be
transformed directly to a revenue in energy terms (i.e. a part of the annual market value). At convergence, for any given year $y$,
the negative shadow price, $-\nu_{y,dis}$, when translated into annual generation terms via capacity factor $\phi_{y,s}$ of dispatchable
technology $s$, should be equal to the scarcity hour surplus revenue divided by annual generation by $s$ in DIETER:
$$\frac{-\nu_{y,dis}}{\phi_{y,dis}*8760} = \frac{\underline{\lambda}_{y,surplus}\underline{G}_{h_{scar},dis}}{\sum_h \underline{G}_{y,h,dis}}.$$ (H9)
In practice, this equivalence is confirmed by numerical results (e.g. Fig. 8 subplot for OCGT).
Using this equivalence, we can show as follows, that at convergence, $\lambda_y$ should be equal to DIETER power price without
surplus scarcity price $\underline{J}'$ (Eq. (H5)), and $\lambda_y + \underline{\eta}'_{y,s}$ should be equal to DIETER market value without scarcity price $\underline{MV}'$ (Eq.
(H4)).
At convergence, the annual generations have identical solutions in the two models, i.e. $\sum_h \underline{G}_{y,h,s} = G_{y,s}(1-\alpha_{y,s})$. We plug this
and REMIND capacity factor $\phi_{y,s} = \frac{G_{y,s}(1-\alpha_{y,s})}{P_{y,s}*8760}$ into Eq. (H9) to obtain
$$\nu_y P_{y,s} = \underline{\lambda}_{y,surplus}\underline{G}_{y,h_{scar},s}.$$ (H10)
Take Eq. (H7), and only consider REMIND annual revenue by multiplying generation $\sum_y G_{y,s}(1-\alpha_{y,s})$ then on the right-
hand-side, take both revenue and the capacity shadow revenue contribution from $\nu_{y,s}$ for a single year, which is equal to the total
single-year REMIND revenue:
$$\Theta_{y,s} = -\underbrace{\nu_{y,s}P_{y,s}}_{\text{Capacity shadow revenue from c(7)}_s} + \underbrace{\left(\lambda_y + \underline{\eta}'_{y,s}\right)G_{y,s}\left(1-\alpha_{y,s}\right)}_{\text{Generation revenue}'_s}$$
and plug in (H10), (H6),
$$\Theta_{y,s} = \underbrace{\underline{\lambda}_{y,surplus}\underline{G}_{y,h_{scar},s}}_{\text{surplus scarcity revenue in scarcity hour}_s} + \underbrace{\left(\underline{MV}'_{y,s} - \underline{J}'_y + \lambda_y\right)G_{y,s}\left(1-\alpha_{y,s}\right)}_{\text{Generation revenue}'_s}.$$
Plugging in (H4),
$$\Theta_{y,s} = \underline{\lambda}_{y,surplus}\underline{G}_{y,h_{scar},s} + \sum_{h\neq h_{scar}}\underline{G}_{y,h,s}\underline{\lambda}_{y,h} + \underline{G}_{y,h_{scar},s} * \max\left(\underline{\lambda}_{y,h|h\neq h_{scar}}\right) - \underline{J}'_y G_{y,s}\left(1-\alpha_{y,s}\right) + \lambda_y G_{y,s}\left(1-\alpha_{y,s}\right)$$
Lastly, plug in the definition for $\underline{\lambda}_{y,surplus}$ (Eq. (H3)),
$$\Theta_{y,s} = \sum_h \underline{\lambda}_{y,h}\underline{G}_{y,h,s} - \underline{J}'_y G_{y,s}\left(1-\alpha_{y,s}\right) + \lambda_y G_{y,s}\left(1-\alpha_{y,s}\right).$$ (H11)
Since the single-year revenue $\Theta_{y,s}$ in REMIND should be aligned with DIETER due to harmonization condition (h3), and the
DIETER revenue is $\underline{\Theta}_{y,s} = \sum_h \underline{\lambda}_{y,h}\underline{G}_{y,h,s}$, that means the last two terms in (H11) should sum to 0. Therefore REMIND
electricity price $\lambda_y$ should be equal to $\underline{J}'_y$.



**H.2 Stabilization techniques using prefactors**

In this Appendix, we describe the detailed implementations of prefactors for information exchanged from DIETER to REMIND.

1. Markup prefactor:

   In order to facilitate convergence in REMIND, we implement an endogenous prefactor $f_{y,s}^{\eta}$ for MV in the REMIND markup equation Eq. (5):

   $$\eta_{y,s}(i) = f_{y,s}^{\eta}(i) * \underline{MV}'_{y,s}(i-1) - \underline{J}'_y(i-1) \,. \tag{H12}$$

   The endogenous prefactor $f_{y,s}^{\eta}$ is dependent on the difference between in-iteration endogenous generation share and last-iteration DIETER generation share:

   $$f_{y,s}^{\eta}(i) = 1 - \underline{b}_{y,s}(i-1)\Delta S_{y,s}, \tag{H13}$$

   where $\underline{b}_{y,s}$ is a positive parameter, equal to the ratio between market values and average price depending on their relationship in the last iteration DIETER,

   $$\underline{b}_{y,s} = \frac{MV'_{y,s}}{\underline{J}'_y} \text{ if } \underline{MV}'_{y,s} > \underline{J}'_y \,,$$

   $$\underline{b}_{y,s} = \frac{\underline{J}'_y}{MV'_{y,s}} \text{ if } \underline{MV}'_{y,s} < \underline{J}'_y \,,$$

   and where the generation share difference across models and consecutive iteration $\Delta S_{y,s}$ is,

   $$\Delta S_{y,s} = \frac{G_{y,s}(i)\left(1-\alpha_{y,s}(i)\right)}{\sum_s[G_{y,s}(i)(1-\alpha_{y,s}(i))]} - \frac{\sum_h \underline{G}_{y,s}(i-1)}{\sum_{h,s}\underline{G}_{y,s}(i-1)} \,.$$

   The values of $\underline{b}_{y,s}$ are heuristically determined (see Sect. 6.2).

   When in-iteration REMIND solar generation share increases due to the price signal from the last-iteration DIETER market value, such that the REMIND share is larger than in the last DIETER iteration, the formula Eq. (H13) results in a prefactor smaller than one, decreasing in-iteration markup $\eta_{y,s}(i)$.

2. Peak demand prefactor:

   The peak demand in REMIND $d_{residual,y}$ depends on the last iteration DIETER peak hourly residual demand $\underline{d}_{residual}(y, i-1)$. Implementing it in constraint (c7),

   $$\sum_{dis} P_{y,dis} < d_{residual,y} * f_y^{d_{residual}}(i) \,,$$

   for iteration $i$, we use $f_y^{d_{residual}}(i)$ as a prefactor for stabilization,

   $$f_y^{d_{residual}}(i) = 1 - b_{y,peak} * \Delta S_{y,wind}.$$

   $b_{y,peak}$ is a heuristic constant dependent on $y$, $\Delta S_{y,wind}$ is the wind generation share. We use the wind generation share in the current iteration of REMIND for stabilization, because in the peak residual demand hour, there usually is some wind production for the historical year we chose (but no solar). In general, $b_{y,peak}$ is 0.5 for earlier years, and increasing to 1 for later years, under a baseline scenario. For climate scenarios, $b_{y,peak}$ is around 1.5 for less stringent scenarios, and for more stringent scenarios, it is 0.5 for earlier years, and increasing to 3 for later years.

3. Capacity factor prefactor:

   We set REMIND capacity factor $\phi_{y,dis}$ to be equal to the DIETER annual average capacity factor from the last iteration multiplied by a prefactor:

   $$\phi_{y,dis}(i) = \underline{\phi}_{dis}(y, i-1) * f_{y,s}^{\phi_{dis}}(i),$$





where DIETER annual average capacity factor is $\underline{\phi}_{dis} = \frac{\sum_h \underline{G}_{h,dis}}{\underline{P}_{dis} * 8760}$ for each year $y$. In order to facilitate convergence, a
similar prefactor $f_{y,s}^{\phi_{dis}}$ as in Eq. (H13) is implemented:
$f_{y,s}^{\phi_{dis}}(i) = 1 - 0.5\Delta S_{y,s}$    if $\underline{\phi}_{dis}(y, i-1) < 0.5$ (i.e. the plant is "peaker" or "mid-load" type in the last iteration),
$f_{y,s}^{\phi_{dis}}(i) = 1 + 0.5\Delta S_{y,s}$    if $\underline{\phi}_{dis}(y, i-1) \geq 0.5$ (i.e. the plant is "base-load" type in the last iteration),
where 0.5 is a heuristic factor.
The sign in the prefactor formula is determined based on the observation that under a system with variable renewable
generations, for generator plants that have relatively high running cost and low investment cost, i.e. they are most
economically operated as "peaker" plants or as "mid-load" plants of lower capacity factor, so when their generation share
incrementally increases, their capacity factor decreases. Conversely, for generators with relatively low running cost and
high investment cost, i.e. they are most economically operated as "base-load" plants, when their generation share
incrementally increases, their capacity factor increases.
4.  Curtailment prefactor:
The curtailment ratio in REMIND $\alpha_{y,vre}$ is equal to last iteration DIETER curtailment ratio, multiplied by prefactor $f_{y,vre}^{\alpha}$:
$\alpha_{y,vre}(i) = \frac{\sum_h \underline{\gamma}_{h,vre(y,i-1)}}{\sum_{h,s} \underline{G}_{h,vre(y,i-1)}} * f_{y,vre}^{\alpha}(i)$ ,
where the prefactor is $f_{y,vre}^{\alpha}(i) = 1 + \Delta S_{y,vre}$.
5.  Capture price prefactor:
Similar to the case of markup from the demand side, the markup for any demand-side technology given to REMIND is:
$\eta_{y,s_d}(i) = f_{y,s_d}^{\eta}(i) * \underline{CP}_{y,s_d}(i-1) - \underline{J}_y(i-1)$ ,
where $\underline{J}_y$ is the annual average electricity price of all demand types $s_d$ for period $y$,
$\underline{J} = \frac{\sum_h \left( \sum_{s_d} \underline{d}_{h,s_d} \right) * \underline{\lambda}_h}{\sum_{h,s_d} \underline{d}_{h,s_d}}$ ,
and $f_{y,s_d}^{\eta}(i)$ is an endogenous stabilization prefactor for the flexible-demand markup based on shares of demand by $s_d$ in
total demand for each year.
**Appendix I: Derivation for equilibrium condition for REMIND in the case of additional adjustment cost**
Adjustment cost – an additional linear term in the objective function, acts as an inertia against fast or slow capacity additions or
retirement. The implementation of positive adjustment costs mimics the challenges of scaling up the supply chains and of
training new workers to do installation and construction. Adjustment costs are applied to all model time periods, so it is by
nature intertemporal. The objective function for power sector including the adjustment cost $\Xi_{y,s}$ is
$Z = \sum_{y,s}(c_{y,s} P_{y,s} + o_{y,s} G_{y,s} + \Xi_{y,s})$ ,
where $\Xi_{y,s}$ is a quadratic function of the difference between capacity additions of subsequent time periods $y - \Delta y$ and $y$:
$\Xi_{y,s} = c_{y,s} k_s \left( \frac{\Delta P_{y,s} - \Delta P_{y - \Delta y,s}}{\Delta y^2} \right)^2 / \left( \frac{\Delta P_{y - \Delta y,s}}{\Delta y} + \beta_{y,s} \right)$,
where $\Delta P_{y,s}$ is as before the capacity addition during time period $y$ of technology $s$, $\beta_{y,s}$ is an offset parameter to offset additions
in initial time periods, $k_s$ is a regional technological coefficient, $c_{y,s}$ is the capital expenditure cost per capacity unit as before.
Because the adjustment cost is a quadratic function of the endogenous variable $P_{y,s}$, it turns the power sector cost minimization
in REMIND into a nonlinear problem.




Similar to the case without adjustment costs in Sect. 3.2.3, the first stationary condition becomes:
$\frac{\partial \mathcal{L}}{\partial P_{y,s}} = 0, \Rightarrow c_{y,s} + \omega_{y,s} - \mu_{y,s}\phi_{y,s} - \sigma_{y,s} + \gamma_{y,s} + 2c_{y,s}k_s \frac{\Delta P_{y,s} - \Delta P_{y-\Delta y,s}}{(\Delta P_{y-\Delta y,s} + \beta_{y,s})\Delta^2} = 0$ ,
simplifying,
$c_{y,s} = -\omega_{y,s} + \mu_{y,s}\phi_{y,s} + \sigma_{y,s} - \gamma_{y,s} - a_{y,s}c_{y,s}$ ,
where $a_{y,s} = 2k_s \frac{\Delta P_{y,s} - \Delta P_{y-\Delta y,s}}{(\Delta P_{y-\Delta y,s} + \beta_{y,s})\Delta y^2}$ is the endogenous adjustment factor of investment, and is a function of capacity.
The new ZPR including the adjustment cost in terms of cost and revenue for technology $s$, can be derived
$\sum_y[(c_{y,s} + a_{y,s}c_{y,s})P_{y,s} + o_{y,s}G_{y,s} + \lambda_y \alpha_{y,s}G_{y,s} + (\omega_{y,s} - \sigma_{y,s} + \gamma_{y,s})P_{y,s}] = \sum_y(\lambda_y G_{y,s})$ .
The adjustment cost $a_{y,s} c_{y,s}$ can act as a disincentive or an incentive to capacity additions. If capacity addition in the current
period is higher than in the last period $\Delta P_{y,s} > \Delta P_{y-\Delta y,s}$, i.e. a ramp-up case of capacity addition, the adjustment cost is positive
and acts as a disincentive, so the ramp-up speed is slower. When added capacities are decreasing with time, i.e. a ramp-down
case of capacity addition, adjustment cost is negative and acts as an incentive, and as a result, the ramp-down speed is slower.
In the coupled run we see only a moderate adjustment cost which drops down fast as a function of time (see e.g. Fig.6).
**Appendix J: Comparing the coupled and uncoupled run**

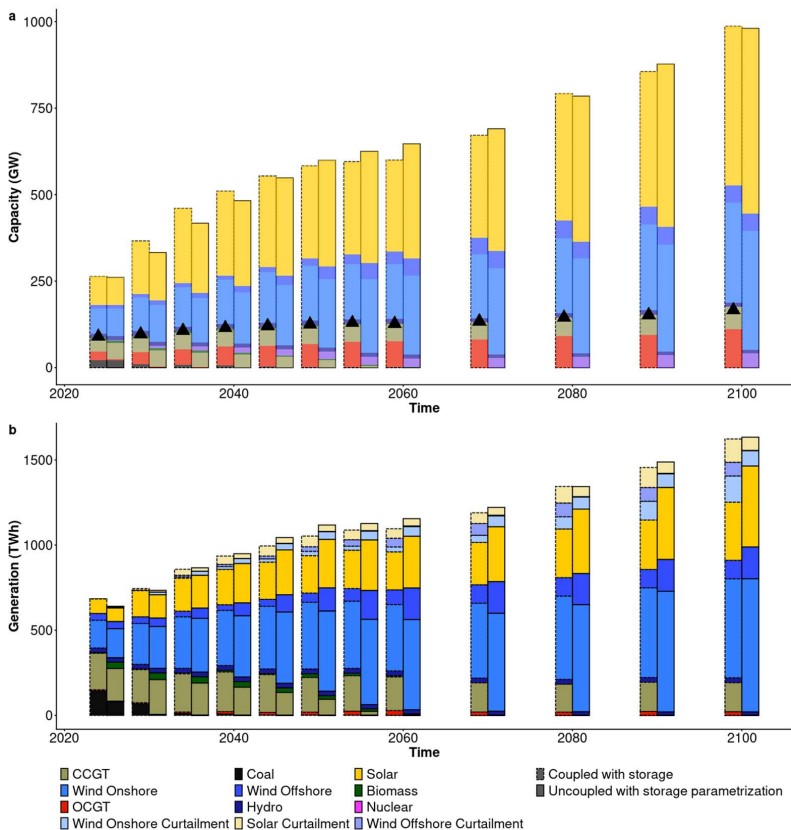




**Figure J1: Under the 2C global scenario (no German net-zero goal), we compare (a) the capacity mix and (b) the generation mix of Germany for the DIETER-coupled version of REMIND with endogenous storage (dashed bar) and for the uncoupled version of REMIND with parametrized storage (solid bar). In (a), triangle dots indicate the peak residual demand of the year as determined in DIETER.**

**Appendix K: Complete list of mathematical symbols**

The units used in the two models are usually different. Here we uniformly use MWh for energy units, and MW for capacity units. In the main text, underscore $\_$ is used to denote DIETER parameters and variables. An apostrophe is used to indicate a modified version of the variable. An asterisk is used to indicate the values of variables at the optimum of objective functions.

| Symbol | Description | Unit | Symbol | Description | Unit |
|---|---|---|---|---|---|
| $y, \Delta y$ | REMIND time period, REMIND time step | - | $h$ | Hour | - |
| $s$ | Supply-side technology type | - | $dis, vre$ | Dispatchable generators, Variable Renewable | - |
| $s_d$ | Demand-side technology type | - | $i$ | Iteration | - |
| $reg$ | Region | - | $\mathcal{L}$ | Lagrangian | $ |
| $Z$ | Objective function | $ | $G$ | Generation | MWh |
| $c$ | Fixed cost | $/MW | $\psi$ | Total annual renewable potential | MWh |
| $o$ | Variable cost | $/MWh | $\phi$ | Capacity factor | 1 |
| $\alpha$ | Annual curtailment to pre-curtailment generation ratio in REMIND model | 1 | $d$ | Exogenous demand | MWh |
| $P$ | Capacity | MW | $p$ | Standing capacity in REMIND | MW |
| $\Gamma$ | Curtailment | MWh | $\eta$ | Markup | $/MWh |
| $\lambda$ | Shadow price of power supply-demand balance equation / power price | $/MWh | MV | Market value | $/MWh |
| $q$ | Near-term ramp up constraint for capacities in REMIND | MW | $\theta$ | revenue | $ |
| $M$ | Difference in total revenues in the two models | $ | $\xi$ | Shadow price due to positive generation | $/MWh |





| $\omega$ | Shadow price due to limited renewable potential | \$/MW | $\gamma$ | Shadow price due to near-term ramp up constraint | \$/MW |
|---|---|---|---|---|---|
| $\mu$ | Shadow price due to limit on generation from capacity | \$/MWh | $\varsigma$ | DIETER shadow price due to standing capacity constraint from REMIND | \$/MW |
| $\sigma$ | Shadow price due to standing capacities in REMIND | \$/MW | CP | Capture price of demand-side technologies | \$/MWh |
| $\upsilon$ | Shadow price due to peak residual demand constraint | \$/MWh | $\Delta S$ | Difference in generation shares between models | 1 |
| $f$ | Prefactor for numeric stabilization | 1 | $W$ | Economic welfare | - |
| $b, b_{peak}$ | Multiplicative prefactors parameter | 1 | $\varrho$ | Pure rate of time preference | 1 |
| $\Xi$ | Adjustment cost | \$ | $\beta$ | Offset parameters in adjustment cost | \$ |
| $\chi$ | Consumption | \$ | $a$ | Adjustment factor of investment | 1 |
| $V$ | Population | 1 | $k$ | Regional technological coefficient for adjustment cost | 1 |
| ER | Early retirement rate in REMIND | 1 | $J$ | Annual average DIETER electricity price | \$/MWh |

**Table K1: Complete list of mathematical symbols. For simplicity, in general, we only list the symbols, not their indices or in which model they are used.**

**Appendix L: Complete list of abbreviations**

| Abbreviation | Description | Abbreviation | Description |
|---|---|---|---|
| IAM | Integrated assessment model | LCOE | Levelized cost of electricity |
| PSM | Power sector model | MV | Market value |
| VRE | Variable renewable | O&M | Operation and maintenance |
| GHG | Greenhouse gas | OMF | Operation and maintenance fixed cost |
| NLP | Nonlinear programming | OMV | Operation and maintenance variable cost |
| LP | Linear programming | OCGT | Open cycle gas turbine |



| CES | Constant elasticity of substitution | CCGT | Combined cycle gas turbine |
|---|---|---|---|
| IPCC | Intergovernmental Panel on Climate Change | CP | Capture price |
| RLDC | Residual load duration curve | PtG | Power-to-Gas |
| ZPR | Zero-profit rule | PDC | Price duration curves |
| KKT | Karush–Kuhn–Tucker | CCS | Carbon capture and storage |
| EV | Electric Vehicles | GAMS | General Algebraic Modeling System |

**Table L1: Complete list of abbreviations.**

**Code and data availability:** The coupled and uncoupled REMIND code are implemented in GAMS, and the code and data management is done using R. The coupled and the uncoupled DIETER are entirely implemented in GAMS. The default uncoupled REMIND v3.0.0 code is available from the GitHub website: https://github.com/remindmodel/remind (last access: 1 September 2022), and is archived on Zenodo under the GNU Affero General Public License, version 3 (AGPLv3) (Luderer et al., 2022b). The technical model documentation is available under https://rse.pik-potsdam.de/doc/remind/3.0.0/ (last access: 1 September 2022). The coupled version of REMIND is available from https://github.com/cchrisgong/remind-coupling-dieter/tree/couple (last access: 2 September 2022); coupled DIETER is available from: https://github.com/cchrisgong/dieter-coupling-remind (last access: 2 September 2022). The two sets of coupling codes are archived at Zenodo under Creative Commons Attribution 4.0 International License (Luderer et al., 2022c). The GAMS code, results, and scripts to produce the figures shown in this paper are archived at Zenodo (Gong, 2022).

**Author contribution**: Methodology development was done by CG, FU, and RP. CG designed and carried out the numerical implementation, and performed theoretical analysis of the methodology. The methodology was first conceptualized by GL. Supervision and funding acquisition were carried out by FU and GL. OA participated in development of model post-processing and the overall structuring of the manuscript. MK and WPS performed theoretical and conceptual validation of the manuscript. CG prepared the manuscript with contributions from all co-authors.

**Competing interests**: The authors declare that they have no conflict of interest.

**Acknowledgement:** The authors thank Professor Dr. Tom Brown at The Technical University of Berlin, and Dr. Marian Leimbach, Dr. Renato Rodrigues, Dr. Nico Bauer at the Potsdam Institute for Climate Impact Research for discussion. We gratefully received financial support by the German Federal Ministry of Education and Research (BMBF) via the project Kopernikus-Ariadne (FKZ 03SFK5N0, FKZ 03SFK5A) and via the INTEGRATE project (FKZ 01LP1928A). We also received financial support from the German Federal Environmental Foundation (Deutsche Bundesstiftung Umwelt).

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
