# Peer review of "Bidirectional coupling of a long-term integrated assessment model"

_EGUsphere, 2022_

## Author Response (AR1)

**First reviewer**

Gong et al. is an excellent and important article addressing urgent research needs in the integrated assessment modeling (IAM) community. IAMs results are heavily used in IPCC reports, informing global and regional mitigation pathways and energy systems projections. Power system planning and capacity expansion is a key for system-wide deep decarbonization. Driven by the rapid decline of renewable costs and the increasing role of energy storage, higher spatial and temporal resolution and feedback are necessary to inform a reasonable capacity expansion pathway. However, IAMs typically run every 5- or 10-year modeling period. In contrast, power system operations details need to be resolved at much finer scales (such as hours) to capture meaningful dynamics, especially related to intermediate capacity dispatch and energy storage operation. The authors proposed a solution to address this bottleneck via a bidirectional coupling between a well-established IAM and an hourly power sector model, allowing bidirectional feedback between the two models. Authors leveraged the advantages of both models and, more importantly, provided theoretical bases for their approach. I appreciate authors provided extensive details about their modeling approaches and designed different thought and numerical experiments to enhance the readability.

The authors thank the reviewer for the positive summary of the paper and comments.

I think this paper is suitable for publication, but I appreciate authors could elaborate a little more on the following aspects:

1) This study mainly compared results between REMIND and DIETER, demonstrating good consistency. However, It could be helpful to also compare with other studies/scenarios for some simple metrics, such as the total capacity or generation in a net-zero scenario. One primary application and improvement through this study is to enhance REMIND's capability for long-term system planning, so it's straightforward to compare with existing literature and show the literature range.

We are thankful for the reviewer's suggestions. For net-zero scenarios, one of the most up-to-date model comparison using state-of-the-art models for Germany is the report of the flagship project "Ariadne" funded by the German ministry for Education and Research (BMBF) (https://ariadneprojekt.de/media/2022/03/2022-03-16-Big5-Szenarienvergleich_final.pdf, page 16-18, accessed 16.04.2023). The modeled results in our paper are closest to the "Ariadne-REMIND-Mix" scenario for net-zero Germany (using uncoupled REMIND), and to the "Ariadne-REMod-Mix" (using the energy system model REMod). However, some differences remain, which we explain below.

For generation and capacity of solar, onshore wind and offshore wind in 2045, the following comparison can be made:

| Generation (2045) | solar PV (TWh) | wind onshore (TWh) | wind offshore (TWh) | total (TWh) |
|---|---|---|---|---|
| this study (Fig 10) | 600 | 592 | 92 | 1344 |
| Ariadne-REMIND-Mix | 325 | 582 | 114 | 1100 |
| Ariadne-REMod-Mix | 473 | 545 | 360 | 1487 |

| Capacity (2045) | solar PV (GW) | wind onshore (GW) | wind offshore (GW) |
|---|---|---|---|
| this study (Fig 10) | 750 | 232 | 28 |
| Ariadne-REMIND-Mix | 329 | 218 | 29 |
| Ariadne-REMod-Mix | 456 | 216 | 75 |

Compared to uncoupled REMIND, the largest difference can be observed for total generation, solar PV capacity and solar PV generation. This is likely due to the fact that uncoupled REMIND only has parametrized power sector investment and dispatch, so the electricity price is higher than in coupled REMIND, where the price is calculated via iteration with DIETER. Lower electricity prices in REMIND incentivizes more power usage. Since power is more competitive compared to other types of energy carriers, total generation is higher in the coupled version in our paper than "Ariadne-REMIND-Mix" which uses uncoupled REMIND. This could be due to the fact that in the uncoupled REMIND, solar PV is parametrized to be integrated at higher cost than in DIETER, where solar generation is explicitly modeled. Another reason could be due to the fact that solar is the cheapest form of electricity in the model per unit generation, as total power demand increases, solar PV is disproportionately used more. This is why we observe more solar in the coupled version than in the uncoupled version. For reference, REMod, which is an hourly resolution energy system model, has results which differ with both coupled and uncoupled versions of REMIND, mostly in offshore wind. This difference likely comes from different assumptions about technology cost and resource availability as well as modeling approaches between REMIND and REMod. However, the metrics are broadly consistent and also comparable to other models in the Ariadne project report.

We added this in the main text under Section 5.1.2, and in Supplemental material S4.

2) In Appendix E, I found the solution time for the coupled run, taking 6-10 hours for a detailed configuration under a climate policy. Understandably, bidirectional coupling takes time to solve. Still, I'm concerned that this may indicate a significant barrier to moving forward someday when more regions are added or improve German with sub-national details. Plus, with more regions, power system transmission has to be considered (I think the authors already noted this). This coupled modeling framework will ultimately be limited by computation capacity despite a well-presented theory behind the model. Therefore, a section should be devoted to discussing a little more about how the authors would envision a solution for increasing computational cost.

We thank the reviewer for their valuable suggestion.

Even though via soft-coupling IAM can obtain hourly resolution with only a moderate computational cost increase, it nevertheless increases the complexity of the whole problem, increasing the solver time of the IAM, especially before convergence is reached under the iteration with a PSM. With additional complexity of endogenous climate policies, computational time can be long for scenarios under climate constraint (see Appendix E). This can be potentially overcome by several measures, which can be the topics for future research:

1) Optimize for computational costs in individual models. Individual IAM and PSM are usually developed incrementally, which results over time in less overall computational efficiency. However, because individually the models are not too costly to run, there is less incentive to manage computational cost when they are run as standalone models. However, when coupled, the computational cost may become a barrier. One of the easiest ways to reduce coupled run time is to reduce run times of the individual coupled models. Because the soft-coupling takes many iterations, a small reduction in computational time in either model will multiply to give a large reduction in iteratively soft-coupled runs.

2) Other internal iterations of the IAM (if they exist) can be optimized. For example, in REMIND, most of the iterations (usually 30-50 iterations) in the coupled runs are dedicated to converging inter-regional trade between the 21 regions in the model, because DIETER iteration converges usually quite fast (5-10 iterations). By making the algorithm for the convergence of inter-regional trade faster, we can reduce total coupled iterations, therefore reducing overall computational cost. Less computational time can also be achieved, if DIETER is no longer run together with REMIND after DIETER-REMIND iteration convergence is reached, and when trade adjustment (or other internal adjustments in REMIND) is small enough to not have substantial impact on the power sector results. This is especially the case if PSM gets more complex and its computational time exceeds far more than single-iteration REMIND

time (also see Appendix E for the contribution to runtime due to REMIND internal iteration and due to PSM).

3) Limiting endogenous investments of capacities of certain technologies only in one model. For example, in the case of electricity transmission, more than one region (e.g. Germany with neighboring European countries) will need to be hard-coupled together in the PSM, which naturally increases computational cost of the PSM. But when the solutions are passed to the IAM, the regions can again be parallelized, as long as IAM does not engage in the endogenous investment of the transmission capacity. Hence the increased cost of computation due to implementing transmission is only limited to PSM. This is also the case if within Germany the spatial resolution is increased.

4) Only include essential features in PSM. Some PSMs are quite detailed and complicated for the purpose of studying specific technologies and the behavior of many agents or users. To couple to IAM, PSM should consider coarse-graining or aggregating some details, while retaining the essence of the dynamics being studied. For example, to implement smart EV charging (e.g. vehicle-to-grid), modelers of PSM should create a version for coupling which aggregates the many time series of charging and discharging of EVs to only one or two time series

Faster solvers and faster supercomputers will also contribute to improving the computational efficiency of the coupled model.

We added a section 6.4 with four suggestions on how to manage computational cost.

3) To what extent the building energy demand is consistent with the weather year (and climate projections)? i.e., what are the climate scenario assumptions to determine the building cooling/heating demand?

Thank you for the question. The heating and cooling demand (final energy) in REMIND (baseline scenario) is calculated based on yearly degree days calculated on gridded daily temperature data from the ISIMIP project (https://www.pnas.org/doi/10.1073/pnas.1312330110). This calculation is carried out in an energy demand model for buildings "EDGE-B" (Levesque et al., 2018). We assume constant climate from now on into the future as REMIND generally does not include climate impacts in its current default version for consistency with other parts of the model. Extremes are not captured by averaging to obtain yearly degree days. Therefore we also don't use representative weather years or the like in REMIND. DIETER's time series data for power demand uses historical data from 2019, and therefore also does not explicitly model future demand changes due to increasing climate impact, nor investigates the impact of extreme weather variability.

This is an ongoing development for the whole IAM and energy system modeling community.

We also added this to the section on limitation.

Antoine Levesque, Robert C. Pietzcker, Lavinia Baumstark, Simon De Stercke, Arnulf Grübler, Gunnar Luderer, "How much energy will buildings consume in 2100? A global perspective within a scenario framework", Energy, Volume 148, 2018, 514-527, https://doi.org/10.1016/j.energy.2018.01.139.

4) For educational purposes, a brief overview (or a table like Table A1) of detailed capacity planning models for German (or the EU) will be helpful. In the "Current modeling approaches and limitation" section, the authors just indicated: "PSMs typically have narrower spatial and sectoral scopes and shorter time horizons, but provide higher resolutions and increased technological detail". A set of citations are provided in this statement, but general readers wouldn't necessarily know what those models are or what level of resolution and technological details are.

Thank you for the suggestion. We added a section S5 in supplemental material (table S4) to compare several PSMs and their specifications, including references to more systematic and comprehensive reviews.

5) My last comment will be more of a philosophical question (I don't know the answer myself): REMIND is an inter-temporal optimization model (perfect foresight), and DIETER is also an optimization model. With this bidirectional coupling, this paper presents a picture of "perfect power sector planning and operation", and even a near-term capacity projection would "know" the long-term net-zero goal. In reality, however, the lock-in emission by existing energy infrastructures is a known and major issue for deep decarbonization (for example, see https://www.nature.com/articles/s41586-019-1364-3). In other words, power system planning never has a "perfect foresight.". The basic modeling philosophy (whether having inter-temporal optimization) would have completely different real-world implications, for example, for the financial risk of stranded assets. Even though this is a methodological paper, I would love to hear the authors' opinion about how "perfect" our models should be to capture the real-world "imperfect" human decisions.

We thank the reviewer for this important reflection. It ties into many existing discussions related to the differences between the "ideal world" depicted in IAM and energy system modeling on the one hand and "imperfect" but realistic real-world decision making and political economy on the other (Ellenbeck and Lilliestam, 2019; Geels et al., 2016; Keppo et al., 2021; Staub-Kaminski et al., 2014; Pahle et al., 2022). It is important to acknowledge that most IPCC scenarios contain models using "perfect foresight" assumptions. These models are interpreted often as producing benchmark scenarios that guide policy decisions on technology support and carbon pricing. Furthermore, perfect foresight assumptions are relaxed for example when "delayed action scenarios" are analyzed. Here two phases are distinguished: i) a near-term future of implemented national policies and ii) a mid-to-long term future (e.g. >2030) with a sudden change to very ambitious mitigation policies that try reaching global climate targets.

In addition, it makes sense to compare perfect foresight models to so-called "myopic models", where agents have limited time horizons. Considering perfect foresight models such as REMIND dominate IPCC model results, it is especially important to understand the differences between the two approaches. Such work has been carried out in studies such as

Fuso Nerini et al., 2017; Sitarz et al., 2023. If myopia is introduced in the model, the climate policy exemplified by carbon prices still follows an increasing expectation for more and more stringent climate policies, but the trajectory can be less smooth, and in the near-term looks more "flat", hence inducing the lock-in effect the reviewer mentioned. Without quantitative methodologies developed to model both types of behavior, we cannot obtain quantitative differences between the two, and try to understand what additional policies are needed to ensure agents behave less myopically, and the expectation about future climate policies can be made more certain.

In addition, I think it is important to realize that ultimately no models are perfect. Even in real-world policy making, not all policy makers' decisions are based on short-term constraints, and some decisions are indeed made more long-term, usually by "patient capital" which has more tolerance for risks, such as governments, state-owned banks and public financial entities. This is why despite the high cost at the time, German government around early 2000 heavily encouraged the use of renewable energies using climate protection policies. This mass adoption of PV solar (and to a lesser extent wind) generation technologies eventually lowered their learning curves, making them cheaper (Buchholz et al., 2019). The German government preemptively acted to foster technologies and lower their production cost via economies of scale, which lowers overall mitigation cost when considering the decades following this action. Implementing endogenous learning using perfect foresight helps our scenarios to imitate this type of long-term strategic thinking.

A shortened version of this reply has been added under point 8) under sec 6.3 on limitation of coupled results.

Ellenbeck, S. and Lilliestam, J., "How modelers construct energy costs: Discursive elements in Energy System and Integrated Assessment Models", Energy Research & Social Science, 47, 69-77 (2019). https://doi.org/10.1016/j.erss.2018.08.021

Geels, F., Berkhout, F. and van Vuuren, D. "Bridging analytical approaches for low-carbon transitions". Nature Climate Change **6**, 576–583 (2016). https://doi.org/10.1038/nclimate2980

Staub-Kaminski, I., Zimmer, A., Jakob, M. and Marschinski, R. "Climate policy in practice: a typology of obstacles and implications for integrated assessment modeling." Climate Change Economics 05(01):1440004 (2014).

Sitarz, J., Pahle, M., Osorio, S., Luderer, G. and Pietzcker, R. "EU carbon prices signal high policy credibility and farsighted actors", preprint , 2023. https://doi.org/10.21203/rs.3.rs-2761645/v1

Fuso Nerini, F., Keppo, I., Strachan, N., 2017. Myopic decision making in energy system decarbonisation pathways. A UK case study. Energy Strategy Reviews 17, 19–26. https://doi.org/10.1016/j.esr.2017.06.001

Pahle, M., Tietjen, O., Osorio, S., Egli, F., Steffen, B., Schmidt, T. S. and Edenhofer O. Safeguarding the energy transition against political backlash to carbon markets. *Nat Energy* **7**, 290–296 (2022). https://doi.org/10.1038/s41560-022-00984-0

Buchholz, W., Dippl, L. and Eichenseer, M. "Subsidizing renewables as part of taking leadership in international climate policy: The German case", Energy Policy, 129, 765-773 (2019). https://doi.org/10.1016/j.enpol.2019.02.044.

**Minor**

1) In Figure 4a, I suggest changing the color scheme into a two-color version (red and blue) to indicate positive and negative price differences.

This has been corrected as the reviewer suggests.

2) In Figure 13, the y-axis labels in panels a and b overlap.

This has been corrected as the reviewer suggests.

3) Figure 13, panel c, why is there some discontinuity between the first two modes on the left and in the right-most few hours?

There are some switching behaviors between more and less battery discharging over summer nights, as well as PtG H2 being produced at consecutively discontinuous levels over the same period as battery discharge. The authors believe these are minor artifacts due to the fact that there are no ramping costs applied to electrolyzers. In reality, modern electrolyzers are quite flexible, so potentially no ramping costs need to be applied in a model with an hourly resolution. In future research, potentially such discontinuities can be smoothed over to create better visualizations.
A short sentence has been added to Fig 13 caption to explain this.

**Second reviewer**

This is a proof-of-concept study presenting a new method of soft-coupling (or linking) an IAM with a coarse temporal and spatial resolution and a power system model with an hourly temporal resolution for Germany.

The authors address an important area of research as it is acknowledged that improving the technical representation of power systems with high levels of renewables within IAM is needed from a policy and technical perspective.  From a technical perspective, the coupling methodology maps Karush–Kuhn–Tucker Lagrangians of reduced versions of both the IAM and power model

The coupling method has been demonstrated to attain a high degree of convergence between models and the equations and method presented seem reasonable.

The limitations of the methodology are well articulated in terms of legacy issues spilling over from IAM assumptions in existing plant, neighbour effects from broader regions in REMIND not represented in DIETER/copperplate/limited weather years etc.

the converged results of the two models are compared in the study by examining the long-term capacity and generation power mix , prices of electricity,  dispatch and are shown

to find good agreement (tolerances are presented within the study) between the two models at the end of convergence.

We thank the reviewer for their positive summary and reaction towards our study.

The portfolio results are not remarkable or surprising and reflect the proof-of-concept nature of the study rather than any explicit policy message. The least-cost pathway under net zero scenario shown to have very high levels of wind and solar, storage and decarbonised gas as a back up. The results are likely constrained somewhat by the limited options for decarbonisation within the case study. Nuclear is exogenously forced, limited offshore wind capacity and limited number of non V-RES alternatives and this is not a weakness of the method presented but a consequence of how they were applied.

We agree with the reviewer, that the resulting mix is largely due to limited options within the available energy portfolio due to Germany's energy policy and natural resources. In future research, we would like to apply the same method to other world regions, where technology options may be less constrained. This has been added to Sec. 6.3 (9).

The authors outline a vision for future work which would add value to the presented concepts, namely the broadening of the methodology to vectors other than electricity such as heat and perhaps a wider sector coupling. It would be beneficial to the wider community to develop the methodology to have a more realistic representation of grid/transmission development as this is likely influencing the results for storage. Currently a simplified grid capacity equation is used that seems to be linked to the % of V-RES curtailment and it not clear how sensitive results are to this assumption.

The reviewer is correct in pointing out that the methodology presented in this work would benefit from being expanded to include a realistic representation of grid and transmission. Currently our team is working on applying the same coupling method to REMIND and PyPSA-Eur, which contains a detailed representation of the European power grid.

Currently, the grid capacity equation is parametrized to be proportional to pre-curtailment variable renewable generation, and the parametrization is rather optimistic based on PSM studies conducted in Pietzcker et al., 2017. As hinted in a recent work by Frysztacki et al., 2022, lower level of spatial detail results in an underestimation of constraints present in a real electric system, leading to an underestimation of system cost.

Frysztacki, M. M., Hörsch, J., Hagenmeyer, V. and Brown, T.: The strong effect of network resolution on electricity system models with high shares of wind and solar, Applied Energy, 291, 116726, https://doi.org/10.1016/j.apenergy.2021.116726, 2021.

Further note:
1. All references in author's reply which have not been part of the previous version have been added.
2. Recent work on soft-coupling long- and short-term energy system models have been added to the references under Sec. 1.2.
3. It has come to the attention of the authors that our proposed methodology bears certain similarities (but also differences) to Benders Decomposition method. We have added a paragraph on this under Sec. 1.2.

---

## Author Response (AR2)

Both reviewers seem to have really enjoyed your original manuscript, and you've responded well to the comments they did have. I just have a few extra, both minor:

We thank the editor for both points raised.

(1) In your response to Reviewer 1, you mention that ISIMIP climate data are used in the simulations. Please mention (and cite) this in Supplement S1, and provide details (e.g., how is "constant climate" calculated/implemented?).

The ISIMIP citation has been added in Supplement S1. And "constant climate", i.e. constant temperature, based on historical extrapolation is explained. The following paragraph is added to S1:

"The heating and cooling demand (final energy) in REMIND (baseline scenario) is calculated based on annual degree days calculated on gridded daily temperature data from the ISIMIP project (Warszawski et al., 2014). This calculation is carried out in an energy demand model for buildings "EDGE-B" (Levesque et al., 2018) then fed into REMIND. In this calculation we assume constant climate from now on into the future. Specifically, the average of the last few years of historical data which lasts until 2010 is used for future data. "

(2) Re: your response to that same comment: Please mention in #7 under Sect. 6.3 that neither extremes *nor trend* are considered. Please also mention that average annual degree-days are used, as this helps explain why extremes aren't considered.

We wrote the following under #7 in Sect 6.3:

"Climate impacts under various scenarios on building sector power demand is not included in current version of REMIND or its energy demand model for building sector "EDGE-B" (Levesque et al., 2018). Climate extremes such as heat waves are not included in either model due to the fact that annual degree days are used which are the results of temporal averaging. Representative weather years which maintain the temperature extremes and can represent long-term trends are also not used. However, the demand projection does change in a minor way based on SSP scenarios due to their different population projections."

---

## Author Response (AR3)

Thanks for your revisions. Unfortunately I'm looking for more detail than that on the climate data. If you used the ISIMIP climate forcings, there are five climate models listed in Warszawski et al. (2014)—which did you use? And how many years exactly did you use for the averaging? Note #2 on the GMD Obligations For Authors page (https://www.geoscientific-model-development.net/policies/obligations_for_authors.html): "A paper should contain sufficient detail and references to public sources of information to permit the author's peers to replicate the work."

We thank the editor for the follow-up question. We understand the necessity in addressing all the details in replicating the model results.

To answer your question, we used Global Soil Wetness Project Phase 3 (GSWP3) observational dataset from the ISIMIP project for average daily temperature, from which we use the standard definition to calculate average degree days. The observational data lasts until 2010. For data beyond 2010, we use the average of year 2006 to 2010.

We have added this detail to S1.